# DISTILLING DATASET INTO NEURAL FIELD

**Donghyeok Shin**[1], **HeeSun Bae**[1], **Gyuwon Sim**[1], **Wanmo Kang**[1] **& Il-Chul Moon**[1,2]
[1]Korea Advanced Institute of Science and Technology (KAIST), [2]summary.ai
`{tlsehdgur0,cat2507,gkwlaks4886,wanmo.kang,icmoon}@kaist.ac.kr`

## ABSTRACT

Utilizing a large-scale dataset is essential for training high-performance deep learning models, but it also comes with substantial computation and storage costs. To overcome these challenges, dataset distillation has emerged as a promising solution by compressing the large-scale dataset into a smaller synthetic dataset that retains the essential information needed for training. This paper proposes a novel parameterization framework for dataset distillation, coined Distilling Dataset into Neural Field (DDiF), which leverages the neural field to store the necessary information of the large-scale dataset. Due to the unique nature of the neural field, which takes coordinates as input and output quantity, DDiF effectively preserves the information and easily generates various shapes of data. We theoretically confirm that DDiF exhibits greater expressiveness than some previous literature when the utilized budget for a single synthetic instance is the same. Through extensive experiments, we demonstrate that DDiF achieves superior performance on several benchmark datasets, extending beyond the image domain to include video, audio, and 3D voxel. We release the code at https://github.com/aailab-kaist/DDiF.

## 1 INTRODUCTION

High performances from recent deep learning models are mainly driven by scaling laws (Bengio et al., 2007; Kaplan et al., 2020), which heavily rely on a large-scale dataset. However, utilizing the large-scale dataset incurs significant computation and storage costs. *Dataset distillation* has been proposed as a potential solution to address these challenges (Wang et al., 2018). The goal of dataset distillation is to synthesize a small-scale synthetic dataset that encapsulates the essential information needed to train a deep learning model on a large-scale dataset. Naturally, one of the research directions in dataset distillation is defining the essential information and developing efficient methods to learn it (Zhao et al., 2020; Zhao & Bilen, 2023; Cazenavette et al., 2022).

In parallel, another crucial research direction is parameterizing the small-scale synthetic dataset under the limited storage budget. The naive parameterization method constructs the synthetic instance in the same structure as the original instance. Due to the limitations of scalability and redundancy in this approach, various parameterization methods have been proposed to improve efficiency within the limited storage budget. Generally speaking, parameterization methods commonly employ low-dimensional codes and decoding functions that transform a code in reduced dimensions into a data instance of the original input space. Conceptually, the decoding functions can be categorized into 1) static decoding (Kim et al., 2022; Shin et al., 2024); 2) parameterized decoding (Deng & Russakovsky, 2022; Sachdeva et al., 2023; Lee et al., 2022; Liu et al., 2022; Wei et al., 2024); and 3) deep generative prior (Cazenavette et al., 2023; Su et al., 2024a;b; Zhong et al., 2024). Although these methods have shown promising results, they still exhibit limitations in coding efficiency, expressiveness, and applicability to diverse data structures. Additionally, there has been limited exploration of the theoretical foundations underlying these methods.

This paper introduces a new parameterization framework for dataset distillation that stores information into *synthetic neural fields* under the limited storage budget, coined **D**istilling **D**ataset **i**nto Neural **F**ield (DDiF). A *field* is a function that takes a coordinate as an input and returns a corresponding quantity, and a *neural field* parameterizes the field using a neural network. DDiF employs the synthetic neural field as a container of distilled information that utilizes a small budget, and it decodes a synthetic instance by inputting a set of coordinates corresponding to the original instance. We emphasize that the neural field has a structural difference compared to conventional decoding

functions in parameterization methods, which map a low-dimensional space to an instance-level space. Thanks to the continuous and coordinate-oriented nature of the neural field, DDiF effectively encodes information from high-dimensional data, which is a crucial challenge in dataset distillation. Furthermore, due to the inherent flexibility of neural networks (Lu et al., 2017; Raghu et al., 2017), the synthetic instance decoded by DDiF exhibits high expressiveness. In addition, DDiF is capable of encoding grid-based data from various modalities and decoding a data instance of resolution that was not encountered during the distillation process. We present a theoretical analysis of parameterization methods by investigating the expressiveness of the decoded synthetic instance through the feasible space view. Based on this theoretical understanding, we demonstrate that DDiF possesses a larger feasible space compared to some previous literature under the same utilized budget for a single synthetic instance. Across various evaluation scenarios, DDiF consistently exhibits performance improvements, generalization, robustness, and adaptability on diverse modality datasets.

In summary, our contributions are as follows:

- We propose a new parameterization framework for dataset distillation, Distilling Dataset into Neural Field (DDiF), which employs a neural field to parameterize a synthetic instance.
- We theoretically analyze the expressiveness of DDiF by investigating the feasible space covered by its decoded synthetic instances. Furthermore, we confirm that DDiF exhibits high expressiveness by comparing it with previous methods under the same utilized budget for a single synthetic instance.
- Through extensive experiments, we demonstrate that DDiF consistently achieves high performance on diverse grid-based data, including image, video, audio, and 3D voxel. Moreover, we present a new experimental design, *cross-resolution generalization*, capable of measuring generalization performance on resolutions not encountered during training.

## 2 PRELIMINARY

### 2.1 PROBLEM FORMULATION

This paper focuses on dataset distillation for classification tasks. We define a given large dataset that needs to be distilled as $\mathcal{T} := (X_\mathcal{T}, Y_\mathcal{T}) = \{(x_i, y_i)\}_{i=1}^{|\mathcal{T}|}$, where $X_\mathcal{T} := \{x_i\}_{i=1}^{|\mathcal{T}|}$ denotes a set of $D$-dimensional input data $x_i \in \mathcal{X} \subseteq \mathbb{R}^D$, and $Y_\mathcal{T} := \{y_i\}_{i=1}^{|\mathcal{T}|}$ denotes a set of corresponding labels among $C$-classes $y_i \in \mathcal{Y} = \{1, ..., C\}$. We assume that each data pair $(x_i, y_i)$ is drawn i.i.d from the distribution $P$. Let a classifier $f_\theta : \mathcal{X} \to \mathcal{Y}$ be a neural network parameterized by $\theta \in \Theta$. We also define a loss function $\ell : \mathcal{Y} \times \mathcal{Y} \to \mathbb{R}$.

The main goal of dataset distillation is to synthesize a small dataset such that a model trained on this synthetic dataset can generalize well to a large dataset. Formally, given a synthetic dataset $\mathcal{S} := (X_\mathcal{S}, Y_\mathcal{S}) = \{(\tilde{x}_j, \tilde{y}_j)\}_{j=1}^{|\mathcal{S}|}$ where $X_\mathcal{S} := \{\tilde{x}_j\}_{j=1}^{|\mathcal{S}|}$, $Y_\mathcal{S} := \{\tilde{y}_j\}_{j=1}^{|\mathcal{S}|}$, and $|\mathcal{S}| \ll |\mathcal{T}|$, the objective of dataset distillation is formulated as follows:

$$\min_\mathcal{S} \mathbb{E}_{(x,y) \sim P} \big[ \ell(f_{\theta_\mathcal{S}}(x), y) \big] \text{ subject to } \theta_\mathcal{S} = \arg\min_\theta \frac{1}{|\mathcal{S}|} \sum_{(\tilde{x}_j, \tilde{y}_j) \in \mathcal{S}} \ell\big(f_\theta(\tilde{x}_j), \tilde{y}_j\big) \quad (1)$$

Nonetheless, the optimization of Eq. (1) is both computationally intensive and lacks scalability, as it entails a bi-level optimization problem for both $\theta$ and $\mathcal{S}$ (Zhao et al., 2020; Borsos et al., 2020). To overcome these issues, several studies have suggested the surrogate objectives to effectively capture essential information needed for training the neural network on $\mathcal{T}$, such as matching gradient (Zhao et al., 2020), distribution (Zhao & Bilen, 2023), and trajectory (Cazenavette et al., 2022). For the sake of brevity, we denote these objectives as $\mathcal{L}(\mathcal{T}, \mathcal{S})$ throughout this paper.

### 2.2 PARAMETERIZATION OF DATASET DISTILLATION

Input-sized parameterization, which sets a synthetic instance in the same format as a real instance, suffers from scalability as the dimension of a given instance increases. Also, input-sized parameterization does not utilize the storage budget efficiently because it contains redundant or irrelevant information (Lei & Tao, 2023; Yu et al., 2023; Sachdeva & McAuley, 2023).

Addressing these concerns, several studies have proposed parameterization methods to enhance the efficiency of synthetic dataset. In general, parameterization methods employ 1) codes $Z := \{z_j\}_{j=1}^{|Z|}$

where $z_j \in \mathbb{R}^{D'}$ and 2) decoding function $g_\phi : \mathbb{R}^{D'} \to \mathbb{R}^D$ to construct the synthetic dataset.[1] Under this framework, a decoded synthetic instance is represented by a combination of code and decoding function i.e. $\tilde{x}_j = g_\phi(z_j)$. Also, a set of decoded synthetic instances become $X_S = \{g_\phi(z)|z \in Z\}$. Typically, parameterization methods use $Z$ and/or $\phi$ as targets for optimization and storage. Therefore, the storage budget for the parameterization is calculated based on the total number of parameters comprising $Z$ and/or $g_\phi$, and it is adjusted to ensure the budget constraint.

Based on the structure of the decoding function, they can be broadly categorized into 1) static decoding, 2) parameterized decoding, and 3) deep generative prior. Static decoding employs a non-parameterized decoding function $g$, such as resizing (Kim et al., 2022) and frequency transform (Shin et al., 2024). These methods are fast, easy to apply, and do not require a budget for the decoding function. However, since this decoding function is fixed, the structure of code $z$ becomes limited without the ability to adaptively transform $g$. Also, using a static decoding function inevitably reduces expressiveness from a generalization perspective, leading to information loss.

Parameterized decoding utilizes a linear combination with learnable coefficients (Deng & Russakovsky, 2022; Sachdeva et al., 2023), decoder (Lee et al., 2022; Zheng et al., 2023), autoencoder (Liu et al., 2022; Duan et al., 2023), or transformer structure (Wei et al., 2024) as the decoding function $g_\phi$. Although flexible decoding functions are employed, the optimized parameters of the decoding function also be stored within the limited budget, necessitating the use of a simple structure (i.e., linear combination) or a lightweight neural network. It results in limited flexibility and presents challenges when extending to complex data types, such as video, 3D, and medical images.

Deep generative prior leverages a pretrained deep generative model without additional training, focusing only on optimizing the latent vector (Cazenavette et al., 2023; Su et al., 2024a;b; Zhong et al., 2024). This framework encourages better generalization to unseen architecture and scale to the high-dimensional dataset. However, it assumes easy access to a well-trained generative model, which might restrict the range of applications. Also, since the deep generative model contains a large number of parameters, the decoding process and backward update process take a long time.

### 2.3 NEURAL FIELD REPRESENTATION

In physics and mathematics, a *field $F$* is a function that takes a coordinate in space and outputs a quantity (Xie et al., 2022). If we apply the concept of the field to data modeling, a single grid-based data can be regarded as a field. For example, an RGB image is a function that maps from pixel locations to pixel intensity. Similarly, a video datatype is a function that additionally takes time as input, and a 3D datatype is a function that outputs occupancy value on the 3D coordinate system.

According to the universal approximation theorem (Cybenko, 1989), any field can be parameterized by a neural network, which is referred to as a *neural field $F_\psi$*. It implies that grid-based data can be expressed as a neural network. Let $\mathcal{C} := \{c_i\}_{i \in \mathcal{I}}$ be a set of coordinates of grid-based data and $\mathcal{Q} := \{q_i\}_{i \in \mathcal{I}}$ be a set of corresponding quantities. To encode a single grid-based data using a neural field $F_\psi$, we minimize a distortion measure, such as squared error, over all given coordinates as:

$$\min_\psi \sum_{i \in \mathcal{I}} \|F_\psi(c_i) - q_i\|_2^2 \tag{2}$$

Recently, the neural field has been adopted to many applications, such as representation learning (Park et al., 2019; Mildenhall et al., 2021), generative modeling (Skorokhodov et al., 2021; Dupont et al., 2021), medical imaging (Shen et al., 2022; Zang et al., 2021), and robotics (Li et al., 2022).

## 3 METHODOLOGY

This section proposes a new parameterization framework for dataset distillation that stores information of a real dataset in *synthetic neural fields* under a limited storage budget, coined Distilling Dataset into Neural Field (DDiF). The core idea of this paper is to store the distilled information in the synthetic function. Although there are several candidates for the form of synthetic function, we primarily focus on (neural) *field*. Figure 1 illustrates the overview of DDiF. In the following, we begin by explaining the reasons for choosing neural field as the form of synthetic function. Then, we introduce our framework, DDiF, which parameterizes a synthetic instance using a neural field. Finally, we provide a theoretical analysis for a better understanding of parameterization and DDiF.

---

[1]Although some studies (Deng & Russakovsky, 2022; Moser et al., 2024) use both $\tilde{y}$ and $z$ as inputs for $g_\phi$, we expressed it as $g_\phi(z)$ for the sake of uniformity.

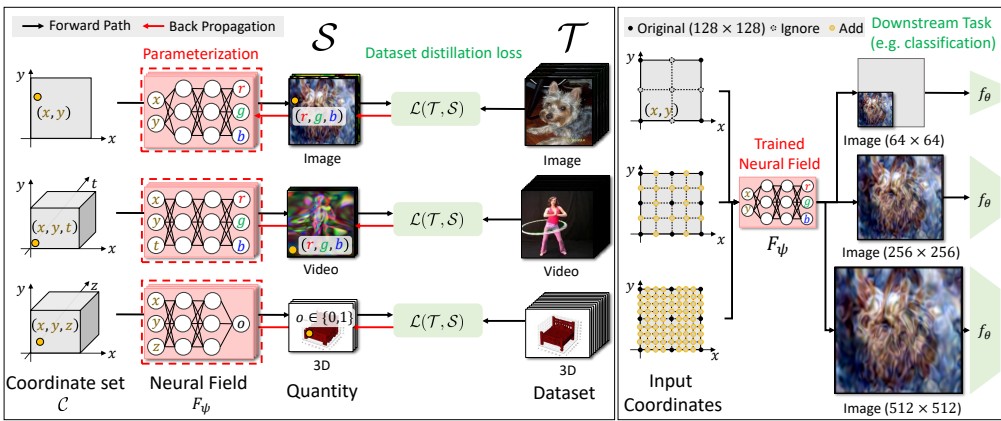

Figure 1: Overview of DDiF. Each decoded synthetic instance is constructed by the output of each synthetic neural field $F_\psi$ by inputting coordinate set $\mathcal{C}$ (left). DDiF optimizes only the parameters $\psi$, as coordinate set $\mathcal{C}$ does not require optimization or storage. Also, DDiF is capable of encoding grid-based data from various modalities. In the evaluation stage (right), DDiF can decode the data with sizes that were not encountered during the distillation stage by adjusting the input coordinates.

## 3.1 WHY NEURAL FIELD IN DATASET DISTILLATION?

Although the neural field is widely adopted, no research has yet been conducted on integrating it into dataset distillation. Herein, we provide several properties, which are beneficial for dataset distillation due to its unique structural characteristic.

**Coding Efficiency.** The neural field efficiently encodes information of high-dimensional grid-based data. Distilling the high-dimensional dataset remains a crucial challenge in expanding the applicability of dataset distillation (Lei & Tao, 2023). Input-sized parameterization typically scales poorly with resolution due to the *curse of discretization* (Mescheder, 2020). While employing the decoding function $g_\phi : \mathbb{R}^{D'} \to \mathbb{R}^D$ could improve scalability, its output ultimately depends on the data dimension $D$. It implies that as the data dimension $D$ increases, a more complex decoding function $g_\phi$ is required, which becomes problematic given a limited storage budget. In contrast, the neural field stores information independently of the data dimension $D$. Its input and output dimensions are determined by the dimensions of the coordinate space and the quantity space, respectively, as depicted in Figure 2. For high-dimensional data, the neural field only requires a larger set of input co-

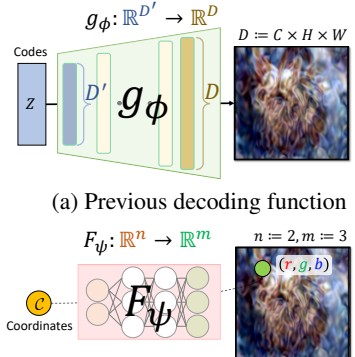

(a) Previous decoding function

(b) Neural field

Figure 2: Illustration of structural difference between conventional decoding function and neural field.

ordinates. Moreover, we emphasize that the neural field can represent complex grid-based data, such as videos and 3D voxels, which are not limited to the image domain. Since such data are typically high-dimensional, the high coding efficiency of the neural field is particularly advantageous for complex grid-based dataset distillation. Therefore, the neural field provides a unified and efficient representation of high-dimensional grid-based data.

**Resolution Adaptiveness.** The neural field is robust to diverse resolution. In the real world, it is often necessary to resize data depending on downstream tasks (Wang et al., 2020b; Shorten & Khoshgoftaar, 2019). Existing methods can only apply postprocessing on the optimized synthetic dataset, leading to insufficient or distorted information. Meanwhile, the neural field easily obtains various sizes of data by adjusting the set of input coordinates due to the continuous nature. Furthermore, since the neural field is a continuous function, it provides more accurate values for unseen coordinates. Please refer to Section 4.1 for the empirical evidence of this claim by introducing a new experiment design, coined cross-resolution generalization.

## 3.2 DDiF: Distilling Dataset into Neural Field

In this section, we introduce a basic framework for integrating the neural field into dataset distillation. Specifically, DDiF parameterizes a single synthetic instance $\tilde{x}_j$ as a single neural field $F_{\psi_j}$. DDiF consists of two main components: 1) *Coordinate set $\mathcal{C}$* and 2) *Synthetic neural fields $F_\Psi$*.

**Coordinate Set $\mathcal{C}$.** To define the function, it is first necessary to define the input space of the function. Fundamentally, a decoded synthetic instance by the parameterization method is the same shape as the real instance. Therefore, our synthetic function must be defined in the space where the information of the real instances is stored. Suppose that the given real instance $x \in X_\mathcal{T}$ is $n$-dimensional grid representation with resolution $N_k, k = 1, ..., n$, and each element contains $m$ values, i.e. $m = 3$ of RGB values. Formally, the real instance $x$ is element in $\mathbb{R}^{m \times N_1 \times \cdots \times N_n}$. Then, the coordinate set $\mathcal{C}$, where the values of $x$ are stored, is defined by a bounded set of lattice points:

$$\mathcal{C} := \left\{ (i_1, i_2, ...i_n) \Big| i_k \in \{0, 1, \cdots, N_k\}, \ \forall k = 1, ..., n \right\}$$

Note that there are several properties of coordinate set $\mathcal{C}$, which become advantages in dataset distillation. First, since every real instance $x \in X_\mathcal{T}$ is defined on the same coordinate set $\mathcal{C}$ (the only difference is the value on each coordinate), we do not need to consider the coordinate individually. Also, $\mathcal{C}$ is easily obtained if only the shape of the decoded instance is defined, without any additional information. For instance, assume that we want to get $N \times N$-shaped data instances. Then, $\mathcal{C}$ is a set of lattice points in $[0, N] \times [0, N]$. Due to these static and bounded characteristics, DDiF does not need to optimize or store the coordinate set $\mathcal{C}$. Based on this property, DDiF is a decoder-only parameterization framework that leverages a flexible decoding function $g_\phi$ without inferring codes $Z$, which structurally differs from previous methods.

**Synthetic Neural Fields $F_\Psi$.** DDiF utilizes neural field $F_\psi : \mathbb{R}^n \to \mathbb{R}^m$ to obtain the decoded synthetic instance $\tilde{x}$ by inputting the coordinate set $\mathcal{C}$.[2] Specifically, given a coordinate set $\mathcal{C}$, the decoded synthetic instance by DDiF is $\tilde{x} = F_\psi(\mathcal{C}) := [F_\psi(c)]_{c \in \mathcal{C}}$. In DDiF, since a decoded synthetic instance $\tilde{x}$ and a synthetic neural field $F_\psi$ have one-to-one correspondence, obtaining $K$ decoded synthetic instances requires $K$ synthetic neural fields. We denote the parameter set of synthetic neural fields as $\Psi := \{\psi_j\}_{j=1}^{|\Psi|}$ and the set of synthetic neural fields as $F_\Psi := \{F_{\psi_j}\}_{j=1}^{|\Psi|}$. For the structure of the synthetic neural field $F_\psi$, we follow the tradition of the neural field (Mildenhall et al., 2021; Tancik et al., 2020), which utilizes a simple $L$-layer neural network:

$$F_\psi(c) = W^{(L)}(h^{(L-1)} \circ \cdots \circ h^{(0)})(c) + b^{(L)}, \quad h^{(l)}(c) = \sigma^{(l)}(W^{(l)}c + b^{(l)})$$

where $W^{(l)} \in \mathbb{R}^{d_l \times d_{l-1}}$, $b^{(l)} \in \mathbb{R}^{d_l}$ are weights and bias at layer $l$. $\sigma^{(l)}$ denotes a nonlinear activation function. Under this formulation, $\psi$ becomes $\{W^{(l)}, b^{(l)}\}_{l=0}^L$. To avoid the spectral bias (Rahaman et al., 2019; Xu et al., 2019) that limits the expressiveness of neural fields, we employ a sine activation function (Sitzmann et al., 2020) by default, which is widely used in neural fields.

**Learning Framework.** Given a coordinate set $\mathcal{C}$ and a parameter set of synthetic neural fields $\Psi$, a set of decoded synthetic instances is represented by $X_\mathcal{S} = \{F_{\psi_j}(\mathcal{C})\}_{j=1}^{|\Psi|}$. Under the arbitrary dataset distillation loss $\mathcal{L}(\mathcal{T}, \mathcal{S})$, the overall optimization of DDiF is formulated as follows:

$$\min_\Psi \mathcal{L}(\mathcal{T}, \mathcal{S}) \quad \text{where} \quad \mathcal{S} = \left\{ (F_{\psi_j}(\mathcal{C}), \tilde{y}_j) \right\}_{j=1}^{|\Psi|} \tag{3}$$

DDiF has no limitations in applying an optimizable soft label (Sucholutsky & Schonlau, 2021; Cui et al., 2023), but we utilize predefined one-hot labels to ensure consistency with previous parameterization i.e. $\tilde{y}_j = y_j \in Y_\mathcal{T}$. Please refer to Appendix C.3 for the compatibility with soft label. In practice, dataset distillation commonly utilizes randomly sampled real instance $x \in X_\mathcal{T}$ as the initialization of synthetic instance $\tilde{x}$. In the same context, we conduct the warm-up training for synthetic neural fields $F_\Psi$. Concretely, we train $F_\Psi$ using Eq. (2) with randomly selected $|\Psi|/C$ samples for each class. Appendix B.5 specifies training procedure and decoding process of DDiF.

---

[2] We defined $\mathcal{C}$ for dataset distillation training, but the domain of $F_\psi$ is the entire $n$-dimensional space $\mathbb{R}^n$.

**Budget calculation.** As Eq. (3) demonstrates, the optimization target of DDiF is $\Psi = \{\psi_j\}_{j=1}^{|\Psi|}$. In detail, each synthetic neural field $F_\psi$ utilize $d_0(n+1) + \sum_{l=1}^{L-1} d_l(d_{l-1}+1) + m(d_{L-1}+1) =: b$ parameters. We emphasize that $b$ does not depend on the data dimension $D$, so high resolution does not necessarily increase the budget of $F_\psi$. When the storage budget is limited to $B$, we configure the structure of $F_\psi$, such as the width $d_l$ and the number of layers $L$, to satisfy $|\Psi| \times b \le B$.

## 3.3 THEORETICAL ANALYSIS

Even though several parameterization methods for dataset distillation are proposed, there has been little discussion regarding the theoretical understanding of their methods. In this section, we provide a theoretical analysis of parameterization methods by investigating the expressiveness of the decoded synthetic instance. Next, we analyze the expressiveness of DDiF when employing the sine activation function. Lastly, we demonstrate that DDiF exhibits higher expressiveness than the previous work, FreD (Shin et al., 2024), under the same utilized budget for a single synthetic instance.

We conjecture that the expressiveness of the decoded synthetic instance corresponds to the coverage of its respective data space. Accordingly, the optimization feasibility of dataset distillation depends on the coverage or *feasible space* of the decoded synthetic instance. Based on this conceptual idea, Proposition 3.1 characterizes the relationship between the feasible space of decoded synthetic instances and the optimal value of the dataset distillation objective, when the number of decoded synthetic instances is the same. Herein, we assume only synthetic inputs $X_S$ as the optimization variable. Consequently, dataset distillation loss $\mathcal{L}(\mathcal{T}, \mathcal{S})$ is simply expressed as a function of $X_S$ i.e. $\mathcal{L}(\mathcal{T}, \mathcal{S})$ is represented by $\mathcal{L}(X_S)$.

**Proposition 3.1.** *Consider two functions $g_1, g_2$ where $g_i : \mathcal{Z}_i \to \mathbb{R}^D$ for $i = 1, 2$. Also, consider two matrix variables $Z_i := [z_{i1}, ..., z_{iM}]$ where their columns $z_{ij} \in \mathcal{Z}_i$ for $i = 1, 2$ and $j = 1, ..., M$. We denotes $\widehat{g_i}(Z_i) := [g_i(z_{i1}), ..., g_i(z_{iM})]$ for $i = 1, 2$. Set $\mathcal{G}_i := \{g | g : \mathcal{Z}_i \to \mathbb{R}^D\}$ for $i = 1, 2$. If $g_1(\mathcal{Z}_1) \subseteq g_2(\mathcal{Z}_2)$ for any $g_1 \in \mathcal{G}_1$ and $g_2 \in \mathcal{G}_2$, then $\min_{g_1 \in \mathcal{G}_1, Z_1 \in \mathcal{Z}_1^M} \mathcal{L}(\widehat{g_1}(Z_1)) \ge \min_{g_2 \in \mathcal{G}_2, Z_2 \in \mathcal{Z}_2^M} \mathcal{L}(\widehat{g_2}(Z_2))$.*

Please refer to Appendix A.1 for proof. Proposition 3.1 demonstrates that the optimal value of dataset distillation loss becomes lower as the feasible space of decoded synthetic instances becomes larger, when the number of decoded synthetic instances is fixed. The previous experimental finding supports Proposition 3.1 when $\mathcal{G}_1 = \mathcal{G}_2$: an increase in frequency dimension leads to improved dataset distillation performance (Shin et al., 2024).

Now, we investigate the feasible space of DDiF. We consider a neural field $F_\psi : \mathbb{R} \to \mathbb{R}$ with two hidden layers, each having a width of $d$ and employing a sine activation function. Extending the result of (Novello, 2022) using trigonometric identities, the output space of DDiF, which corresponds to the feasible space of the decoded synthetic instance, is represented as a sum of cosines:

$$F_\psi(x) = b^{(2)} + \sum_{k \in \mathbb{Z}^d} \sum_{i=1}^{d} A_{k,i} \cos\left(\omega_k x + \varphi'_{k,i}\right) \quad \text{where} \quad \varphi'_{k,i} = \varphi_{k,i} - \frac{\pi}{2} \tag{4}$$

where $\omega_k := \langle k, W^{(0)} \rangle$, $\varphi_{k,i} := \langle k, b^{(0)} \rangle + b_i^{(1)}$, $\alpha_k := \sum_{i=1}^{d} A_{k,i} \sin(\varphi_{k,i})$ and $\beta_k := \sum_{i=1}^{d} A_{k,i} \cos(\varphi_{k,i})$. Also, $A_{k,i} := W_i^{(2)} \prod_{j=1}^{d} J_{k_j} W_{ij}^{(1)}$ where $J_{k_j}$ denotes Bessel function of the first kind of order $k_j$. Please refer to Appendix A.2 for derivation. According to Eq. (4), it should be noted that the amplitudes $A_{k,i}$, frequencies $\omega_k$, phases $\varphi'_{k,i}$, and shift $b^{(0)}$ are tunable as the combination of neural network parameters $\psi = \{W^{(l)}, b^{(l)}\}_{l=0}^{L}$ in DDiF.

The feasible space of DDiF in Eq. (4) has a similar form of the feasible space of previous work, FreD (Shin et al., 2024), one of the static parameterization method in dataset distillation. FreD optimizes frequency coefficients, which are selected by the explained variance ratio. They basically use the inverse discrete cosine transform (IDCT) to decode synthetic instances from the frequency domain. Suppose that FreD utilizes IDCT with $N$ equidistant locations on $\mathbb{R}$. Also, when $\mathcal{U} \subset \mathcal{C}_N := \{0, ..., N-1\}$ is the index set of selected frequency dimension; the optimized frequency coefficients is denoted as $\Gamma := \{\gamma_u | u \in \mathcal{U}\}$. Then, the feasible space of decoded synthetic instance by FreD is expressed in the form of a function over $\mathcal{C}_N$:

$$g(x; \Gamma) = \sum_{u \in \mathcal{U}} \gamma_u \cos\left(\frac{\pi u}{N} x + \frac{\pi u}{2N}\right) \quad \text{where} \quad x \in \mathcal{C}_N \tag{5}$$

Now, we turn our attention to compare the feasible spaces of DDiF and FreD. Theorem 3.2 provides the relationship between the feasible spaces of DDiF and FreD when the same budget is utilized.

**Theorem 3.2.** *Consider the truncation of Eq. (4) over $\mathcal{K}_\zeta := \{k|\|k\|_\infty \leq \zeta\}$ i.e. $\tilde{F}_\psi(x) :=$ $b^{(2)} + \sum_{k \in \mathcal{K}_\zeta} \sum_{i=1}^d A_{k,i} \cos\left(\omega_k x + \varphi'_{k,i}\right)$. Suppose that FreD and DDiF utilize the same number of parameters, i.e., $|\mathcal{U}|$ and the number of parameters in $\psi$ are fixed at a given value $B$. If $B \geq 6$ and $\zeta \geq \frac{1}{2}\left(\exp\left(\frac{\log(2B+1)}{\lfloor\sqrt{3+B}-2\rfloor}\right) - 1\right)$, then $g(x; \Gamma) \subsetneq \tilde{F}_\psi(x)$ for any $x \in \mathcal{C}_N$.*

Please refer to Appendix A.3 for proof. Theorem 3.2 implies that the feasible space of decoded values in DDiF is larger than that in FreD when the parameters used for a single synthetic instance are fixed. Consequently, according to Proposition 3.1, when the number of decoded synthetic instances is equal, DDiF achieves a lower optimal value for the dataset distillation loss than FreD.

To support Theorem 3.2, we investigate the performance of the reconstruction task by varying the utilized budget. We speculate that the low reconstruction error is due to the large feasible space. For the reconstruction task, we utilize 10 randomly selected images per class on ImageNet-Subset (Cazenavette et al., 2022; 2023) with 128 resolution. For the dataset distillation task, we employ TM (Cazenavette et al., 2022) under one image per class, and the same dataset with the reconstruction task. As shown in Figure 3, DDiF achieves higher performance than FreD in both tasks, which indicates greater expressiveness under the same utilized budget.

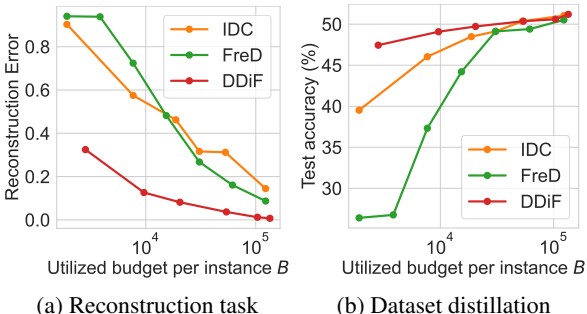

(a) Reconstruction task    (b) Dataset distillation

Figure 3: Performance curve on (a) reconstruction task and (b) dataset distillation under the same utilized budget. Pearson correlation coefficient of reconstruction error and test accuracy is $-0.89$.

Furthermore, we emphasize that DDiF maintains high performance even with a small budget, whereas FreD shows significant performance degradation as the allocated budget for each instance decreases. These experimental results provide further evidence that DDiF exhibits greater expressiveness. In addition, although we do not provide a theoretical comparison with IDC (Kim et al., 2022), Figure 3 presents empirical evidence demonstrating that DDiF exhibits greater expressiveness than IDC. We believe that the proposed theoretical analysis will serve as a starting point for future theoretical comparisons of parameterization methods in the dataset distillation domain.

## 4 EXPERIMENTS

This section presents various empirical results that validate the effectiveness of the proposed method, DDiF. We primarily focus on high-resolution image datasets, such as ImageNet-Subset (Cazenavette et al., 2022; 2023) with resolutions of 128 and 256, since high-dimensional dataset distillation remains a challenging task. In addition, we conduct several experiments to verify the applicability of DDiF on diverse modalities. We utilize miniUCF for video (Wang et al., 2024; Khurram, 2012), Mini Speech Commands for audio (Kim et al., 2022; Warden, 2018), and ModelNet (Wu et al., 2015), ShapeNet (Chang et al., 2015) for 3D. Specific details regarding the dataset, baselines, configuration of DDiF, and experimental settings are in Appendix B.

### 4.1 PERFORMANCE COMPARISON

**Main Results.** Table 1 shows the overall performance for ImageNet-Subset with resolution 128 and 256 under IPC=1. We utilize trajectory matching (TM) for 128 resolution and distribution matching (DM) for 256 resolution. DDiF achieves the best performances in all experimental settings. We highlight that the performance improvement from Vanilla to DDiF is significant, supporting the effectiveness of the proposed method (see the $\Delta$ row in Table 1). In addition, DDiF exhibits a substantial performance margin compared to the second-best performer: up to $4.1\%p$ at 128 resolution, and from $8.4\%p$ to $19.9\%p$ at 256 resolution. Moreover, when a larger budget is available, DDiF shows consistent improvement and highly competitive performance with baseline, as shown in Table 2. These results demonstrate that neural field-based parameterization significantly enhances the dataset distillation performance, particularly when the budget is very limited. Please refer to Appendix C.1 for the additional experimental results such as low-resolution datasets.

Table 1: Test accuracies (%) on ImageNet-Subset for parameterization methods under IPC=1. "IPC" denotes instances per class, which implies the budget constraint. **Bold** and Underline mean the best and second-best performance of each column, respectively. "†" indicates our implementation results. "−" indicates no reported results. "Δ" represents the performance improvement over Vanilla, which denotes input-sized parameterization. The full table with standard deviations is in Appendix C.7.

| | Resolution | $128 \times 128$ | | | | | | $256 \times 256$ | | | | | |
|---|---|---|---|---|---|---|---|---|---|---|---|---|---|
| | Subset | Nette | Woof | Fruit | Yellow | Meow | Squawk | Nette | Woof | Fruit | Yellow | Meow | Squawk |
| Input-sized | Vanilla | 51.4† | 29.7† | 28.8† | 47.5† | 33.3† | 41.0† | 32.1 | 20.0 | 19.5 | 33.4 | 21.2 | 27.6 |
| | FRePo | 48.1 | 29.7 | – | – | – | – | – | – | – | – | – | – |
| Static | IDC | 61.4 | 34.5 | 38.0 | 56.5 | 39.5 | 50.2 | 53.7† | 30.2† | 33.1† | 52.2† | 34.6† | 47.0† |
| | FreD | 66.8 | 38.3 | 43.7 | 63.2 | 43.2 | 57.0 | 54.2† | 31.2† | 32.5† | 49.1† | 34.0† | 43.1† |
| Parameterized | HaBa | 51.9 | 32.4 | 34.7 | 50.4 | 36.9 | 41.9 | – | – | – | – | – | – |
| | SPEED | 66.9 | 38.0 | 43.4 | 62.6 | 43.6 | 60.9 | 57.7 | – | – | – | – | – |
| | Vanilla+RTP | 69.6† | 38.8† | 45.2† | 66.4† | 46.5† | 63.2† | – | – | – | – | – | – |
| | LatentDD | – | – | – | – | – | – | 56.1 | 28.0 | 30.7 | – | 36.3 | 47.1 |
| | NSD | 68.6 | 35.2 | 39.8 | 61.0 | 45.2 | 52.9 | – | – | – | – | – | – |
| DGM Prior | GLaD | 38.7 | 23.4 | 23.1 | – | 26.0 | 35.8 | – | – | – | – | – | – |
| | H-GLaD | 45.4 | 28.3 | 25.6 | – | 29.6 | 39.7 | – | – | – | – | – | – |
| Function | DDiF | **72.0** | **42.9** | **48.2** | **69.0** | **47.4** | **67.0** | **67.8** | **39.6** | **43.2** | **63.1** | **44.8** | **67.0** |
| | Δ(%p) | +20.6 | +13.2 | +19.4 | +21.5 | +14.1 | +26.0 | +35.7 | +19.6 | +23.7 | +29.7 | +23.6 | +39.4 |
| Entire dataset $\mathcal{T}$ | | 87.4 | 67.0 | 63.9 | 84.4 | 66.7 | 87.5 | 92.5 | 80.1 | 70.2 | 90.5 | 72.2 | 93.2 |

Table 2: Test accuracies (%) on ImageNet-Subset ($128 \times 128$) under IPC=10.

| Method | Nette | Woof | Fruit | Yellow | Meow | Squawk |
|---|---|---|---|---|---|---|
| TM | 63.0 | 35.8 | 40.3 | 60.0 | 40.4 | 52.3 |
| FRePo | 66.5 | 42.2 | – | – | – | – |
| IDC | 70.8 | 39.8 | 46.4 | 68.7 | 47.9 | 65.4 |
| FreD | 72.0 | 41.3 | 47.0 | 69.2 | 48.6 | 67.3 |
| HaBa | 64.7 | 38.6 | 42.5 | 63.0 | 42.9 | 56.8 |
| SPEED | 72.9 | 44.1 | **50.0** | **70.5** | **52.0** | 71.8 |
| DDiF | **74.6** | **44.9** | 49.8 | **70.5** | 50.6 | **72.3** |
| Entire $\mathcal{T}$ | 87.4 | 67.0 | 63.9 | 84.4 | 66.7 | 87.5 |

Table 3: Average test accuracies (%) on ImageNet-Subset ($128 \times 128$) across AlexNet, VGG11, ResNet18, and ViT, under IPC=1.

| Method | Nette | Woof | Fruit | Yellow | Meow | Squawk |
|---|---|---|---|---|---|---|
| TM | 22.0 | 14.8 | 17.1 | 22.3 | 16.2 | 25.5 |
| IDC | 27.9 | 19.5 | 23.9 | 28.0 | 19.8 | 29.9 |
| FreD | 36.2 | 23.7 | 23.6 | 31.2 | 19.1 | 37.4 |
| GLaD | 30.4 | 17.1 | 21.1 | – | 19.6 | 28.2 |
| H-GLaD | 30.8 | 17.4 | 21.5 | – | 20.1 | 28.8 |
| LD3M | 32.0 | 19.9 | 21.4 | – | 22.1 | 30.4 |
| DDiF | **59.3** | **34.1** | **39.3** | **51.1** | **33.8** | **54.0** |

**Cross-architecture Generalization.** The network structure used for training with the distilled dataset may differ from the one used for distillation. Accordingly, parameterization methods should achieve consistent performance enhancements across various test network architectures. In this study, we employ AlexNet (Krizhevsky et al., 2012), VGG11 (Simonyan & Zisserman, 2014), ResNet18 (He et al., 2016), and ViT (Dosovitskiy, 2020) while ConvNetD5 is utilized in training. Table 3 presents that DDiF consistently outperforms. Notably, DDiF achieves a remarkable performance gap over the second-best performer from $10.4\%p$ to $23.1\%p$. These results indicate that DDiF effectively encodes important task-relevant information regardless of the training network.

**Universality to Distillation Loss.** Another important evaluation criterion for parameterization is whether it constantly improves the performance across various dataset distillation losses. We examine the performance of DDiF using gradient matching (DC) and distribution matching (DM) to evaluate its steady improvements. In Table 4, DDiF achieves the best performances in both distillation losses. These results confirm the universality of DDiF to distillation loss, representing its wide adaptiveness.

Table 4: Test accuracies (%) on ImageNet-Subset ($128 \times 128$) across DC and DM under IPC=1.

| $\mathcal{L}$ | Method | Nette | Woof | Fruit | Meow | Squawk |
|---|---|---|---|---|---|---|
| DC | Vanilla | 34.2 | 22.5 | 21.0 | 22.0 | 32.0 |
| | IDC | 45.4 | 25.5 | 26.8 | 25.3 | 34.6 |
| | FreD | 49.1 | 26.1 | 30.0 | 28.7 | 39.7 |
| | GLaD | 35.4 | 22.3 | 20.7 | 22.6 | 33.8 |
| | H-GLaD | 36.9 | 24.0 | 22.4 | 24.1 | 35.3 |
| | DDiF | **61.2** | **35.2** | **37.8** | **39.1** | **54.3** |
| DM | Vanilla | 30.4 | 20.7 | 20.4 | 20.1 | 26.6 |
| | IDC | 48.3 | 27.0 | 29.9 | 30.5 | 38.8 |
| | FreD | 56.2 | 31.0 | 33.4 | 33.3 | 42.7 |
| | GLaD | 32.2 | 21.2 | 21.8 | 22.3 | 27.6 |
| | H-GLaD | 34.8 | 23.9 | 24.4 | 24.2 | 29.5 |
| | DDiF | **69.2** | **42.0** | **45.3** | **45.8** | **64.6** |

**Various Modality.** Conventional parameterization methods have primarily been developed within the image domain, and their applicability to other modalities has not been sufficiently explored. Since the neural field provides a generalized representation of diverse grid-based data structures, DDiF can be naturally applied to various modalities beyond the image domain. We conduct experiments in the video, audio, and 3D voxel domains to evaluate its efficacy. First, Figure 4 shows the test performances on the video domain with regard to the required budget for running each method. SDD (Wang et al., 2024), which focuses on

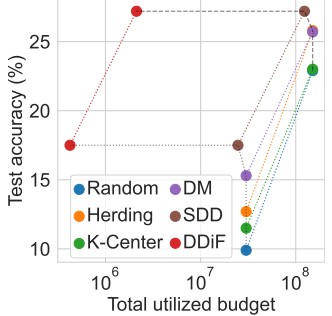

Figure 4: Test accuracies (%) on Video domain. Each black line denotes the same number of decoded instances per class, 1 and 5, respectively.

Table 5: Test accuracies (%) on Audio domain.

| Spec. / class | 10 | 20 |
|---|---|---|
| Random | 42.6 | 57.0 |
| Herding | 56.2 | 72.9 |
| DSA | 65.0 | 74.0 |
| DM | 69.1 | 77.2 |
| IDC-I | 73.3 | 83.0 |
| IDC-I + HaBa | 74.5 | 84.3 |
| IDC | 82.9 | 86.6 |
| DDiF | **90.5** | **92.7** |
| Entire $\mathcal{T}$ | | 93.4 |

Table 6: Test accuracies (%) on 3D voxel domain under IPC=1.

| $\mathcal{L}$ | Method | ModelNet | ShapeNet |
|---|---|---|---|
| — | Random | 60.9 | 68.5 |
| DC | Vanilla | 56.3 | 48.6 |
| | IDC | 78.7 | 79.9 |
| | FreD | 85.6 | 88.2 |
| | DDiF | **87.1** | **89.6** |
| DM | Vanilla | 75.3 | 80.4 |
| | IDC | 85.6 | 85.3 |
| | FreD | 87.3 | 90.6 |
| | DDiF | **88.4** | **93.1** |
| | Entire $\mathcal{T}$ | 91.6 | 98.3 |

the video domain, achieves higher performance than coreset selections and DM; however, it still requires a large storage budget. On the other hand, DDiF achieves competitive performance even with a budget equivalent to only 1.7% of SDD. In addition, Tables 5 and 6 show the test performances of the audio and 3D voxel domains, respectively. Both tables demonstrate that DDiF achieves the highest performance. These results confirm the efficacy of DDiF across various modality datasets.

**Cross-resolution Generalization.** As mentioned in Section 3.1, previous studies can only perform postprocessing to resize the optimized synthetic datasets, resulting in information distortion. On the contrary, DDiF can easily decode data of various sizes by adjusting the coordinate set, due to the continuous nature of the neural field. We introduce a novel experiment in the dataset distillation community that assesses generalization performance when the evaluation data size differs from that used for distillation. We define this experiment as a *cross-resolution generalization*. We apply the interpolation techniques, such as nearest, bilinear, and bicubic, for the previous parameterization methods.

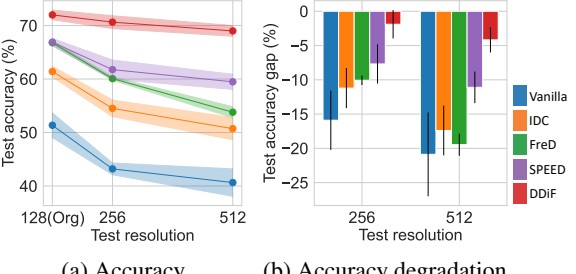

(a) Accuracy  (b) Accuracy degradation

Figure 5: (a) Test accuracies (%) with different image resolutions. The original resolution is $128 \times 128$. (b) Test accuracy gap (%) from original resolution. We use bilinear interpolation for previous studies.

Figure 5a shows test accuracies on each test resolution when utilizing the corresponding network architecture, ConvNetD6 for 256 and ConvNetD7 for 512. DDiF shows the best performance over all test resolutions. Figure 5b shows the percentage of decrease in test accuracy with the resolution change i.e. $(ACC_{org} - ACC_{test})/ACC_{org}$. DDiF shows the least decrease with regard to resolution difference, being evidently robust to resolution change. Its robustness opens a new adaptability of dataset distillation methods to more wide-range situations. Please refer to Appendix C.4 for the experimental results on different resizing techniques and the same test architecture.

## 4.2 ADDITIONAL ANALYSIS AND ABLATION STUDY

**Fixed Number of Decoded Instances.** To further investigate the coding efficiency and expressiveness, we hold the number of decoded synthetic instances constant while varying the budget allocated to each instance, and then evaluate performance. Under the IPC=1 setting, FreD, SPEED, and DDiF decode 8, 15, and 51 synthetic instances per class at a resolution of 128, respectively. We consider two settings in which the number of decoded synthetic instances is determined by either baselines or DDiF. Table 7 shows that DDiF achieves strong performance while using less budget than the baselines given the same number of decoded instances. It is also worth noting that, when using the same budget, DDiF decodes more synthetic instances and achieves significantly higher performance. These results support the high coding efficiency and expressiveness of DDiF, demonstrating that its superiority arises from improvements in both quality and diversity.

Table 7: Test accuracies (%) on ImageNet-Subset ($128 \times 128$) under several DIPCs. "DIPC" indicates the number of decoded synthetic instances per class. Note that the total number of budget parameters for IPC=1 is 491.52k.

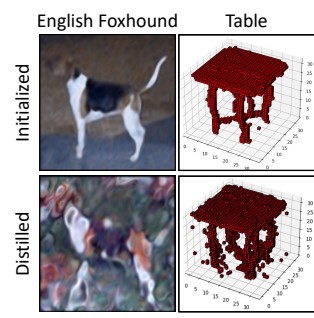

English Foxhound     Table

Initialized

Distilled

| DIPC | Method | Utilized budget | Nette | Woof | Fruit | Yellow | Meow | Squawk |
|---|---|---|---|---|---|---|---|---|
| 1 | Vanilla | 491.52k | **51.4** | **29.7** | **28.8** | **47.5** | **33.3** | **41.0** |
|   | DDiF | 9.63k | 49.1 | 29.4 | 27.3 | 44.8 | 31.9 | 39.0 |
| 8 | FreD | 491.52k | 66.8 | **38.3** | **43.7** | **63.2** | 43.2 | 57.0 |
|   | DDiF | 77.04k | **67.1** | 37.8 | 43.6 | 61.5 | **44.5** | **58.3** |
| 15 | SPEED | 491.52k | 66.9 | 38.0 | 43.4 | 62.6 | 43.6 | 60.9 |
|   | DDiF | 144.45k | **68.3** | **39.7** | **45.9** | **65.8** | **45.4** | **61.1** |
| 51 | Vanilla | 25067.52k | **73.0** | 42.8 | 48.2 | **69.1** | 47.2 | **69.0** |
|   | DDiF | 491.52k | 72.0 | **42.9** | 48.2 | 69.0 | **47.4** | 67.0 |

Figure 6: Visualization of the decoded synthetic instances from DDiF on image (left) and 3D voxel (right).

**Qualitative Analysis.** Figure 6 visualizes the decoded synthetic instances by DDiF on various modalities. As shown in the first row in Figure 6, DDiF effectively encodes high-dimensional data even with a very small budget, regardless of the modality. For instance, each synthetic neural field utilizes a budget that is $1.96\%$ of the original data size for images and $2.87\%$ for 3D voxel. After the distillation stage, each decoded synthetic instance involves class-discriminative features, even with significant budget reductions (see the second row). Notably, since the synthetic neural field is a continuous function, the quantity changes smoothly as the position changes. Please refer to Appendix C.7 for additional visualizations of images, videos, and 3D voxels.

**Ablation Studies.** DDiF employs a neural field, which is a neural network, to store the distilled information. We conduct several ablation studies to investigate the effect of different components of the neural network. Figures 7a and 7b demonstrate that DDiF consistently enhances performance, regardless of changes in the number of layers or network width. As the number of layers and the network width increase, the number of decoded synthetic instances decreases because the number

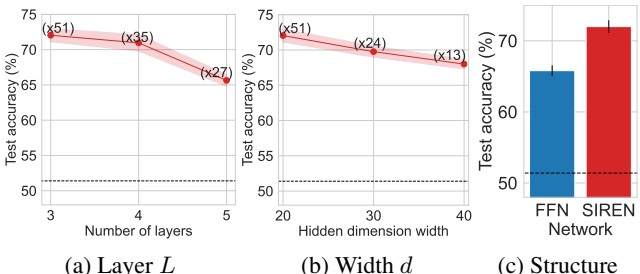

(a) Layer $L$      (b) Width $d$      (c) Structure

Figure 7: Ablation studies on (a) layer $L$, (b) width $d$, and (c) structure of neural field. The bottom black dashed line indicates the performance of Vanilla (TM).

of parameters in each synthetic neural field increases. By comparing the layer and width, the width has fewer decoded synthetic instances, but it has a more gradual performance change than the layer. To explain the rationale, as shown in Eq. (4), width $d$ directly affects the number of basis functions $k$, while layer $L$ only affects other factors. Thus, increasing $d$ leads to a modest change, as the increase in expressiveness offsets the reduction in quantity. In contrast, increasing $L$ results in a relatively larger change, as the expressiveness remains similar while the quantity decreases. Even when the structure of the neural field is changed as FFN (Tancik et al., 2020), DDiF consistently shows performance improvement (see Figure 7c).

## 5 CONCLUSION

This paper introduces DDiF, a novel parameterization framework for dataset distillation that encodes information from the large-scale dataset into synthetic neural fields under a constrained storage budget. By utilizing neural fields, DDiF efficiently encapsulates distilled information from high-dimensional grid-based data and easily decodes data of various sizes. We theoretically analyze the expressiveness of DDiF by investigating the feasible space of decoded synthetic instances and demonstrate that DDiF possesses greater expressiveness than the previous method. Through extensive experiments, we demonstrate that DDiF consistently exhibits performance improvements, high generalization and robustness, and broad adaptability across diverse modality datasets.

ACKNOWLEDGMENTS

This work was supported by Samsung Electronics Co., Ltd (No. IO231011-07374-01). This work was also supported by the IITP(Institute of Information & Coummunications Technology Planning & Evaluation)-ITRC(Information Technology Research Center) grant funded by the Korea government(Ministry of Science and ICT)(IITP-2025-RS-2024-00437268). We thank Wonjun Choi (wonjun4.choi@samsung.com) and Seongeun Kim (se91.kim@samsung.com) from the Core Algorithm Lab, AI Center, for their warm and keen discussions on our research.

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

## A    PROOFS AND MORE ANALYSIS

### A.1    PROOF OF PROPOSITION 3.1

**Proposition 3.1.** *Consider two functions $g_1, g_2$ where $g_i : \mathcal{Z}_i \to \mathbb{R}^D$ for $i = 1, 2$. Also, consider two matrix variables $Z_i := \begin{bmatrix} z_{i1}, ..., z_{iM} \end{bmatrix}$ where their columns $z_{ij} \in \mathcal{Z}_i$ for $i = 1, 2$ and $j = 1, ..., M$. We denotes $\widehat{g_i}(Z_i) := \begin{bmatrix} g_i(z_{i1}), ..., g_i(z_{iM}) \end{bmatrix}$ for $i = 1, 2$. Set $\mathcal{G}_i := \{g | g : \mathcal{Z}_i \to \mathbb{R}^D\}$ for $i = 1, 2$. If $g_1(\mathcal{Z}_1) \subseteq g_2(\mathcal{Z}_2)$ for any $g_1 \in \mathcal{G}_1$ and $g_2 \in \mathcal{G}_2$, then $\min_{g_1 \in \mathcal{G}_1, Z_1 \in \mathcal{Z}_1^M} \mathcal{L}(\widehat{g_1}(Z_1)) \geq \min_{g_2 \in \mathcal{G}_2, Z_2 \in \mathcal{Z}_2^M} \mathcal{L}(\widehat{g_2}(Z_2)).$*

*Proof.* For $i = 1, 2$, let $g_i^*, Z_i^* = \arg\min_{g_i \in \mathcal{G}, Z_i \in \mathcal{Z}_i^M} \mathcal{L}(\widehat{g_i}(Z_i))$. Note that $g_1^*(z_{1j}^*) \in g_1^*(\mathcal{Z}_1) \subseteq g_2(\mathcal{Z}_2)$ for $j = 1, ..., M$ and any $g_2 \in \mathcal{G}_2$. It implies that there exists some $g_2 \in \mathcal{G}_2, Z_2 \in \mathcal{Z}_2^M$ such that $\widehat{g_2}(Z_2) = \widehat{g_1^*}(Z_1^*)$. By the definition of $g_2^*, Z_2^*$, for any $g_2 \in \mathcal{G}_2, Z_2 \in \mathcal{Z}_2^M, \mathcal{L}(\widehat{g_2^*}(Z_2^*)) \leq \mathcal{L}(\widehat{g_2}(Z_2))$. Therefore, $\mathcal{L}(\widehat{g_2^*}(Z_2^*)) \leq \mathcal{L}(\widehat{g_1^*}(Z_1^*))$. In conclusion, $\min_{g_1 \in \mathcal{G}_1, Z_1 \in \mathcal{Z}_1^M} \mathcal{L}(\widehat{g_1}(Z_1)) \geq \min_{g_2 \in \mathcal{G}_2, Z_2 \in \mathcal{Z}_2^M} \mathcal{L}(\widehat{g_2}(Z_2)).$ □

The following Corollary A.1 states that the relationship between the feasible space of synthetic instances from input-sized parameterization and the optimal value of the dataset distillation objective, when the number of synthetic instances is the same.

**Corollary A.1.** *Let two matrix variables $X_1 := \begin{bmatrix} x_{11}, ..., x_{1M} \end{bmatrix}$ and $X_2 := \begin{bmatrix} x_{21}, ..., x_{2M} \end{bmatrix}$ consisting of columns $x_{ij} \in \mathcal{X}_i \subseteq \mathbb{R}^D$ for $i = 1, 2$ and $j = 1, ..., M$. If $\mathcal{X}_1 \subseteq \mathcal{X}_2$, then $\min_{X_1 \in \mathcal{X}_1^M} \mathcal{L}(X_1) \geq \min_{X_2 \in \mathcal{X}_2^M} \mathcal{L}(X_2).$*

To support Corollary A.1, we investigate the performance changes of input-sized parameterization while imposing constraints on the feasible space of synthetic instances under the fixed number of synthetic instances. We consider two types of constraints: 1) dimension masking and 2) value clipping. As shown in Figure 8, when the feasible space becomes smaller (i.e., as the restrictions are enforced more strongly), the dataset distillation performances decrease. These results serve as direct evidence of Corollary A.1.

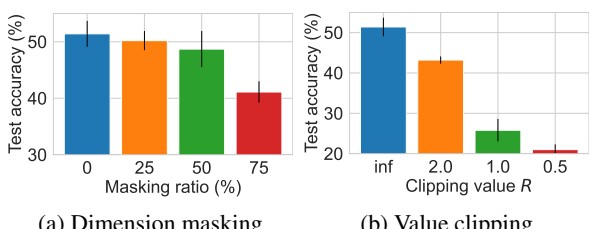

(a) Dimension masking          (b) Value clipping

Figure 8: Test accuracies (%) of input-sized parameterization. We utilize TM on ImageNette ($128 \times 128$) under IPC=1.

### A.2    DERIVATION OF EQ. (4)

To derive the feasible space of DDiF, we begin with the theorem from Novello (2022), which we state as follows:

**Theorem A.2.** *(Novello, 2022) Consider a neural network $F_\psi : \mathbb{R} \to \mathbb{R}$ with two hidden layers and width d. If $F_\psi$ utilizes a sine activation function, then $F_\psi$ represents a function which is the sum of sines and cosines:*

$$F_\psi(x) = b^{(2)} + \sum_{k \in \mathbb{Z}^d} \alpha_k \cos(\omega_k x) + \beta_k \sin(\omega_k x) \tag{6}$$

*where $\omega_k := \langle k, W^{(0)} \rangle$, $\varphi_{k,i} := \langle k, b^{(0)} \rangle + b_i^{(1)}$, $\alpha_k := \sum_{i=1}^d A_{k,i} \sin(\varphi_{k,i})$ and $\beta_k := \sum_{i=1}^d A_{k,i} \cos(\varphi_{k,i})$. Also, $A_{k,i} := W_i^{(2)} \lambda_k(W_i^{(1)})$ and $\lambda_k(W_i^{(1)}) := \prod_{j=1}^d J_{k_j} W_{ij}^{(1)}$ where $J_{k_j}$ denotes Bessel function of the first kind of order $k_j$.*

Then, by using trigonometric identity, Eq. (6) is represented by the sum of cosines as follows:

$$F_\psi(x) = b^{(2)} + \sum_{k \in \mathbb{Z}^d} \alpha_k \cos(\omega_k x) + \beta_k \sin(\omega_k x)$$

$$= b^{(2)} + \sum_{k \in \mathbb{Z}^d} \left\{ \sum_{i=1}^{d} A_{k,i} \sin(\varphi_{k,i}) \cos(\omega_k x) + \sum_{i=1}^{d} A_{k,i} \cos(\varphi_{k,i}) \sin(\omega_k x) \right\}$$

$$= b^{(2)} + \sum_{k \in \mathbb{Z}^d} \sum_{i=1}^{d} A_{k,i} \sin\left(\omega_k x + \varphi_{k,i}\right)$$

$$= b^{(2)} + \sum_{k \in \mathbb{Z}^d} \sum_{i=1}^{d} A_{k,i} \cos\left(\omega_k x + \varphi'_{k,i}\right) \quad \text{where} \quad \varphi'_{k,i} = \varphi_{k,i} - \frac{\pi}{2}$$

## A.3 PROOF OF THEOREM 3.2

**Lemma A.3.** *For $d \in \mathbb{Z}^+$, there exists $x \in \mathbb{R}$ such that $\prod_{j=1}^{d} J_{k_j}(x) \neq 0$ where $k_j \in \mathbb{Z}, j = 1, 2, ..., d$ and $J_{k_j}$ denotes the Bessel function of the first kind of order $k_j$.*

*Proof.* For $j = 1, 2, ..., d$, denote the zero set of $J_{k_j}$ on the real line by $Z(J_{k_j}) := \{x \in \mathbb{R} | J_{k_j}(x) = 0\}$. Note that $Z(J_{k_j})$ is discrete set, when $k_j$ is integer, so it is countable set (Watson, 1922; Abramowitz & Stegun, 1948; Kerimov, 2014). Then, $\bigcup_{j=1}^{d} Z(J_{k_j})$ is also countable set. Since $\mathbb{R}$ is uncountable set, there exists some $x \in \mathbb{R}$ such that $x \notin \bigcup_{j=1}^{d} Z(J_{k_j})$. Equivalently, $J_{k_j}(x) \neq 0$ for every $j = 1, 2, ..., d$. Hence, for that choice of $x$, $\prod_{j=1}^{d} J_{k_j}(x) \neq 0$ ☐

**Theorem 3.2.** *Consider the truncation of Eq. (4) over $\mathcal{K}_\zeta := \{k | \|k\|_\infty \leq \zeta\}$ i.e. $\tilde{F}_\psi(x) := b^{(2)} + \sum_{k \in \mathcal{K}_\zeta} \sum_{i=1}^{d} A_{k,i} \cos\left(\omega_k x + \varphi'_{k,i}\right)$. Suppose that FreD and DDiF utilize the same number of parameters, i.e., $|\mathcal{U}|$ and the number of parameters in $\psi$ are fixed at a given value $B$. If $B \geq 6$ and $\zeta \geq \frac{1}{2}\left(\exp\left(\frac{\log(2B+1)}{\lfloor\sqrt{3+B}-2\rfloor}\right) - 1\right)$, then $g(x; \Gamma) \subsetneq \tilde{F}_\psi(x)$ for any $x \in \mathcal{C}_N$.*

*Proof.* Since we consider $\tilde{F}_\psi : \mathbb{R} \to \mathbb{R}$, the number of parameters of DDiF is $d^2 + 4d + 1$. For a given budget constraint $B$, $d = \lfloor\sqrt{3+B}-2\rfloor$ is the maximum width of $\tilde{F}_\psi$ which satisfy $d^2 + 4d + 1 \leq B$. Since $B \geq 6$, DDiF is able to construct a valid neural network, i.e., $d \geq 1$. To prove $g(x; \Gamma) \subseteq \tilde{F}_\psi(x)$, it is sufficient to show that there exist neural network parameters $\psi = \{W^{(j)}, b^{(j)}\}_{j=0}^{2}$ which satisfy $g(x; \Gamma) = \tilde{F}_\psi(x)$ for any $\Gamma$ and $x \in \mathcal{C}_N = \{0, ..., N-1\}$. First, we decompose Eq. (4) into a sum over $\mathcal{K} \subseteq \mathbb{Z}^d$ and the other terms:

$$\tilde{F}_\psi(x) = b^{(2)} + \sum_{k \in \mathcal{K}_\zeta} \sum_{i=1}^{d} A_{k,i} \cos\left(\omega_k x + \varphi_{k,i}\right)$$

$$= b^{(2)} + \sum_{k \in \mathcal{K}} \sum_{i=1}^{d} A_{k,i} \cos\left(\omega_k x + \varphi_{k,i}\right) + \sum_{k \in \mathcal{K}_\zeta \backslash \mathcal{K}} \sum_{i=1}^{d} A_{k,i} \cos\left(\omega_k x + \varphi_{k,i}\right)$$

Let $\mathcal{K} = \left\{k \in \mathcal{K}_\zeta | \sum_{i=1}^{d} k_i = u, u \in \mathcal{U}\right\}$ and $1_N = [1, 1, ..., 1]^T \in \mathbb{R}^N$ is the one-vector of size $N$. Set $W^{(0)} = \frac{\pi}{N} 1_N$, $b^{(0)} = \frac{\pi}{2N} 1_N$, $b^{(1)} = \frac{\pi}{2} 1_N$, and $b^{(2)} = -\sum_{k \in \mathbb{Z}^d \backslash \mathcal{K}} \sum_{i=1}^{d} A_{k,i} \cos\left(\omega_k x + \varphi'_{k,i}\right)$. Note that the absolute value of the Bessel function of the first kind has a finite upper bound $|J_p(r)| < \frac{(\frac{|r|}{2})^p}{p!}$ (Paris, 1984) for any $p, r > 0$. Also, by Lemma A.3, $W^{(1)}$ and $W^{(2)}$ can be configured to satisfy $\sum_{i=1}^{d} A_{k,i} = \sum_{i=1}^{d} W_i^{(2)} \prod_{j=1}^{d} J_{k_j}(W_{ij}^{(1)}) = \gamma_u$ for a fixed $u$. Under these parameters, $\tilde{F}_\psi(x)$ is same as $g(x; \Gamma)$:

$$\tilde{F}_\psi(x) = b^{(2)} + \sum_{k \in \mathcal{K}} \sum_{i=1}^{d} A_{k,i} \cos\left(\omega_k x + \varphi'_{k,i}\right) + \sum_{k \in \mathcal{K}_\zeta \backslash \mathcal{K}} \sum_{i=1}^{d} A_{k,i} \cos\left(\omega_k x + \varphi'_{k,i}\right)$$

$$= \sum_{u \in \mathcal{U}} \gamma_u \cos\left(\frac{\pi u}{N} x + \frac{\pi u}{2N}\right) = g(x; \Gamma)$$

Next, we prove that there is no $\Gamma$ such that $g(x; \Gamma) = \tilde{F}_\psi(x)$ for some $\psi$ and $x \in \mathcal{C}_N$. Note that $\tilde{F}_\psi$ can represent up to $\frac{(2\zeta+1)^d - 1}{2}$ (Novello, 2022). By recalling that $d = \lfloor \sqrt{3+B} - 2 \rfloor$, we can show the following equivalence from the inequality assumption of $\zeta$:

$$\zeta \geq \frac{1}{2}\Big( \exp\big(\frac{\log(2B+1)}{\lfloor \sqrt{3+B} - 2 \rfloor}\big) - 1\Big) \quad \Leftrightarrow \quad \log(2\zeta + 1) \geq \frac{\log(2B+1)}{\lfloor \sqrt{3+B} - 2 \rfloor}$$

$$\Leftrightarrow \quad \frac{(2\zeta+1)^{\lfloor \sqrt{3+B} - 2 \rfloor} - 1}{2} \geq B$$

It implies that DDiF can cover more frequency than FreD. Consequently, it is possible to choose $k$ and $W^{(0)}$ which satisfy $\omega_k = \langle k, W^{(0)} \rangle \neq \frac{\pi u}{N}$ for any $u \in \mathcal{U}$. Due to the orthogonality of cosine functions having different frequencies, there is no $\Gamma$ that represents $\cos\big(\omega_k x + \varphi'_{k,i}\big)$ terms. It implies $g(x; \Gamma)$ cannot express $F_\psi(x)$ with some $\psi$ parameters. □

# B EXPERIMENTAL DETAILS

## B.1 DATASETS

**Image Domain.** We evaluate DDiF on various benchmark image datasets. 1) ImageNet-Subset (Howard, 2019; Cazenavette et al., 2022) is a dataset consisting of a subset of similar characteristics in the ImageNet. In the experiment, we consider various types of subsets by following Cazenavette et al. (2022): ImageNette (various objects), ImageWoof (dog breeds), ImageFruit (fruit category), ImageMeow (cats), ImageSquawk (birds), ImageYellow (yellowish objects). Each subset has 10 classes and more than 10,000 instances. We utilize two types of resolution: $128 \times 128$ and $256 \times 256$. 2) CIFAR-10 (Krizhevsky et al., 2009) consists of 60,000 RGB images in 10 classes. Each image has a $32 \times 32$ size. Each class contains 5,000 images for training and 1,000 images for testing. 3) CIFAR-100 (Krizhevsky et al., 2009) consists of 60,000 $32 \times 32$ RGB images of 100 categories. Each class is split into 500 for training and 100 for testing.

**Video Domain.** We utilize MiniUCF (Wang et al., 2024), a subset of UCF101 (Soomro, 2012) which includes 50 classes. The videos are sampled to 16 frames, and the frames are cropped and resized to $112 \times 112$. Each data has $16 \times 3 \times 112 \times 112$ size.

**Audio Domain.** We utilize Mini Speech Commands (Kim et al., 2022), a subset of the original Speech Commands dataset (Warden, 2018). We follow the data processing of Kim et al. (2022). The dataset consists of 8 classes, and each class has 875/125 data for training/testing, respectively. Each data is $64 \times 64$ log-scale magnitude spectrograms by short-time Fourier transform (STFT).

**3D Domain.** We utilize a core version of ModelNet-10 (Wu et al., 2015) and ShapeNet (Chang et al., 2015), which are widely used in 3D. They include 10 classes and 16 classes, respectively. Each 3D point cloud data is converted into $32 \times 32 \times 32$ voxel.

## B.2 NETWORK ARCHITECTURES.

**ConvNet.** By following previous studies, we leverage the ConvNetD$n$ as a default network architecture for both distillation and evaluation of synthetic datasets. The ConvNetD$n$ is a convolutional neural network with $n$ duplicate blocks. Each $n$ blocks consist of a convolution layer with $3 \times 3$-shape 128 filters, an instance normalization layer, ReLU, and an average pooling with $2 \times 2$ kernel size with stride 2. Lastly, it contains a linear classifier, which outputs the logits. Depending on the resolution of the real dataset, we utilize different depth $n$. Specifically, ConvNetD3 for $32 \times 32$ CIFAR-10 and CIFAR-100, ConvNetD4 for $64 \times 64$ Audio spectrograms, ConvNetD5 for $128 \times 128$ ImageNet-Subset, ConvNetD6 for $256 \times 256$ ImageNet-Subset, and ConvNetD7 for $512 \times 512$ ImageNet-Subset.

**AlexNet.** AlexNet is a basic convolutional neural network architecture suggested in (Krizhevsky et al., 2012). It consists of 5 convolution layers, 3 max-pooling layers, 2 Normalized layers, 2 fully connected layers, and 1 SoftMax layer. In each convolution layer, the ReLU activation function is utilized. We adopt this network to evaluate cross-architecture performance of DDiF.

**VGG11.** VGG11 (Simonyan & Zisserman, 2014) is also applied for evaluation, which attributes to 11 weighted layers. It consists of 8 convolution layers and 3 fully connected layers. Its design is straightforward yet powerful, providing a balance between depth and computational efficiency. The number of trainable parameters is around 132M, making it larger than earlier models but still suitable for medium-scale tasks. We adopt this network to evaluate the cross-architecture performance of DDiF.

**ResNet18.** ResNet18 (He et al., 2016) introduces residual connections, which help mitigate the vanishing gradient problem in deep networks by allowing gradients to bypass certain layers. It consists of 18 layers with 4 residual blocks, each composed of two convolutional layers followed by activation and normalization with around 11M trainable parameters. We utilize ResNet18 as one of the architecture for evaluating synthetic datasets.

**ViT.** Vision Transformer (Dosovitskiy, 2020) utilizes the transformer architecture, initially designed for sequence modeling tasks in NLP. For image classification, it divides images into non-overlapping patches and processes them as a sequence using self-attention mechanisms. ViT has around 10M trainable parameters in its base form and offers a competitive alternative to CNNs, demonstrating the effectiveness of transformers in vision tasks. We selected ViT as the final network to evaluate synthetic image datasets.

**Conv3DNet.** For the 3D voxel domain, we utilize Conv3DNet (Shin et al., 2024), a 3D version of ConvNet. Conv3DNet consists of three repeated blocks, each containing a $3 \times 3 \times 3$ convolutional layer with 64 filters, 3D instance normalization, ReLU activation, and 3D average pooling with a $2 \times 2 \times 2$ filter, and a stride of 2. Lastly, it contains a linear classifier.

### B.3 BASELINES.

Since our main focus lies on the parameterization of dataset distillation, we compare DDiF with 1) static decoding, which are IDC (Kim et al., 2022) and FreD (Shin et al., 2024); 2) parameterized decoding, which is RTP (Deng & Russakovsky, 2022), HaBa (Liu et al., 2022), SPEED (Wei et al., 2024), LatentDD (Duan et al., 2023), and NSD (Yang et al., 2024); and 3) deep generative prior, which include GLaD (Cazenavette et al., 2023), H-GLaD (Zhong et al., 2024), and LD3M (Moser et al., 2024). We also demonstrate the performance improvement of DDiF compared to input-sized parameterization, denoted as Vanilla.

### B.4 IMPLEMENTATION SETTINGS.

Although any loss can be adapted to DDiF, we utilize TM (Cazenavette et al., 2022) for $\mathcal{L}$ as a default unless specified. Following previous studies, we use DSA (Zhao & Bilen, 2021), which consists of color jittering, cropping, cutout, flipping, scaling, and rotation. We adopt ZCA whitening on CIFAR-10 (IPC=1, 10) and CIFAR-100 (IPC=1) with the Kornia (Riba et al., 2020) implementation. We adopt SIREN (Sitzmann et al., 2020) for synthetic field $F_\psi$ as a default. SIREN is a multilayer perceptron with a sinusoidal activation function, and it is widely used in the neural field area due to its simple structure. We use the same width across all layers in a synthetic neural field i.e. $d_l = d$ for all $l$. We utilize normalized coordinates defined on $[-1, 1]^n$ for $n$-dimension data to enhance stability (Sitzmann et al., 2020), rather than using integer coordinates, which have a wide range. For cross-resolution experiments, we utilize the coordinate set $C$, which consists of evenly spaced points within the interval $[-1, 1]$ according to the target resolution. We provide the detailed configuration of the synthetic neural field, the resulting size of each neural field, the number of synthetic instances per class, and the total number of neural fields in Table 8. We use Adam optimizer (Kingma & Ba, 2017) for all experiments. We fix the iteration number and learning rate for warm-up initialization of synthetic neural field as 5,000 and 0.0005. Without any description to distillation loss, we generally use matching training trajectory (TM) objective for dataset distillation loss $\mathcal{L}$. Following the previous studies, we utilize two types of default TM hyperparameters same as SPEED (Wei et al., 2024) and FreD (Shin et al., 2024). We run 15,000 iterations for TM and 20,000 iterations for DM. We use a mixture of RTX 3090, L40S, and Tesla A100 to run our experiments. We follow the conventional evaluation procedure of the previous studies: train 5 randomly initialized networks with an optimized synthetic dataset and evaluate the classification performance. We provide the detailed hyperparameters in Table 9.

Table 8: Configuration of the synthetic neural field. In the case of Video, there is no increment of decoded instances because we experimented with the fixed number of decoded instances.

| Modality | Dataset | IPC | $n$ | $L$ | $d$ | $m$ | size($\psi$) | Increment of decoded instances |
|---|---|---|---|---|---|---|---|---|
| Image | CIFAR10 | 1 | 2 | 2 | 6 | 3 | 81 | $\times 37$ |
| | | 10 | 2 | 2 | 6 | 3 | 81 | $\times 37.9$ |
| | | 50 | 2 | 2 | 20 | 3 | 543 | $\times 5.64$ |
| | CIFAR100 | 1 | 2 | 2 | 10 | 3 | 173 | $\times 17$ |
| | | 10 | 2 | 2 | 15 | 3 | 333 | $\times 9.2$ |
| | | 50 | 2 | 2 | 30 | 3 | 1113 | $\times 2.76$ |
| | ImageNet-Subset ($128 \times 128$) | 1 | 2 | 3 | 20 | 3 | 963 | $\times 51$ |
| | | 10 | 2 | 3 | 20 | 3 | 963 | $\times 51$ |
| | | 50 | 2 | 3 | 40 | 3 | 3523 | $\times 13.94$ |
| | ImageNet-Subset ($256 \times 256$) | 1 | 2 | 3 | 40 | 3 | 3523 | $\times 55$ |
| Video | MiniUCF | 1 | 3 | 6 | 40 | 3 | 8483 | – |
| | | 5 | 3 | 6 | 40 | 3 | 8483 | – |
| Audio | Mini Speech Commands | 10 | 2 | 3 | 10 | 1 | 261 | $\times 15.6$ |
| | | 20 | 2 | 3 | 10 | 1 | 261 | $\times 15.6$ |
| 3D | ModelNet | 1 | 3 | 3 | 20 | 1 | 941 | $\times 34$ |
| | ShapeNet | 1 | 3 | 3 | 20 | 1 | 941 | $\times 34$ |

Table 9: Configuration of hyperparameters for optimization.

(a) Gradient matching (DC)

| Dataset | IPC | Synthetic batch size | Learning rate (Neural field) |
|---|---|---|---|
| ImageNet-Subset ($128 \times 128$) | 1 | - | $5 \times 10^{-5}$ |
| Mini Speech Commands | 10 | 64 | $10^{-5}$ |
| | 20 | 64 | $10^{-4}$ |
| ModelNet | 1 | - | $10^{-4}$ |
| ShapeNet | 1 | - | $10^{-4}$ |

(b) Distribution matching (DM)

| Dataset | IPC | Synthetic batch size | Learning rate (Neural field) |
|---|---|---|---|
| ImageNet-Subset ($128 \times 128$) | 1 | - | $5 \times 10^{-5}$ |
| ImageNet-Subset ($256 \times 256$) | 1 | - | $10^{-5}$ |
| MiniUCF | 1 | - | $10^{-4}$ |
| | 5 | - | $10^{-4}$ |
| ModelNet | 1 | - | $10^{-4}$ |
| ShapeNet | 1 | - | $10^{-4}$ |

(c) Trajectory matching (TM)

| Dataset | IPC | Synthetic steps | Expert epochs | Max start epoch | Synthetic batch size | Learning rate (Neural field) | Learning rate (Step size) | Learning rate (Teacher) |
|---|---|---|---|---|---|---|---|---|
| CIFAR-10 | 1 | 60 | 2 | 10 | 74 | $10^{-3}$ | $10^{-5}$ | $10^{-2}$ |
| | 10 | 60 | 2 | 10 | 256 | $10^{-3}$ | $10^{-5}$ | $10^{-2}$ |
| | 50 | 60 | 2 | 40 | 235 | $10^{-4}$ | $10^{-5}$ | $10^{-2}$ |
| CIFAR-100 | 1 | 60 | 2 | 40 | 170 | $10^{-3}$ | $10^{-5}$ | $10^{-2}$ |
| | 10 | 60 | 2 | 40 | 230 | $10^{-3}$ | $10^{-5}$ | $10^{-2}$ |
| | 50 | 60 | 2 | 40 | 276 | $10^{-3}$ | $10^{-5}$ | $10^{-2}$ |
| ImageNet-Subset ($128 \times 128$) | 1 | 20 | 2 | 10 | 102 | $10^{-4}$ | $10^{-6}$ | $10^{-2}$ |
| | 10 | 40 | 2 | 20 | 30 | $10^{-4}$ | $10^{-5}$ | $10^{-2}$ |
| | 50 | 40 | 2 | 20 | 30 | $10^{-4}$ | $10^{-5}$ | $10^{-2}$ |

## B.5 Algorithm of DDiF

The main difference between DDiF and conventional parameterization methods is the decoding process for synthetic instances. Basically, the neural field takes coordinates as input and output quantities. To generate a single synthetic instance, each coordinate $c \in \mathcal{C}$ is input into the synthetic neural field $F_\psi$, and then the resulting value is assigned to the corresponding coordinate of the decoded synthetic instance $\tilde{x}_j^{(c)}$. Algorithms 1 and 2 specify a training procedure and decoding process of DDiF, respectively. In Algorithm 2, we include a loop over coordinates for clarity. However, it should be noted that in the actual implementation, the coordinate set is input to the neural network in a full-batch manner.

---

**Algorithm 1** Training procedure of DDiF

**Input:** Original real dataset $\mathcal{T}$; Dataset distillation loss $\mathcal{L}$; Initialized $\Psi = \{\psi_j\}_{j=1}^{|\Psi|}$; Learning rate $\eta$
**Output:** Parameterized synthetic dataset $\{(\psi_j, \tilde{y}_j)\}_{j=1}^{|\Psi|}$
1: Initialize coordinate set $\mathcal{C}$ from $x \in \mathcal{T}$
2: **for** $j = 1$ to $|\Psi|$ **do**
3:     Sample a real instance $(x, y) \sim \mathcal{T}$
4:     $\tilde{y}_j \leftarrow y$
5:     Optimize $\psi_j$ with Eq. (2)
6: **end for**
7: **repeat**
8:     Sample a real mini-batch $\mathcal{B}_\mathcal{T} \sim \mathcal{T}$
9:     $\mathcal{B}_\mathcal{S} \leftarrow \{F_\psi(\mathcal{C}) \mid \psi \in \Psi_\mathcal{B}\}$ from $\Psi_\mathcal{B} \sim \Psi$
10:     $\Psi \leftarrow \Psi - \eta \nabla_\Psi \mathcal{L}(\mathcal{B}_\mathcal{T}, \mathcal{B}_\mathcal{S})$
11: **until** convergence

---

**Algorithm 2** Decoding process of Synthetic instances in DDiF

**Input:** Set of parameters of synthetic neural fields $\Psi$; Coordinate set $\mathcal{C}$
**Output:** Set of decoded synthetic instances $X_S$
1: Initialize $X_S \leftarrow \emptyset$
2: **for** $j = 1$ to $|\Psi|$ **do**
3:     Initialize $\tilde{x}_j \in \mathbb{R}^{m \times N_1 \times \cdots \times N_n}$
4:     **for** $c \in \mathcal{C}$ **do**
5:         $\tilde{x}_j^{(c)} \leftarrow F_{\psi_j}(c)$
6:     **end for**
7:     $X_S \leftarrow X_S \cup \tilde{x}_j$
8: **end for**

---

## C  Additional Experimental Results

### C.1  Performance Comparison on Low-dimensional Datasets

To verify the wide applicability of DDiF, we conduct experiments on low-dimensional datasets, such as CIFAR-10 and CIFAR-100. In Table 10, DDiF exhibits highly competitive performances with previous studies. These results demonstrate that DDiF is also properly applicable to low-dimensional datasets, while DDiF shows significant performance improvement in high-dimensional datasets.

Table 10: Test accuracies (%) on CIFAR-10 and CIFAR-100. **Bold** and Underline means best and second-best performance of each column, respectively. "$-$" indicates no reported results.

| | Dataset | CIFAR10 | | | CIFAR100 | | |
|---|---|---|---|---|---|---|---|
| | IPC | 1 | 10 | 50 | 1 | 10 | 50 |
| Input-sized | TM | $46.3_{\pm 0.8}$ | $65.3_{\pm 0.7}$ | $71.6_{\pm 0.2}$ | $24.3_{\pm 0.3}$ | $40.1_{\pm 0.4}$ | $47.7_{\pm 0.2}$ |
| | FRePo | $46.8_{\pm 0.7}$ | $65.5_{\pm 0.4}$ | $71.7_{\pm 0.2}$ | $28.7_{\pm 0.1}$ | $42.5_{\pm 0.2}$ | $44.3_{\pm 0.2}$ |
| Static | IDC | $50.0_{\pm 0.4}$ | $67.5_{\pm 0.5}$ | $74.5_{\pm 0.2}$ | $-$ | $-$ | $-$ |
| | FreD | $60.6_{\pm 0.8}$ | $70.3_{\pm 0.3}$ | $75.8_{\pm 0.1}$ | $34.6_{\pm 0.4}$ | $42.7_{\pm 0.2}$ | $47.8_{\pm 0.1}$ |
| Parameterized | HaBa | $48.3_{\pm 0.8}$ | $69.9_{\pm 0.4}$ | $74.0_{\pm 0.2}$ | $-$ | $-$ | $47.0_{\pm 0.2}$ |
| | RTP | $66.4_{\pm 0.4}$ | $71.2_{\pm 0.4}$ | $73.6_{\pm 0.5}$ | $34.0_{\pm 0.4}$ | $42.9_{\pm 0.7}$ | $-$ |
| | HMN | $65.7_{\pm 0.3}$ | $73.7_{\pm 0.1}$ | $76.9_{\pm 0.2}$ | $36.3_{\pm 0.2}$ | $45.4_{\pm 0.2}$ | $48.5_{\pm 0.2}$ |
| | SPEED | $63.2_{\pm 0.1}$ | $73.5_{\pm 0.2}$ | **$77.7_{\pm 0.4}$** | $40.4_{\pm 0.4}$ | $45.9_{\pm 0.3}$ | $49.1_{\pm 0.2}$ |
| | NSD | **$68.5_{\pm 0.8}$** | $73.4_{\pm 0.2}$ | $75.2_{\pm 0.6}$ | $36.5_{\pm 0.3}$ | **$46.1_{\pm 0.2}$** | $-$ |
| Function | DDiF | $66.5_{\pm 0.4}$ | **$74.0_{\pm 0.4}$** | $77.5_{\pm 0.3}$ | **$42.1_{\pm 0.2}$** | $46.0_{\pm 0.2}$ | **$49.9_{\pm 0.2}$** |

## C.2 Additional Performance Comparison on High-dimensional Datasets

**Comparison under Large Budget (IPC=50).** In general, high-dimensional dataset distillation under a large budget is rarely addressed in previous studies due to their significant computational cost. To demonstrate the efficacy of DDiF even in a large budget setting, we conducted experiments on ImageNette ($128 \times 128$) under IPC=50. DDiF with TM achieves $75.2\%\pm1.3\%$ while vanilla with TM achieves $72.8\%\pm0.8\%$. It means that DDiF effectively improves the dataset distillation performance even with a larger storage budget for high resolution.

**Fixed DIPC under 256 resolution.** We further compare with input-sized parameterization on ImageNet-Subset ($256 \times 256$) under the fixed number of decoded synthetic instances. Under the IPC=1 setting, DDiF can decode 55 synthetic instances per class on 256 resolution. Similar to the main paper, we consider two settings: 1) varying the number of decoded synthetic instances from the baselines, and 2) varying the number of decoded synthetic instances from DDiF. Table 11 presents mixed performances on ImageNet-Subset ($256 \times 256$). However, we emphasize that DDiF utilizes a much smaller budget. Moreover, it is worth noting that DDiF achieves higher performance than Vanilla in some cases. We believe that it is related to previous findings that input-sized parameterization includes superfluous or irrelevant information (Lei & Tao, 2023; Yu et al., 2023; Sachdeva & McAuley, 2023). These experimental results support that DDiF involves sufficient representational power while using a much smaller budget compared to the input-sized parameterization.

Table 11: Test accuracies (%) on ImageNet-Subset ($256 \times 256$) with input-sized parameterization (Vanilla) and DDiF. "DIPC" denotes the number of decoded instances per class.

| DIPC | Methods | Utilized Budget | Nette | Woof | Fruit | Yellow | Meow | Squawk |
|---|---|---|---|---|---|---|---|---|
| 1 | Vanilla | 1,966.08k | **32.1** | 20.0 | 19.5 | 33.4 | **21.2** | 27.6 |
| | DDiF | 79.53k | $31.2_{\pm0.8}$ | $21.2_{\pm0.9}$ | $21.3_{\pm1.5}$ | $34.8_{\pm1.0}$ | $20.0_{\pm0.9}$ | $27.9_{\pm2.5}$ |
| 55 | Vanilla | $55 \times 1,966.08k$ | $70.1_{\pm1.0}$ | $37.5_{\pm1.2}$ | $41.3_{\pm0.8}$ | $64.2_{\pm1.6}$ | $47.1_{\pm0.5}$ | $64.3_{\pm1.2}$ |
| | DDiF | 1,966.08k | $67.8_{\pm1.0}$ | $39.6_{\pm1.6}$ | $43.2_{\pm1.7}$ | $63.1_{\pm0.8}$ | $44.8_{\pm1.1}$ | $67.0_{\pm0.9}$ |

## C.3 Compatibility with Soft Label

Recently, various previous literatures have demonstrated that using soft labels instead of one-hot labels leads to significant performance improvements in dataset distillation (Cui et al., 2023; Qin et al., 2024). As mentioned in the main paper, DDiF can also utilize a soft label technique, but we employ a fixed one-hot label for a fair comparison with the previous parameterization methods. Therefore, we conduct additional experiments to validate the compatibility of DDiF with soft label. We consider RDED

Table 12: Test accuracies (%) obtained by varying the synthetic label under IPC=50 and ConvNet.

| | Method | CIFAR-100 | ImageNette ($128 \times 128$) |
|---|---|---|---|
| Fixed one-hot label | RDED | $33.5_{\pm0.2}$ | $57.8_{\pm1.2}$ |
| | DATM | **50.8** | — |
| | DDiF | $49.9_{\pm0.2}$ | $\mathbf{75.2}_{\pm1.3}$ |
| Soft label | RDED | $57.0_{\pm0.1}$ | $83.8_{\pm0.2}$ |
| | DATM | $55.0_{\pm0.2}$ | — |
| | DDiF | $\mathbf{58.2}_{\pm0.2}$ | $\mathbf{86.9}_{\pm1.0}$ |

(Sun et al., 2024) and DATM (Guo et al., 2023) as our baselines since they employ the soft label and exhibit state-of-the-art performances on CIFAR-10, CIFAR-100, and ImageNette ($128 \times 128$). To apply soft labels to DDiF, we utilize the region-level soft label technique from RDED on the decoded synthetic instances, which are already obtained through training with TM. Table 12 presents the results as follows:

- RDED and DATM using soft label exhibit significant performance improvements compared to their performance without soft labels, consistently demonstrating the previous findings.

- Applying RDED's soft label technique to DDiF results in notable performance improvement, consistent with previous studies.

- Under the soft label setting, DDiF achieves higher performance with a significant gap than RDED and DATM, establishing a new state-of-the-art (SOTA).

In summary, we prioritize experiments under the fixed one-hot label setting to ensure a fair comparison with existing parameterization methods. However, our experiments confirm that DDiF can effectively utilize soft label, achieving substantial performance improvements when applied.

## C.4 ADDITIONAL RESULTS ON CROSS-RESOLUTION GENERALIZATION

**Detailed experimental results.** We apply the spatial-domain upsampling methods, such as nearest, bilinear, and bicubic, for the optimized synthetic instances of previous parameterization methods. Particularly, FreD can also utilize frequency-domain upsampling since it stores masked frequency coefficients. The most widely used for frequency-domain upsampling is zero-padding the frequency coefficients before inverse frequency transform, which means assigning zeros to the high-frequency components (Dugad & Ahuja, 2001). For example, in the case of DCT, which is the default setting of FreD, the process of upsampling an $N$-resolution frequency coefficient $F$ to an $M(> N)$-resolution image can be described as

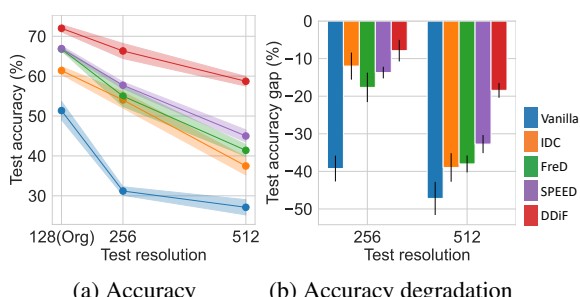

(a) Accuracy     (b) Accuracy degradation

Figure 9: (a) Test accuracies (%) with different image resolutions. The original resolution is $128 \times 128$. (b) Test accuracy gap (%) from original resolution. We use bilinear interpolation for previous studies. We utilize ConvNetD5, which is the same architecture in the distillation stage.

$\text{IDCT} \left( \begin{bmatrix} \lambda \times F & 0_{M-N} \\ 0_{M-N} & 0_{M-N} \end{bmatrix} \right)$ where $\lambda = \left( \frac{M}{N} \right)^2$ is the scaling factor. We refer to this frequency-domain upsampling as "zero-padding". For DDiF, we constructed a coordinate set suitable for the test resolution, and then input it into the optimized synthetic neural field to generate the upsampled synthetic instance. We refer to this method as "coordinate interpolation".

Table 13 presents the detailed experimental results. We observe that the previous parameterizations show drastic performance degradation regardless of the interpolation method used. Whereas, DDiF still achieves the highest cross-resolution generalization performance. Also, Figure 9 presents the cross-resolution generalization performance under the same network architecture in the distillation stage. As seen in Figure 5, DDiF achieves the best performance and shows the least performance degradation over all resolutions. These extensive results consistently demonstrate that DDiF is robust to resolution change, and this robustness is largely driven by the continuous nature of the synthetic neural field.

**Comparison with Full dataset downsampling.** The most intuitive and straightforward way to reduce the budget of a dataset is by downsampling, which reduces the budget size of each instance. It means that it is possible to downsample the full dataset to a specific resolution for storage and then upsample it to the test resolution. One may doubt that this simple method can show high cross-resolution generalization performance. To verify the superiority of DDiF on cross-resolution generalization, we additionally conduct performance comparison with full dataset downsampling.

Table 14 shows the experiment results where the full dataset was downsampled and then upsampled to the original resolution during the test phase. When the budget is similar (when the downsampled resolution is $4 \times 4$), DDiF outperforms the full dataset downsampling method. We also experimented when the downsampled resolution was the same as the resolution of the decoded synthetic instance. In this case, the downsampled dataset achieved better performance, as expected. However, this setting requires 1,289 times more budget than DDiF since the number of instances is not reduced. Such a setup deviates from the core purpose of dataset distillation, which aims to optimize performance under strict budget constraints.

Furthermore, we emphasize that the full dataset downsampling has two limitations on cross-resolution generalization. First, it requires a slightly different assumption. Cross-resolution generalization experiment, which we proposed, is modeled to evaluate the dataset distillation ability to generalize to higher resolution settings. It involves training on a low-resolution ($128 \times 128$) synthetic dataset and testing on high-resolution ($256 \times 256$) data. However, full dataset downsampling diverges from this cross-resolution setting since it assumes the availability of a high-resolution dataset ($256 \times 256$). Second, the training time during the evaluation stage increases since the number of data points remains the same as the full dataset. While the memory budget may be comparable, this increased time cost is undesirable, especially in scenarios where efficiency is critical.

Table 13: Test accuracies (%) with different resolutions and networks. The original resolution is $128 \times 128$. We denote the difference between ordinary and cross-resolution performance as Diff(%) $= ACC_{org} - ACC_{test}$ and relative ratio as Ratio $= \frac{ACC_{org} - ACC_{test}}{ACC_{org}}$.

| Test resolution | Test network | Method | Upsampling | Accuracy (↑) | Diff (↓) | Ratio (↓) |
|---|---|---|---|---|---|---|
| 256 | ConvNetD5 | Vanilla | nearest | $29.5_{\pm1.6}$ | 21.9 | 0.43 |
| | | | bilinear | $31.2_{\pm1.1}$ | 20.2 | 0.39 |
| | | | bicubic | $30.7_{\pm2.0}$ | 20.7 | 0.40 |
| | | IDC | nearest | $54.8_{\pm1.2}$ | 6.6 | 0.11 |
| | | | bilinear | $54.0_{\pm2.0}$ | 7.4 | 0.12 |
| | | | bicubic | $55.0_{\pm1.6}$ | 6.4 | 0.10 |
| | | SPEED | nearest | $58.8_{\pm1.4}$ | 8.1 | 0.12 |
| | | | bilinear | $57.7_{\pm0.8}$ | 9.2 | 0.14 |
| | | | bicubic | $58.1_{\pm1.0}$ | 8.8 | 0.13 |
| | | FreD | nearest | $55.2_{\pm2.2}$ | 11.6 | 0.17 |
| | | | bilinear | $55.0_{\pm2.6}$ | 11.8 | 0.16 |
| | | | bicubic | $56.4_{\pm1.4}$ | 10.4 | 0.16 |
| | | | zero-padding | $53.8_{\pm1.4}$ | 13.0 | 0.19 |
| | | DDiF | coord. interpolation | $\mathbf{66.3}_{\pm1.9}$ | **5.7** | **0.08** |
| | ConvNetD6 | Vanilla | nearest | $44.0_{\pm1.7}$ | 7.3 | 0.14 |
| | | | bilinear | $43.2_{\pm1.1}$ | 8.2 | 0.16 |
| | | | bicubic | $43.9_{\pm1.8}$ | 7.4 | 0.14 |
| | | IDC | nearest | $54.7_{\pm1.6}$ | 6.7 | 0.11 |
| | | | bilinear | $54.5_{\pm1.6}$ | 6.9 | 0.11 |
| | | | bicubic | $55.4_{\pm1.3}$ | 6.0 | 0.10 |
| | | SPEED | nearest | $62.0_{\pm1.0}$ | 4.9 | 0.07 |
| | | | bilinear | $61.8_{\pm1.8}$ | 5.1 | 0.08 |
| | | | bicubic | $62.6_{\pm1.1}$ | 4.3 | 0.06 |
| | | FreD | nearest | $60.9_{\pm0.8}$ | 5.9 | 0.09 |
| | | | bilinear | $60.1_{\pm0.3}$ | 6.7 | 0.10 |
| | | | bicubic | $61.4_{\pm0.8}$ | 5.8 | 0.09 |
| | | | zero-padding | $61.8_{\pm1.0}$ | 5.0 | 0.07 |
| | | DDiF | coord. interpolation | $\mathbf{70.6}_{\pm1.2}$ | **1.4** | **0.02** |
| 512 | ConvNetD5 | Vanilla | nearest | $27.4_{\pm1.4}$ | 24.0 | 0.47 |
| | | | bilinear | $27.1_{\pm1.9}$ | 24.2 | 0.47 |
| | | | bicubic | $27.1_{\pm1.0}$ | 24.3 | 0.47 |
| | | IDC | nearest | $38.6_{\pm2.7}$ | 22.8 | 0.37 |
| | | | bilinear | $37.5_{\pm2.3}$ | 23.9 | 0.39 |
| | | | bicubic | $39.5_{\pm2.1}$ | 21.9 | 0.36 |
| | | SPEED | nearest | $43.3_{\pm2.2}$ | 23.6 | 0.35 |
| | | | bilinear | $45.0_{\pm1.5}$ | 21.9 | 0.33 |
| | | | bicubic | $44.8_{\pm3.0}$ | 22.1 | 0.33 |
| | | FreD | nearest | $42.5_{\pm2.5}$ | 24.3 | 0.36 |
| | | | bilinear | $41.4_{\pm1.5}$ | 25.4 | 0.38 |
| | | | bicubic | $41.6_{\pm1.3}$ | 25.2 | 0.38 |
| | | | zero-padding | $42.9_{\pm1.5}$ | 23.9 | 0.36 |
| | | DDiF | coord. interpolation | $\mathbf{58.7}_{\pm1.2}$ | **13.3** | **0.18** |
| | ConvNetD7 | Vanilla | nearest | $41.2_{\pm1.5}$ | 10.1 | 0.20 |
| | | | bilinear | $40.6_{\pm2.6}$ | 10.7 | 0.21 |
| | | | bicubic | $40.4_{\pm1.9}$ | 10.9 | 0.21 |
| | | IDC | nearest | $51.5_{\pm1.8}$ | 9.9 | 0.16 |
| | | | bilinear | $50.7_{\pm2.1}$ | 10.7 | 0.17 |
| | | | bicubic | $51.2_{\pm2.8}$ | 10.7 | 0.17 |
| | | SPEED | nearest | $59.6_{\pm2.0}$ | 7.3 | 0.11 |
| | | | bilinear | $59.5_{\pm2.0}$ | 7.4 | 0.11 |
| | | | bicubic | $60.1_{\pm1.7}$ | 6.8 | 0.10 |
| | | FreD | nearest | $55.1_{\pm1.2}$ | 11.7 | 0.18 |
| | | | bilinear | $53.8_{\pm1.0}$ | 13.0 | 0.19 |
| | | | bicubic | $54.4_{\pm0.9}$ | 12.4 | 0.19 |
| | | | zero-padding | $56.3_{\pm0.8}$ | 10.5 | 0.16 |
| | | DDiF | coord. interpolation | $\mathbf{69.0}_{\pm1.0}$ | **3.0** | **0.04** |

Table 14: Performance comparison when the test resolution is $256 \times 256$. We utilize bicubic interpolation for full dataset resizing. The relative budget ratio indicates the ratio of full dataset downsampling over DDiF.

| Test network | Method | Original resolution | Downsampled resolution | Relative budget ratio | Accuracy |
|---|---|---|---|---|---|
| ConvNetD5 | Downsample | $256 \times 256$ $256 \times 256$ | $4\times4$ $128 \times 128$ | 1.3 1,289.4 | $45.2_{\pm 2.3}$ $91.3_{\pm 0.5}$ |
| | DDiF | $128 \times 128$ | – | 1.0 | $66.3_{\pm 1.9}$ |
| ConvNetD6 | Downsample | $256 \times 256$ $256 \times 256$ | $4\times4$ $128 \times 128$ | 1.3 1,289.4 | $44.7_{\pm 0.4}$ $91.2_{\pm 0.0}$ |
| | DDiF | $128 \times 128$ | – | 1.0 | $70.6_{\pm 1.2}$ |

## C.5 ROBUSTNESS TO CORRUPTION

We further investigate the robustness against corruption of DDiF in the trained synthetic datasets by evaluating on ImageNet-Subset-C. This subset is designed specifically to assess robustness across varying corruption types and severity levels. We report the average test accuracies over 15 corruption types, each evaluated across 5 levels of severity, for each class in ImageNet-Subset-C. Table 15 summarizes the performance of DDiF and baseline models. The results in Table 15 clearly demonstrate that DDiF has the same substantial robustness to resolution change and corruption.

Table 15: Test accuracies (%) on ImageNet-Subset-C under IPC=1. Accuracy on ImageSquawk-C of TM is not reported in previous works.

| Method | ImageNette-C | ImageWoof-C | ImageFruit-C | ImageYellow-C | ImageMeow-C | ImageSquawk-C |
|---|---|---|---|---|---|---|
| TM | $38.0_{\pm1.6}$ | $23.8_{\pm1.0}$ | $22.7_{\pm1.1}$ | $35.6_{\pm1.7}$ | $23.2_{\pm1.1}$ | - |
| IDC | $34.5_{\pm0.6}$ | $18.7_{\pm0.4}$ | $28.5_{\pm0.9}$ | $36.8_{\pm1.4}$ | $22.2_{\pm1.2}$ | $26.8_{\pm0.5}$ |
| FreD | $51.2_{\pm0.6}$ | $31.0_{\pm0.9}$ | $32.3_{\pm1.4}$ | $48.2_{\pm1.0}$ | $30.3_{\pm0.3}$ | $45.9_{\pm0.6}$ |
| DDiF | $\mathbf{54.5}_{\pm0.6}$ | $\mathbf{34.0}_{\pm0.4}$ | $\mathbf{36.6}_{\pm0.4}$ | $47.2_{\pm0.7}$ | $\mathbf{30.3}_{\pm0.8}$ | $\mathbf{53.8}_{\pm0.5}$ |

## C.6 TIME COMPLEXITY

As mentioned earlier, the neural field takes coordinates as input and produces quantities as output. This distinct characteristic of the neural field offers the advantage of being resolution-invariant but may raise concerns regarding the decoding process time. We admit that the decoding time of DDiF indeed increases as resolution grows due to the need to forward a larger number of coordinates through the neural field. To address this concern, we conducted an experiment measuring the wall-clock time for decoding a single instance.

Table 16 shows the results with 128 image resolution demonstrate that while the time cost of DDiF is larger than methods relying on non-parameterized decoding functions, such as IDC and FreD, it remains comparable to methods that use parameterized decoding functions, such as HaBa and SPEED, and

Table 16: Wall-clock time (ms) of the decoding process for a single synthetic instance. "ms" indicates the millisecond.

| | $128 \times 128$ | $256 \times 256$ |
|---|---|---|
| Vanilla | 0.31 | 0.31 |
| IDC | 0.40 | 2.83 |
| FreD | 0.46 | 1.20 |
| HaBa | 2.81 | – |
| SPEED | 2.20 | – |
| GLaD | 31.33 | – |
| DDiF | 2.49 | 3.25 |

exhibits a lower time cost than methods that utilize pre-trained generative models, such as GLaD. As the resolution increases to 256, the decoding time of DDiF also increases and it is slightly larger than non-parameterized decoding functions. In conclusion, although the decoding process time of DDiF increases as the resolution increases, it does not differ significantly from that of conventional parameterization methods. We attribute this to 1) the use of a small neural network structure for the synthetic neural field and 2) the full-batch forwarding of the coordinate set in the implementation.

## C.7 FULL TABLE WITH STANDARD DEVIATION AND ADDITIONAL VISUALIZATION

For improved layout, we have positioned full tables with standard deviation and additional example figures at the end of the paper. Please refer to Table 17 for ImageNet-Subset ($128 \times 128$) under IPC=1; Table 18 for ImageNet-Subset ($256 \times 256$) under IPC=1; Table 19 for ImageNet-Subset under ($128 \times 128$) IPC=10; Table 20 for cross-architecture; and Table 21 for robustness to the loss. In addition, please refer to Figures 10 to 15 for Image domain; Figure 16 for 3D domain; and Figure 17 for video domain.

# D DISCUSSIONS

## D.1 MORE COMPARISON WITH FRED

As seen in Eqs. (4) and (5), the functions represented by DDiF and FreD are similar, both being the sum of cosine functions. Therefore, we can perform a term-by-term comparison of both equations.

- DDiF enables to change the amplitudes $A_{k,i}$, frequencies $\omega_k$, phases $\varphi'_{k,i}$, and shift $b^{(0)}$, while FreD only allows the amplitudes. It indicates that DDiF has higher representation ability than FreD.

- Although FreD is a finite sum of cosine functions with a fixed frequency, DDiF represents an infinite sum of cosine functions with tunable frequency. It means that DDiF can cover a wide range of frequencies from low to high by selecting various $k$. According to the empirical findings in Wang et al. (2020a), it has been demonstrated that datasets with a larger number of frequencies exhibit improved generalization performance.

- DDiF is a continuous function, whereas FreD is a discrete function. Due to this characteristic, FreD cannot encode information for coordinates that were not provided during the distillation stage. In contrast, since DDiF operates over a continuous domain, it inherently stores information for coordinates that were not supplied during the distillation stage.

In summary, DDiF has a more flexible and expressive function than FreD.

## D.2 DISCUSSION ABOUT THEORETICAL ANALYSIS

We provide the theoretical analysis to propose a framework for comparing parameterization methods, which have traditionally been evaluated solely based on performance, through expressiveness, i.e. the size of the feasible space. Proposition 3.1 indicates that a larger feasible space for decoded synthetic instances through parameterization methods results in lower dataset distillation loss. Theorem 3.2 claims that DDiF has greater expressiveness than prior work (FreD) when the utilized parameters for a single synthetic instance are the same. We believe that the proposed theoretical analysis framework will serve as a cornerstone for future theoretical comparisons of parameterization methods in dataset distillation areas.

However, this theoretical analysis still has room for further improvement. In the theoretical analysis, we compare the expressiveness under the fixed number of decoded instances and/or the same utilized parameters for a single synthetic instance, not the fixed entire storage budget. We empirically demonstrate the superiority of DDiF in each of the aforementioned cases: 1) the fixed number of decoded instances (see Table 7), 2) the same utilized parameters for a single synthetic instance (see Figure 3), and 3) the fixed entire storage budget (see Table 1 and others). Even though the proposed theoretical analysis in this paper is experimentally verified through extensive results, this framework has the limitation of not primarily focusing on the fixed entire storage budget scenario, which is the most basic setting in dataset distillation.

We believe that constructing a theoretical framework to compare the expressiveness of parameterization methods under a fixed entire storage budget is necessary, and the proposed theoretical analysis in this paper can serve as a foundational background for such efforts.

## D.3 COMPARISON WITH DIM

As mentioned in Section 3, the synthetic function has several possible forms. DiM (Wang et al., 2023) employs a probability density function as a synthetic function and utilizes a deep generative model to parameterize it. Specifically, the decoded synthetic instance of DiM is the sampled output of a deep generative model by inputting random noise:

$$\min_{\phi} \mathcal{L}(\mathcal{T}, \mathcal{S}) \text{ where } \mathcal{S} = g_\phi(\mathcal{Z}), \ \mathcal{Z} \sim \mathcal{N}(0, I)$$

DDiF and DiM have in common that they store the distilled information in the synthetic function and only store the parameters of the function without any additional codes. However, there are several structural differences.

- The output of DiM still depends on the data dimension. As aforementioned, this type of decoding function requires a more complicated structure and storage budget as the data dimension grows larger. Actually, DiM has not been extensively tested on high-resolution datasets. On the contrary, DDiF stores information regardless of data dimension, which indicates broader applicability across various resolutions.

- The decoding process of DiM is stochastic. Due to the stochasticity, DiM can sample the diverse decoded synthetic instances and save the redeployment cost. However, at the same time, DiM carries the risk of generating less informative synthetic instances. Consequently, it leads to instability in training on downstream tasks. Furthermore, DiM might suffer from redundant sampling due to mode collapse, a well-known issue of the generative model. Meanwhile, since the decoding process of DDiF is deterministic, DDiF has an advantage in stability.

## D.4 LIMITATION

**Less efficiency on Low-dimensional datasets.** One of the main ideas in parameterization is expanding the trade-off curve between quantity and quality: reducing the utilized budget of each synthetic instance, while maximally preserving the expressiveness of it. While DDiF has an extensive feature coverage theoretically and shows competitive performances with previous studies experimentally, it might be less efficient for some low-dimensional datasets due to the structural features of the neural field. This is because, given the low-dimensional instance, it can be difficult to design a neural field that is sufficiently expressive with fewer parameters. In spite of this issue, we repeatedly highlight that DDiF has a significant performance improvement on high-dimensional datasets. Furthermore, we speculate that a deeper analysis of the neural field structure could be an interesting direction for future research in dataset distillation.

**Individual Parameterization.** Several studies have claimed that storing intra- and inter-class information in a shared component is effective. From this perspective, our proposed method, which has a one-to-one correspondence between synthetic instances and synthetic neural fields, does not have a component to store shared information. We believe that it can be extended by adding modulation or conditional code, and this paper may serve as a good starting point.

Table 17: Test accuracies (%) on ImageNet-Subset ($128 \times 128$) with regard to various parameterization methods under IPC=1.

| | Subset | Nette | Woof | Fruit | Yellow | Meow | Squawk |
|---|---|---|---|---|---|---|---|
| Input-sized | TM | $51.4_{\pm 2.3}$ | $29.7_{\pm 0.9}$ | $28.8_{\pm 1.2}$ | $47.5_{\pm 1.5}$ | $33.3_{\pm 0.7}$ | $41.0_{\pm 1.5}$ |
| | FRePo | $48.1_{\pm 0.7}$ | $29.7_{\pm 0.6}$ | - | - | - | - |
| Static | IDC | $61.4_{\pm 1.0}$ | $34.5_{\pm 1.1}$ | $38.0_{\pm 1.1}$ | $56.5_{\pm 1.8}$ | $39.5_{\pm 1.5}$ | $50.2_{\pm 1.5}$ |
| | FreD | $66.8_{\pm 0.4}$ | $38.3_{\pm 1.5}$ | $43.7_{\pm 1.6}$ | $63.2_{\pm 1.0}$ | $43.2_{\pm 0.8}$ | $57.0_{\pm 0.8}$ |
| Parameterized | HaBa | $51.9_{\pm 1.7}$ | $32.4_{\pm 0.7}$ | $34.7_{\pm 1.1}$ | $50.4_{\pm 1.6}$ | $36.9_{\pm 0.9}$ | $41.9_{\pm 1.4}$ |
| | SPEED | $66.9_{\pm 0.7}$ | $38.0_{\pm 0.9}$ | $43.4_{\pm 0.6}$ | $62.6_{\pm 1.3}$ | $43.6_{\pm 0.7}$ | $60.9_{\pm 1.0}$ |
| | TM+RTP | $\underline{69.6}_{\pm 0.4}$ | $\underline{38.8}_{\pm 1.1}$ | $\underline{45.2}_{\pm 1.7}$ | $\underline{66.4}_{\pm 0.5}$ | $\underline{46.5}_{\pm 1.8}$ | $\underline{63.2}_{\pm 1.0}$ |
| | NSD | $68.6_{\pm 0.8}$ | $35.2_{\pm 0.4}$ | $39.8_{\pm 0.2}$ | $61.0_{\pm 0.5}$ | $45.2_{\pm 0.1}$ | $52.9_{\pm 0.7}$ |
| DGM Prior | GLaD | $38.7_{\pm 1.6}$ | $23.4_{\pm 1.1}$ | $23.1_{\pm 0.4}$ | $-$ | $26.0_{\pm 1.1}$ | $35.8_{\pm 1.4}$ |
| | H-GLaD | $45.4_{\pm 1.1}$ | $28.3_{\pm 0.2}$ | $25.6_{\pm 0.7}$ | $-$ | $29.6_{\pm 1.0}$ | $39.7_{\pm 0.8}$ |
| Function | DDiF | $\mathbf{72.0}_{\pm 0.9}$ | $\mathbf{42.9}_{\pm 0.7}$ | $\mathbf{48.2}_{\pm 1.2}$ | $\mathbf{69.0}_{\pm 0.8}$ | $\mathbf{47.4}_{\pm 1.3}$ | $\mathbf{67.0}_{\pm 1.3}$ |

Table 18: Test accuracies (%) on ImageNet-Subset ($256 \times 256$) with regard to various parameterizaiton methods under IPC=1.

| | Subset | Nette | Woof | Fruit | Yellow | Meow | Squawk |
|---|---|---|---|---|---|---|---|
| Input-sized | DM | 32.1 | 20.0 | 19.5 | 33.4 | 21.2 | 27.6 |
| Static | IDC | $53.7_{\pm 1.2}$ | $30.2_{\pm 1.5}$ | $\underline{33.1}_{\pm 1.5}$ | $\underline{52.2}_{\pm 1.4}$ | $34.6_{\pm 1.8}$ | $47.0_{\pm 1.5}$ |
| | FreD | $54.2_{\pm 1.1}$ | $\underline{31.2}_{\pm 0.9}$ | $32.5_{\pm 1.9}$ | $49.1_{\pm 0.4}$ | $34.0_{\pm 1.2}$ | $43.1_{\pm 1.5}$ |
| Parameterized | SPEED | $\underline{57.7}_{\pm 0.9}$ | $-$ | $-$ | $-$ | $-$ | $-$ |
| DGM Prior | LatentDM | 56.1 | 28.0 | 30.7 | $-$ | $\underline{36.3}$ | $\underline{47.1}$ |
| Function | DDiF | $\mathbf{67.8}_{\pm 1.0}$ | $\mathbf{39.6}_{\pm 1.6}$ | $\mathbf{43.2}_{\pm 1.7}$ | $\mathbf{63.1}_{\pm 0.8}$ | $\mathbf{44.8}_{\pm 1.1}$ | $\mathbf{67.0}_{\pm 0.9}$ |

Table 19: Test accuracies (%) on ImageNet-Subset ($128 \times 128$) with regard to various parameterization methods under IPC=10.

| | Subset | Nette | Woof | Fruit | Yellow | Meow | Squawk |
|---|---|---|---|---|---|---|---|
| Input-sized | TM | $63.0_{\pm 1.3}$ | $35.8_{\pm 1.8}$ | $40.3_{\pm 1.3}$ | $60.0_{\pm 1.5}$ | $40.4_{\pm 2.2}$ | $52.3_{\pm 1.0}$ |
| | FRePo | $66.5_{\pm 0.8}$ | $42.2_{\pm 0.9}$ | $-$ | $-$ | $-$ | $-$ |
| Static | IDC | $70.8_{\pm 0.5}$ | $39.8_{\pm 0.9}$ | $46.4_{\pm 1.4}$ | $68.7_{\pm 0.8}$ | $47.9_{\pm 1.4}$ | $65.4_{\pm 1.2}$ |
| | FreD | $72.0_{\pm 0.8}$ | $41.3_{\pm 1.2}$ | $47.0_{\pm 1.1}$ | $69.2_{\pm 0.6}$ | $48.6_{\pm 0.4}$ | $67.3_{\pm 0.8}$ |
| Parameterized | HaBa | $64.7_{\pm 1.6}$ | $38.6_{\pm 1.3}$ | $42.5_{\pm 1.6}$ | $63.0_{\pm 1.6}$ | $42.9_{\pm 0.9}$ | $56.8_{\pm 1.0}$ |
| | SPEED | $\underline{72.9}_{\pm 1.5}$ | $\underline{44.1}_{\pm 1.4}$ | $\mathbf{50.0}_{\pm 0.8}$ | $\mathbf{70.5}_{\pm 1.5}$ | $\mathbf{52.0}_{\pm 1.3}$ | $\underline{71.8}_{\pm 1.3}$ |
| Function | DDiF | $\mathbf{74.6}_{\pm 0.7}$ | $\mathbf{44.9}_{\pm 0.5}$ | $\underline{49.8}_{\pm 0.8}$ | $\mathbf{70.5}_{\pm 1.8}$ | $\underline{50.6}_{\pm 1.1}$ | $\mathbf{72.3}_{\pm 1.3}$ |

Table 20: Test accuracies (%) on cross Architecture networks with ImageNet Subsets ($128 \times 128$) under IPC=1

| Test Net | Method | Nette | Woof | Fruit | Yellow | Meow | Squawk |
|---|---|---|---|---|---|---|---|
| AlexNet | TM | $13.2_{\pm 0.6}$ | $10.0_{\pm 0.0}$ | $10.0_{\pm 0.0}$ | $11.0_{\pm 0.2}$ | $9.8_{\pm 0.0}$ | $-$ |
| | IDC | $17.4_{\pm 0.9}$ | $16.5_{\pm 0.7}$ | $\underline{17.9}_{\pm 0.7}$ | $\underline{20.6}_{\pm 0.9}$ | $\underline{16.8}_{\pm 0.5}$ | $20.7_{\pm 1.0}$ |
| | FreD | $\underline{35.7}_{\pm 0.4}$ | $\underline{23.9}_{\pm 0.7}$ | $15.8_{\pm 0.7}$ | $19.8_{\pm 1.2}$ | $14.4_{\pm 0.5}$ | $\underline{36.3}_{\pm 0.3}$ |
| | DDiF | $\mathbf{60.7}_{\pm 2.3}$ | $\mathbf{36.4}_{\pm 2.3}$ | $\mathbf{41.8}_{\pm 0.6}$ | $\mathbf{56.2}_{\pm 0.8}$ | $\mathbf{40.3}_{\pm 1.9}$ | $\mathbf{60.5}_{\pm 0.4}$ |
| VGG11 | TM | $17.4_{\pm 2.1}$ | $12.6_{\pm 1.8}$ | $11.8_{\pm 1.0}$ | $16.9_{\pm 1.1}$ | $\underline{13.8}_{\pm 1.3}$ | $-$ |
| | IDC | $19.6_{\pm 1.5}$ | $16.0_{\pm 2.1}$ | $\underline{13.8}_{\pm 1.3}$ | $16.8_{\pm 3.5}$ | $13.1_{\pm 2.0}$ | $\underline{19.1}_{\pm 1.2}$ |
| | FreD | $\underline{21.8}_{\pm 2.9}$ | $\underline{17.1}_{\pm 1.7}$ | $12.6_{\pm 2.6}$ | $\underline{18.2}_{\pm 1.1}$ | $13.2_{\pm 1.9}$ | $18.6_{\pm 2.3}$ |
| | DDiF | $\mathbf{53.6}_{\pm 1.5}$ | $\mathbf{29.9}_{\pm 1.9}$ | $\mathbf{33.8}_{\pm 1.9}$ | $\mathbf{44.2}_{\pm 1.7}$ | $\mathbf{32.0}_{\pm 1.8}$ | $\mathbf{37.9}_{\pm 1.5}$ |
| ResNet18 | TM | $34.9_{\pm 2.3}$ | $20.7_{\pm 1.0}$ | $23.1_{\pm 1.5}$ | $43.4_{\pm 1.1}$ | $22.8_{\pm 2.2}$ | $-$ |
| | IDC | $43.6_{\pm 1.3}$ | $23.2_{\pm 0.8}$ | $32.9_{\pm 2.8}$ | $44.2_{\pm 3.5}$ | $28.2_{\pm 0.5}$ | $47.8_{\pm 1.9}$ |
| | FreD | $\underline{48.8}_{\pm 1.8}$ | $\underline{28.4}_{\pm 0.6}$ | $\underline{34.0}_{\pm 1.9}$ | $\underline{49.3}_{\pm 1.1}$ | $\underline{29.0}_{\pm 1.8}$ | $\underline{50.2}_{\pm 0.8}$ |
| | DDiF | $\mathbf{63.8}_{\pm 1.8}$ | $\mathbf{37.5}_{\pm 1.9}$ | $\mathbf{42.0}_{\pm 1.9}$ | $\mathbf{55.9}_{\pm 1.0}$ | $\mathbf{35.8}_{\pm 1.8}$ | $\mathbf{62.6}_{\pm 1.5}$ |
| ViT | TM | $22.6_{\pm 1.1}$ | $15.9_{\pm 0.4}$ | $23.3_{\pm 0.4}$ | $18.1_{\pm 1.3}$ | $18.6_{\pm 0.9}$ | $-$ |
| | IDC | $31.0_{\pm 0.6}$ | $22.4_{\pm 0.8}$ | $31.1_{\pm 0.8}$ | $30.3_{\pm 0.6}$ | $\underline{21.4}_{\pm 0.7}$ | $32.2_{\pm 1.2}$ |
| | FreD | $\underline{38.4}_{\pm 0.7}$ | $\underline{25.4}_{\pm 1.7}$ | $\underline{31.9}_{\pm 1.4}$ | $\underline{37.6}_{\pm 2.0}$ | $19.7_{\pm 0.8}$ | $\underline{44.4}_{\pm 1.0}$ |
| | DDiF | $\mathbf{59.0}_{\pm 0.4}$ | $\mathbf{32.8}_{\pm 0.8}$ | $\mathbf{39.4}_{\pm 0.8}$ | $\mathbf{47.9}_{\pm 0.6}$ | $\mathbf{27.0}_{\pm 0.6}$ | $\mathbf{54.8}_{\pm 1.1}$ |

Table 21: Compatibility on different dataset distillation loss with ImageNet Subsets ($128 \times 128$) under IPC=1

| $\mathcal{L}$ | Method | Nette | Woof | Fruit | Meow | Squawk |
|---|---|---|---|---|---|---|
| DC | Vanilla | $34.2_{\pm 1.7}$ | $22.5_{\pm 1.0}$ | $21.0_{\pm 0.9}$ | $22.0_{\pm 0.6}$ | $32.0_{\pm 1.5}$ |
| | IDC | $45.4_{\pm 0.7}$ | $25.5_{\pm 0.7}$ | $26.8_{\pm 0.4}$ | $25.3_{\pm 0.6}$ | $34.6_{\pm 0.5}$ |
| | FreD | $\underline{49.1}_{\pm 0.8}$ | $\underline{26.1}_{\pm 1.1}$ | $\underline{30.0}_{\pm 0.7}$ | $\underline{28.7}_{\pm 1.0}$ | $\underline{39.7}_{\pm 0.7}$ |
| | GLaD | $35.4_{\pm 1.2}$ | $22.3_{\pm 1.1}$ | $20.7_{\pm 1.1}$ | $22.6_{\pm 0.8}$ | $33.8_{\pm 0.9}$ |
| | H-GLaD | $36.9_{\pm 0.8}$ | $24.0_{\pm 0.8}$ | $22.4_{\pm 1.1}$ | $24.1_{\pm 0.9}$ | $35.3_{\pm 1.0}$ |
| | DDiF | $\mathbf{61.2}_{\pm 1.0}$ | $\mathbf{35.2}_{\pm 1.7}$ | $\mathbf{37.8}_{\pm 1.1}$ | $\mathbf{39.1}_{\pm 1.3}$ | $\mathbf{54.3}_{\pm 1.0}$ |
| DM | Vanilla | $30.4_{\pm 2.7}$ | $20.7_{\pm 1.0}$ | $20.4_{\pm 1.9}$ | $20.1_{\pm 1.2}$ | $26.6_{\pm 2.6}$ |
| | IDC | $48.3_{\pm 1.3}$ | $27.0_{\pm 1.0}$ | $29.9_{\pm 0.7}$ | $30.5_{\pm 1.0}$ | $38.8_{\pm 1.4}$ |
| | FreD | $\underline{56.2}_{\pm 1.0}$ | $\underline{31.0}_{\pm 1.2}$ | $\underline{33.4}_{\pm 0.5}$ | $\underline{33.3}_{\pm 0.6}$ | $\underline{42.7}_{\pm 0.8}$ |
| | GLaD | $32.2_{\pm 1.7}$ | $21.2_{\pm 1.5}$ | $21.8_{\pm 1.8}$ | $22.3_{\pm 1.6}$ | $27.6_{\pm 1.9}$ |
| | H-GLaD | $34.8_{\pm 1.0}$ | $23.9_{\pm 1.9}$ | $24.4_{\pm 2.1}$ | $24.2_{\pm 1.1}$ | $29.5_{\pm 1.5}$ |
| | DDiF | $\mathbf{69.2}_{\pm 1.0}$ | $\mathbf{42.0}_{\pm 0.4}$ | $\mathbf{45.3}_{\pm 1.8}$ | $\mathbf{45.8}_{\pm 1.1}$ | $\mathbf{64.6}_{\pm 1.1}$ |

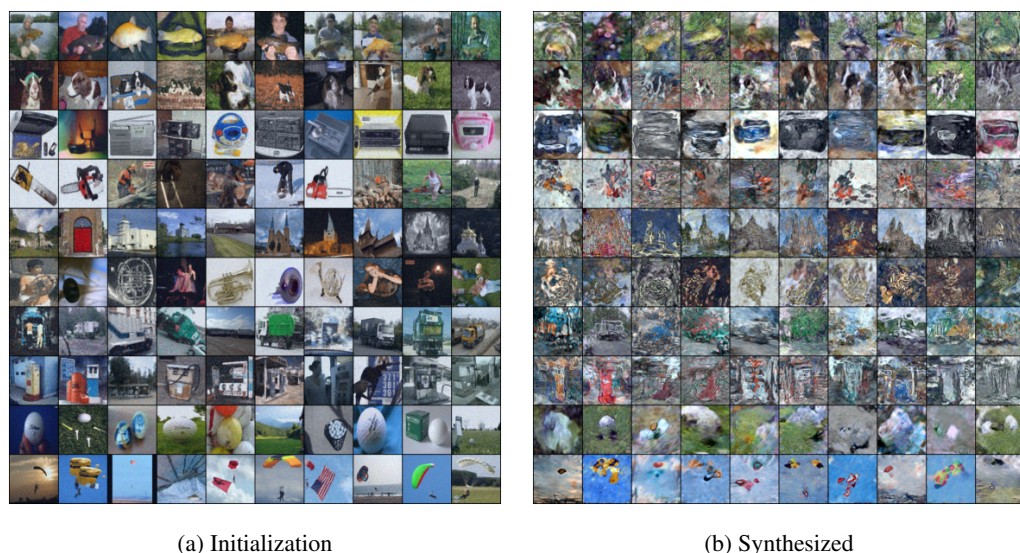

(a) Initialization                    (b) Synthesized

Figure 10: (a) Warm-up initialized images on ImageNette with DDiF, (b) Best-performed synthetic dataset represented by DDiF. We visualize the first 10 images, while DDiF constructs 51 images per class under the same budget.

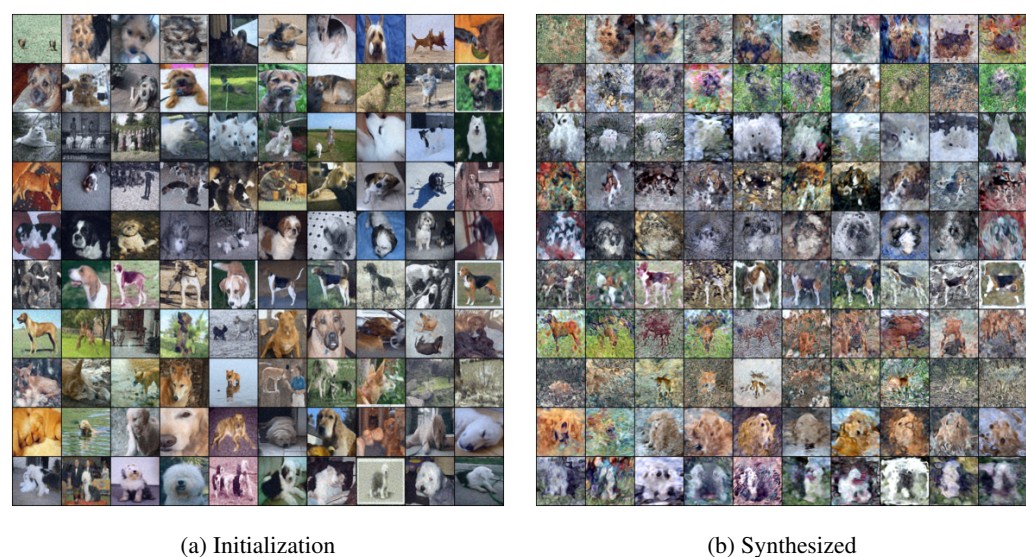

(a) Initialization                    (b) Synthesized

Figure 11: (a) Warm-up initialized images on ImageWoof with DDiF, (b) Best-performed synthetic dataset represented by DDiF. We visualize the first 10 images, while DDiF constructs 51 images per class under the same budget.

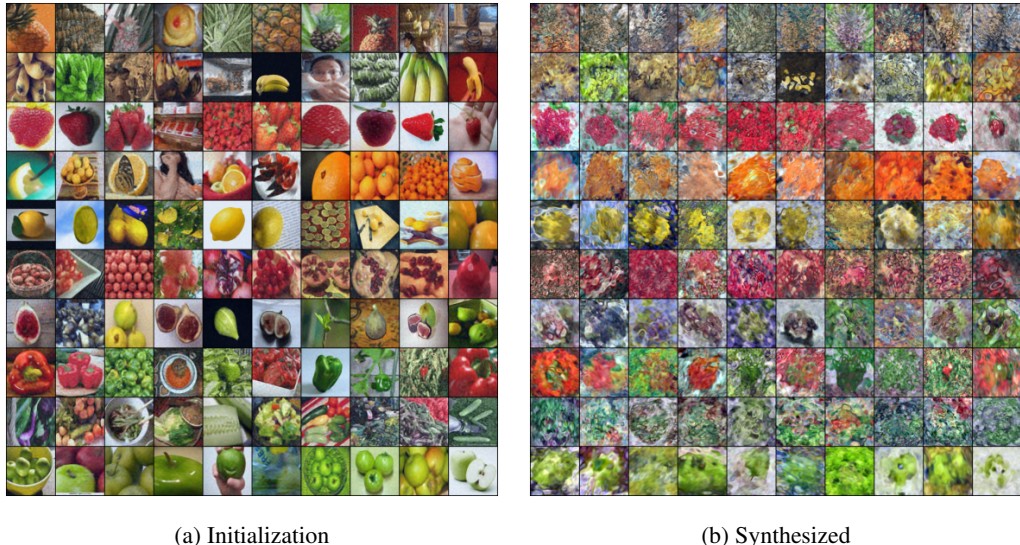

(a) Initialization          (b) Synthesized

Figure 12: (a) Warm-up initialized images on ImageFruit with DDiF, (b) Best-performed synthetic dataset represented by DDiF. We visualize the first 10 images, while DDiF constructs 51 images per class under the same budget.

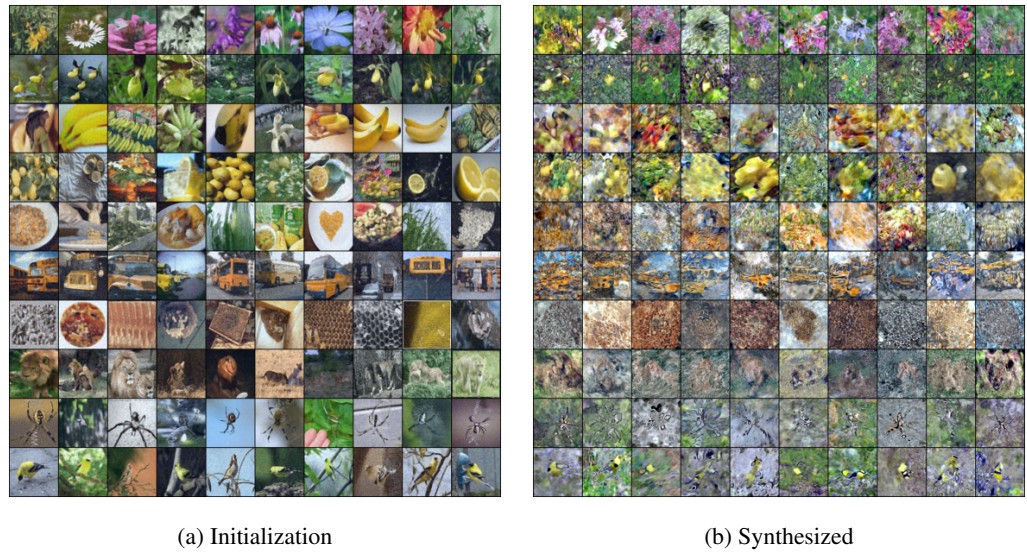

(a) Initialization          (b) Synthesized

Figure 13: (a) Warm-up initialized images on ImageYellow with DDiF, (b) Best-performed synthetic dataset represented by DDiF. We visualize the first 10 images, while DDiF constructs 51 images per class under the same budget.

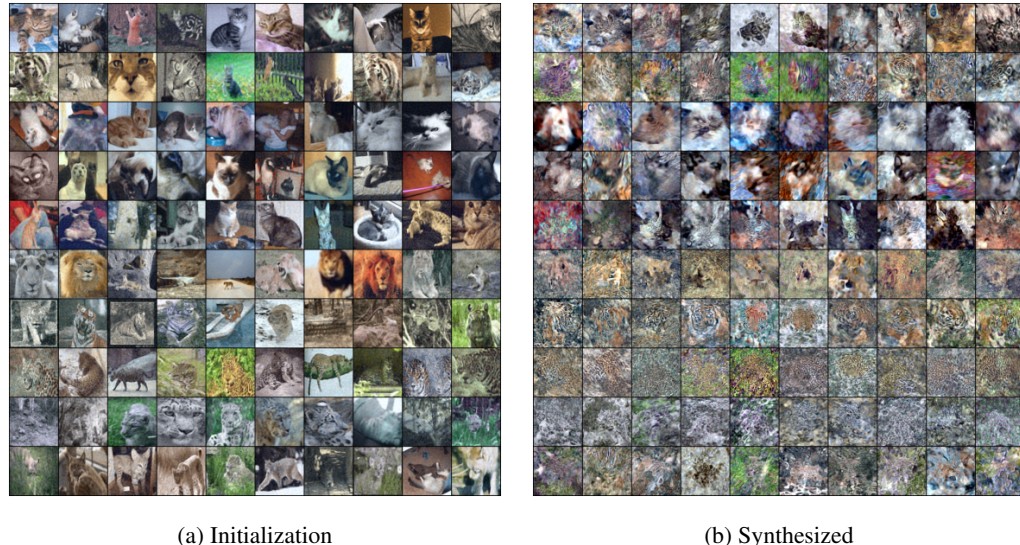

(a) Initialization                              (b) Synthesized

Figure 14: (a) Warm-up initialized images on ImageMeow with DDiF, (b) Best-performed synthetic dataset represented by DDiF. We visualize the first 10 images, while DDiF constructs 51 images per class under the same budget.

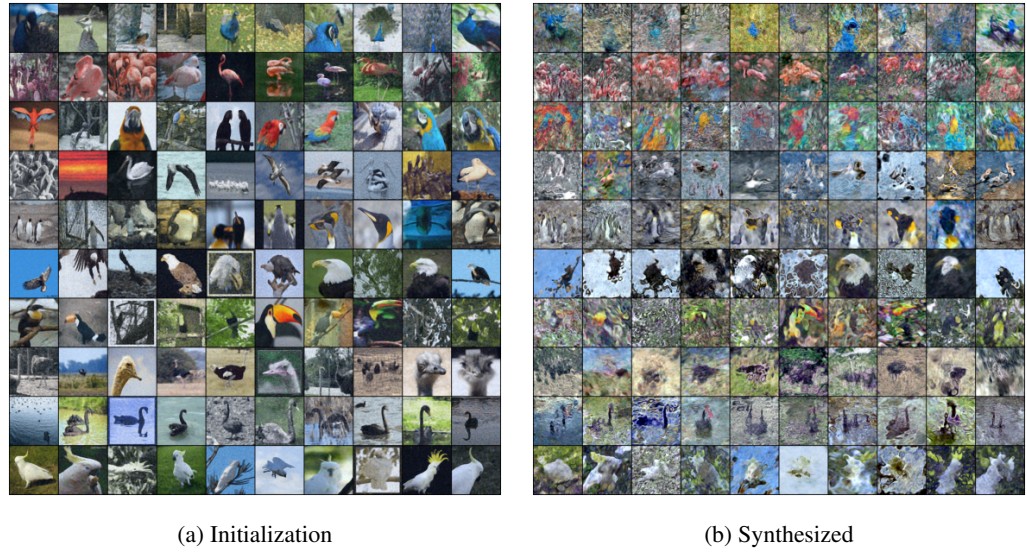

(a) Initialization                              (b) Synthesized

Figure 15: (a) Warm-up initialized images on ImageSquawk with DDiF, (b) Best-performed synthetic dataset represented by DDiF. We visualize the first 10 images, while DDiF constructs 51 images per class under the same budget.

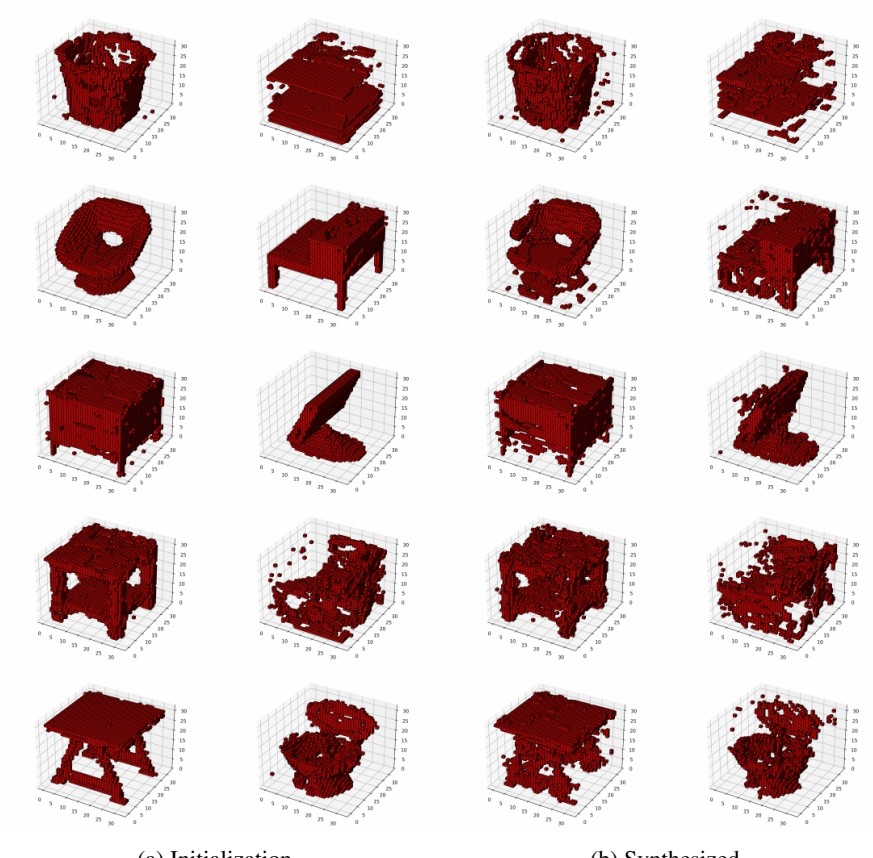

(a) Initialization  (b) Synthesized

Figure 16: (a) Warm-up initialization images and (b) Synthesized images of ModelNet-10. Start reading labels from the top and continue to right: 1) bathtub, 2) bed, 3) chair, 4) desk, 5) dresser, 6) monitor, 7) nightstand, 8) sofa, 9) table, 10) toilet

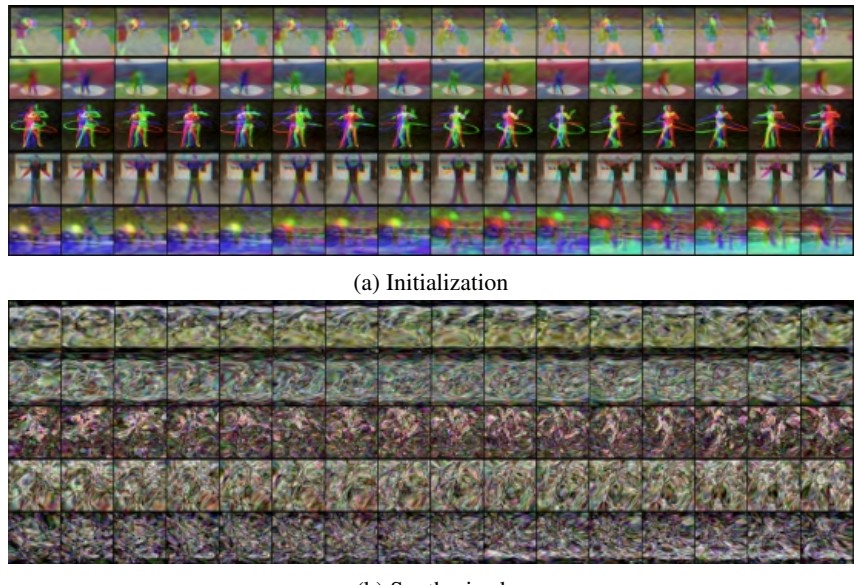

Figure 17: (a) Warm-up initialization images and (b) Synthesized images of MiniUCF. Start reading labels from the top: 1) FrisbeeCatch, 2) HammerThrow, 3) HulaHoop, 4) JumpingJack, 5) Parallel-Bars

