# OpenReview forum: "Distilling Dataset into Neural Field"
_ICLR.cc/2025/Conference — ICLR 2025 Poster_

### Official Review · Reviewer_XRWD · 2024-10-30

**Soundness:** 3
**Presentation:** 3
**Contribution:** 2
**Rating:** 5
**Confidence:** 3

**Summary:**

A neural field-based parameterisation learned to encode the synthetic dataset for the downstream model training termed Distill Dataset into Neural Field (DDiF), is proposed in this submission. Comparied with the line of research that directly stores the synthetic dataset, a few neural fields parameterised by the neural networks are learned to yield pixel values with feed-in coordinates in the inference time, following the principle from FreD. DDiF is empirically tested on dataset condensation benchmarks especially in the image per category 1 and 10 setting on ImageNet with noticeable improvements.

**Strengths:**

1. The idea is simple to follow and easy to implement.
2. The experiment results demonstrate the efficiency of neural field embedding but some concerns may be raised that I will point out later.
3. The writing is clear and easy to follow.

**Weaknesses:**

1. The theoretical analysis concludes that DDiF has lower synthetic data loss than Fred due to the former feasible space being larger in Theorem2. The main reasons are based on Observations 1 and 2. However, there is no clear clue about the source of the observation.

2. The budget comparison is not given. Since the DDiF architecture details are not clear, there is no direct comparison of the cost of saving the neural field and the synthetic dataset.

3. I'm not sure the algorithm can be generalised to large IPC settings since in most experiments, there are only 1 and 10 IPCs, even for Cifar10 and CIf100 in the appendix.

**Questions:**

See weakness.

---

> ### Author Response · Authors · 2024-11-22
> **Response to Reviewer XRWD [1/2]**
>
> We appreciate the constructive reviews and valuable comments. We address the concerns below.
>
> >**W1. [Numerical evidence on Observations 1 and 2]** *The theoretical analysis concludes that DDiF has lower synthetic data loss than Fred due to the former feasible space being larger in Theorem2. The main reasons are based on Observations 1 and 2. However, there is no clear clue about the source of the observation.*
>
> We feel sorry for not referring that the proofs of the observation 1 and 2 are in Appendix A. We will add it in the final manuscript.
> Additionally, we have included performance results by adjusting the feasible space of input-sized parameterization to investigate the empirical evidence behind the observation. The methods considered for adjusting the feasible space of input-sized parameterization are 1) dimension masking and 2) value clipping.
>
> As shown in the table below, when the feasible space becomes smaller (i.e., as the restrictions are enforced more strongly), the dataset distillation performances decrease. These results serve as direct evidence of observations in theoretical analysis.
>
> | Method                           | Accuracy (%)        |
> |----------------------------------|---------------------|
> | TM - No restriction              | 51.4 $\pm$ 2.3      |
> | TM - Masking 25% dimension       | 50.2 $\pm$ 1.7      |
> | TM - Masking 50% dimension       | 48.7 $\pm$ 3.2      |
> | TM - Masking 75% dimension       | 41.1 $\pm$ 1.9      |
>
> | Method                           | Accuracy (%)        |
> |----------------------------------|---------------------|
> | TM - No restriction              | 51.4 $\pm$ 2.3      |
> | TM - Clipping into [-2.0, 2.0]  | 43.2 $\pm$ 0.9      |
> | TM - Clipping into [-1.0, 1.0]  | 25.8 $\pm$ 2.8      |
> | TM - Clipping into [-0.5, 0.5]  | 21.0 $\pm$ 1.3      |
>
> For the empirical evidence of Observation 2, please refer to Figure 5 in [1]. The experimental
> finding in [1] that an increase in frequency dimension leads to improved dataset
> distillation performance supports Observation 2.
>
> [1] Shin, Donghyeok, Seungjae Shin, and Il-Chul Moon. "Frequency domain-based dataset distillation." Advances in Neural Information Processing Systems 36 (2024).

---

> ### Author Response · Authors · 2024-11-22
> **Response to Reviewer XRWD [2/2]**
>
> >**W2. [Budget comparison]** *The budget comparison is not given. Since the DDiF architecture details are not clear, there is no direct comparison of the cost of saving the neural field and the synthetic dataset.*
>
> We would like to ask reviewer to read last paragraph of Section 3.2 and Appendix B.4. These sections already provide the information for budget calculation of DDiF. DDiF constructs a synthetic dataset using a combination of a coordinate set (code) and a synthetic neural field (decoding function). However, as mentioned in Section 3.2 in the main paper, the coordinate set can be easily obtained by determining the size of the decoded instance, so it does not require learning or storage. Therefore, DDiF only learns and stores the parameters of the synthetic neural field that correspond to the decoding function. This “decoder-only” framework structurally differs from conventional parameterization methods. In conclusion, the storage budget of DDiF is calculated based on the number of total parameters of the synthetic neural field, and the structure of the synthetic neural field (such as width and number of layers) is adjusted to ensure that its storage budget is smaller than that of input-sized parameterization methods.
>
> Specifically, DDiF uses a single synthetic neural field to generate one decoded instance, with each synthetic neural field utilizing an MLP with sine activation. Therefore, the budget required to generate a single instance is calculated as follows: $d_{0}(n+1)+\sum_{l=1}^{L-1} d_{l}(d_{l-1}+1)+m(d_{L-1}+1)$. Here, $n$ is the dimension of the coordinate space, $L$ is the number of layers, $d_l$ is the width of layer $l$, and $m$ is the dimension of the quantity. In the actual implementation, we used the same width for each layer i.e. $d_l=d$. For a fair comparison, we adjusted the width, layers, and decoded instances per class of the neural field so that “Total number of classes × Decoded instances per class × Budget per neural field” is smaller than the storage budget of input-sized parameterization methods (i.e., # of classes × instances per class (IPC) × size of a single instance). The specific values used in the experiment and their corresponding storage budget are outlined below. Please refer to Table 7 in Appendix for the specific configurations on other datasets.
>
> | Dataset            | IPC | Input dim (n) | Layers (L) | Width (d) | Output dim (m) | Budget per instance | Decoded instance per class | # of classes | Utilized storage budget | Allowed storage budget |
> |-|-|-|-|-|-|-|-|-|-|-|
> | CIFAR-10           | 1   | 2             | 2          | 6         | 3               | 81                  | 37                              | 10            | 29,970                  | 30,720                  |
> | ImageNet (128)     | 1   | 2             | 3          | 20        | 3               | 963                 | 51                              | 10            | 491,130                 | 491,520                 |
> | ImageNet (256)     | 1   | 2             | 3          | 40        | 3               | 3,523                | 55                              | 10            | 1,908,720                | 1,937,650                |
>
> ---
> >**W3. [Experimental Results on Large IPC]** *I'm not sure the algorithm can be generalised to large IPC settings since in most experiments, there are only 1 and 10 IPCs, even for Cifar10 and CIf100 in the appendix.*
>
> The table below shows test accuracies (%) on CIFAR-10, CIFAR-100, and ImageNet (128) under IPC=50. We utilize trajectory matching (TM) for dataset distillation loss. The table below presents the results as follows:
> 1)	As with IPC=1,10, DDiF achieves performance improvement on TM. It shows that DDiF also works well on large IPC setting.
> 2)	DDiF under IPC=50 achieves higher performance than DDiF under IPC=1,10. It shows that there is no performance degradation as budget increase.
> 3)	DDiF achieves highly competitive performances (Best or Second best) with previous studies.
>
> | IPC=50| CIFAR-10 | CIFAR-100 | ImageNette |
> |---|----------|-----------|------------|
> | TM  | 71.6 $\pm$ 0.2     | 47.7 $\pm$ 0.2      | 72.8 $\pm$ 0.8 |
> | FRePo| 71.7 $\pm$ 0.2     | 44.3 $\pm$ 0.2      | -          |
> | IDC  | 74.5 $\pm$ 0.2    | -         | -          |
> | FreD | 75.8 $\pm$ 0.1    | 47.8 $\pm$ 0.1      | -          |
> | HaBa | 74.0 $\pm$ 0.2     | 47.0 $\pm$ 0.2      | -          |
> | RTP  | 73.6 $\pm$ 0.5     | -         | -          |
> | HMN  | 76.9 $\pm$ 0.2     | 48.5 $\pm$ 0.2      | -          |
> | SPEED| **77.7** $\pm$ 0.4     | *49.1* $\pm$ 0.2      | -          |
> | NSD  | 75.2 $\pm$ 0.6     | -         | -          |
> | DDiF | *77.5* $\pm$ 0.3     | **49.9** $\pm$ 0.2      | **75.2 $\pm$ 1.3**|

---

> > ### Comment · Reviewer_XRWD · 2024-11-27
> >
> > I appreciate the authors' efforts in addressing the concerns through their rebuttal. As stated by the authors, a single synthetic neural field, parameterized by an MLP with sine activation, generates one decoded instance per class. However, I remain concerned about the budget. While the authors demonstrate that in the 1 IPC setting, the neural network sizes are only marginally smaller than those required for storing images, Table 7 clearly shows that in the 50 IPC setting, the neural network sizes are significantly larger, which could result in higher storage costs compared to saving images.
> >
> > Moreover, the number of decoded instances per class raises additional concerns. One of the primary objectives of dataset condensation is to enhance training efficiency by reducing the number of training samples. However, DDiF appears to require more instances as the dataset complexity increases, potentially undermining this goal.
> >
> > Some SOTA methods are not discussed and compared in 50 IPC setting
> >
> > [1] Sun, P., Shi, B., Yu, D. and Lin, T., On the diversity and realism of distilled dataset: An efficient dataset distillation paradigm. In Proceedings of the IEEE/CVF Conference on Computer Vision and Pattern Recognition, 2024.
> >
> > [2] Guo Z, Wang K, Cazenavette G, LI H, Zhang K, You Y. Towards Lossless Dataset Distillation via Difficulty-Aligned Trajectory Matching. In The Twelfth International Conference on Learning Representations, 2024.

---

> > > ### Author Response · Authors · 2024-11-30
> > > **Additional response to Reviewer XRWD [1/3]**
> > >
> > > We appreciate the opportunity to relieve the reviewer's concerns and provide further explanation. We address the uncleared concerns below.
> > >
> > > >**Q1 [Budget calculation in the IPC=50 setting]** *As stated by the authors, a single synthetic neural field, parameterized by an MLP with sine activation, generates one decoded instance per class. However, I remain concerned about the budget. While the authors demonstrate that in the 1 IPC setting, the neural network sizes are only marginally smaller than those required for storing images, Table 7 clearly shows that in the 50 IPC setting, the neural network sizes are significantly larger, which could result in higher storage costs compared to saving images.*
> > >
> > > The table below presents a detailed budget comparison between previous studies and DDiF. We emphasize the following results:
> > > * When IPC=1 case, the budget per instance of DDiF, which is the number of single neural field parameters, is significantly smaller than the budget per instance of Vanilla. Specifically, DDiF only utilizes a budget per instance equivalent to 2.64% and 1.96% of Vanilla’s in CIFAR-10 and ImageNet-Subset (128), respectively.
> > > * When IPC=50 case, the budget per instance of DDiF is larger, but it is still significantly smaller than the budget per instance of Vanilla: 17.68% in CIFAR-10 and 7.17% in ImageNet-Subset (128).
> > > * When the budget per instance increases, the increment of decoded instances decreases inversely. As mentioned in the previous comment, this adjustment ensures a fair comparison by satisfying the utilized budget of DDiF smaller than that of input-sized parameterization. In the formula, we set the decoded instance per class and the budget per instance (the structure of neural field) such that "Decoded instances per class $\times$ Budget per instance $\leq$ IPC $\times$ Size of original instance". (We omit the number of classes which appears equally on both sides)
> > >
> > > |Dataset| IPC                   | Method | Budget per instance | Ratio of Budget per instance | Increment of decoded instances | Performance        |
> > > |-------|-----------------------|--------|---------------------|-----------------------------|--------------------------------|--------------------|
> > > | CIFAR-10      | 1      | Vanilla             | 3,072       | 100%   | $\times$1   | 46.3 $\pm$ 0.8        |
> > > |       |        | FreD      | 192          | 6.25%   | $\times$16                | 60.6 $\pm$ 0.8        |
> > > |       |        | DDiF     | 81       | 2.64%       | $\times$37                | 66.5 $\pm$ 0.4        |
> > > |       | 50     | Vanilla             | 3,072        | 100%       | $\times$1                 | 71.6 $\pm$ 0.2        |
> > > |      |        | FreD                | 768          | 25%      | $\times$4                 | 75.8 $\pm$ 0.1        |
> > > |      |        | DDiF                | 543        | 17.68%   | $\times$5.64              | 77.5 $\pm$ 0.3        |
> > > | ImageNet-Subset (128) | 1      | Vanilla     | 49,152      | 100%    | $\times$1    | 51.4 $\pm$ 2.3        |
> > > |        |        | FreD                | 6,144        | 12.5%            | $\times$8        | 66.8 $\pm$ 0.4        |
> > > |         |        | DDiF                | 963         | 1.96%    | $\times$51                | 72.0 $\pm$ 0.9        |
> > > |         | 50     | Vanilla             | 49,152           | 100%        | $\times$1       | 72.8 $\pm$ 0.8        |
> > > |         |        | DDiF       | 3,523        | 7.17%              | $\times$13.94       | 75.2 $\pm$ 1.3        |
> > >
> > > In a nutshell, DDiF utilizes a significantly smaller budget per instance than Vanilla in both IPC=1 and 50. Also, as IPC increases, we utilize a larger neural field structure. However, the number of decoded instances was reduced to ensure that the storage budget constraint. Therefore, even in IPC=50, DDiF did not utilize a higher storage budget than input-sized parameterization.
> > >
> > > The reason for using larger neural fields as IPC increases is that, as the available budget grows, improving the quality of decoded instances contributes more significantly to performance enhancement than increasing their quantity. When IPC increases, sufficient diversity is already achieved, so enhancing the information content of individual instances becomes essential. This tendency has been studied in [1] (Figure 9 in [1]).
> > >
> > > [1] Shin, Donghyeok, Seungjae Shin, and Il-Chul Moon. "Frequency domain-based dataset distillation." Advances in Neural Information Processing Systems 36 (2024).

---

> > > ### Author Response · Authors · 2024-11-30
> > > **Additional response to Reviewer XRWD [2/3]**
> > >
> > > >**Q2 [Discussion about the number of decoded instances per class]** *Moreover, the number of decoded instances per class raises additional concerns. One of the primary objectives of dataset condensation is to enhance training efficiency by reducing the number of training samples. However, DDiF appears to require more instances as the dataset complexity increases, potentially undermining this goal.*
> > >
> > > We believe that one of the primary objectives of parameterization methods in dataset distillation is to enhance the budget efficiency under the limited budget constraint. Therefore, parameterization methods have evolved to reduce budget redundancy [1,2,3] or to generate decoded synthetic instances by combining shared information in diverse ways [4,5]. As a result, the budget required to generate a single instance has decreased. Since the degree of budget reduction varies across different methods, parameterization methods have utilized IPC as a measure of storage budget for fair comparison. Consequently, in general, parameterization methods use more decoded instances than IPC to satisfy the budget constraint defined by IPC. For instance, FreD [1] uses 8 decoded instances with IPC=1 under ImageNet-Subset (128) while SPEED [3] uses 15 decoded instances under the same setting. Since DDiF has superior coding efficiency, DDiF utilizes notably small budget per instance (1.96% of the original image size with IPC=1 under ImageNet-Subset) and decode large synthetic instances under the fixed storage budget (51 decoded instances per class with IPC=1 under ImageNet-Subset). The reviewer pointed out that DDiF needs more decoded instances for high-dimensional datasets, which may reduce the training efficiency. However, we emphasize that the large number of decoded instances in DDiF results from a fair comparison with previous parameterization methods under the same storage budget.
> > >
> > > To demonstrates that DDiF’s superiority is not solely due to the increase in decoded instances, **we conducted the performance comparisons under the fixed number of decoded instances setting, which are already documented in Appendix C.3**. Since the number of decoded instances is fixed, the training efficiency in test stage is same, and so we can focus even more on the comparison of budget efficiency and expressiveness. We would like to ask reviewer to refer to Appendix C.3 or response comments to other reviewers: *W1(b) of Reviewer Rr6g*, *Q3 of Reviewer Y1Uh*. In short, under the fixed number of decoded instances, DDiF achieves mixed performance results with Vanilla and higher performance than previous parameterization methods, while using less budget. Since we experimented with high-dimensional datasets, these experimental results demonstrate that DDiF has benefits in storage budget and performance without undermining the goal of dataset distillation on complex datasets.
> > >
> > > [2] Kim, Jang-Hyun, et al. "Dataset condensation via efficient synthetic-data parameterization." International Conference on Machine Learning. PMLR, 2022.
> > >
> > > [3] Wei, Xing, et al. "Sparse parameterization for epitomic dataset distillation." Advances in Neural Information Processing Systems 36 (2024).
> > >
> > > [4] Liu, Songhua, et al. "Dataset distillation via factorization." Advances in neural information processing systems 35 (2022): 1100-1113.
> > >
> > > [5] Deng, Zhiwei, and Olga Russakovsky. "Remember the past: Distilling datasets into addressable memories for neural networks." Advances in Neural Information Processing Systems 35 (2022): 34391-34404.

---

> > > ### Author Response · Authors · 2024-11-30
> > > **Additional response to Reviewer XRWD [3/3]**
> > >
> > > >**Q3 [Comparison with SOTA methods in IPC=50 setting]** *Some SOTA methods are not discussed and compared in 50 IPC setting [1] Sun, P., Shi, B., Yu, D. and Lin, T., On the diversity and realism of distilled dataset: An efficient dataset distillation paradigm. In Proceedings of the IEEE/CVF Conference on Computer Vision and Pattern Recognition, 2024. [2] Guo Z, Wang K, Cazenavette G, LI H, Zhang K, You Y. Towards Lossless Dataset Distillation via Difficulty-Aligned Trajectory Matching. In The Twelfth International Conference on Learning Representations, 2024.*
> > >
> > > Thank the reviewer for the meticulous suggestion regarding the comparison with SOTA methods in IPC=50 setting. We firstly compared with previous parameterization methods since DDiF is a parameterization method. However, we agree that a comparison with other high-performance methods is necessary in IPC=50 setting, which was the one of the reviewer’s concerns. The table below shows the test accuracies (%) of RDED [6] and DATM [7], which are the reviewer’s suggestion, on CIFAR-10, CIFAR-100, and ImageNette (128). In the table, DDiF achieves significantly higher performance than baselines in small IPC setting. However, as noted by the reviewer, DDiF shows lower performance in large IPC setting.
> > >
> > > | ConvNet  | | CIFAR-10  | | | CIFAR-100 | | | ImageNette (128) | |
> > > |----------| :---: | :-----: | :--: | :-: | :----: | :--: | :--: | :------: | :---: |
> > > |          | IPC=1     | 10    | 50    | 1     | 10    | 50    | 1     | 10    | 50    |
> > > | RDED [6] | 23.5  | 50.2  | 68.4  | 19.6  | **48.1**  | **57.0**  | 33.8  | 63.2  | **83.8**  |
> > > | DATM [7] | 46.9  | 66.8  | 76.1  | 27.9  | 47.2  | 55.0  | -     | -     | -     |
> > > | DDiF     | **66.5**  | **74.0**  | **77.5**  | **42.1**  | 46.0  | 49.9  | **72.0**  | **74.6**  | 75.2  |
> > >
> > > **We believe that this performance gap can be attributed to the utilization of different synthetic labels.** Various previous studies have demonstrated that using soft labels instead of one-hot labels leads to significant performance improvements in dataset distillation [8,9]. Additionally, we confirmed from the RDED and DATM papers that soft labels significantly contributed to the high performance of each method (Table 8 in [6] / Table 2(b) in [7]).
> > >
> > > As mentioned in Line 266-268 of our manuscript, DDiF can also utilize a soft label technique, but we used a fixed one-hot label for a fair comparison with the previous parameterization methods. Therefore, to validate our conjecture, we conducted additional experiments on two scenarios: 1) applying fixed one-hot labels to RDED and DATM, and 2) applying soft labels to DDiF. To apply soft labels to DDiF, we utilized the region-level soft label technique from RDED on the decoded synthetic instances, which are already obtained through training with TM. The results are summarized in the table below:
> > > * Under the fixed one-hot label setting, RDED and DATM exhibit large performance degradation and DDiF emerges as the best or second-best performer.
> > > * Applying RDED's soft label technique to DDiF results in notable performance improvement, consistent with previous studies.
> > > * Under the soft label setting, DDiF achieves higher performance with significant gap than RDED and DATM, establishing a new state-of-the-art (SOTA).
> > >
> > > In summary, we prioritized experiments under the fixed one-hot label setting to ensure a fair comparison with existing parameterization methods. However, our experiments confirmed that DDiF can effectively utilize soft label technique, achieving substantial performance improvements when applied. We will add these findings into the final manuscript.
> > >
> > > | ConvNet & IPC=50 | Method  | CIFAR-100       | ImageNette (128)  |
> > > |--------|-----| :-------: | :---------: |
> > > | Fixed one-hot label | RDED    | 33.5 $\pm$ 0.2      | 57.8 $\pm$ 1.2        |
> > > |   | DATM       | **50.8**    | -    |
> > > |    | DDiF    | 49.9 $\pm$ 0.2      | **75.2** $\pm$ 1.3        |
> > > | Soft label       | RDED (Reported) | 57.0 $\pm$ 0.1      | 83.8 $\pm$ 0.2        |
> > > |    | RDED (Reproduced) | 57.5 $\pm$ 0.2      | 83.4 $\pm$ 0.4        |
> > > |    | DATM     | 55.0 $\pm$ 0.2      | -    |
> > > |   | DDiF + RDED’s Soft label technique      | **58.2** $\pm$ 0.2      | **86.9** $\pm$ 1.0   |
> > >
> > > [6] Sun, Peng, et al. "On the diversity and realism of distilled dataset: An efficient dataset distillation paradigm." Proceedings of the IEEE/CVF Conference on Computer Vision and Pattern Recognition. 2024.
> > >
> > > [7] Guo, Ziyao, et al. "Towards Lossless Dataset Distillation via Difficulty-Aligned Trajectory Matching." The Twelfth International Conference on Learning Representations.
> > >
> > > [8] Cui, Justin, et al. "Scaling up dataset distillation to imagenet-1k with constant memory." International Conference on Machine Learning. PMLR, 2023.
> > >
> > > [9] Qin, Tian, Zhiwei Deng, and David Alvarez-Melis. "A Label is Worth a Thousand Images in Dataset Distillation." arXiv preprint arXiv:2406.10485 (2024).

---

> ### Author Response · Authors · 2024-12-02
> **Reminder for Reviewer XRWD**
>
> Dear Reviewer XRWD,
>
> We sincerely appreciate your time and effort in reviewing our manuscript.
>
> We fully understand you are busy, but we would like to know if our responses have adequately addressed your concerns.
>
> As the discussion period is ending, we kindly ask the reviewer to consider our responses, where we have tried to incorporate your feedback as thoroughly as possible.
>
> Please do not hesitate to let us know if you have any additional concerns or questions.
>
> We are always open to further discussion about our research at any time.
>
> Again, we appreciate the reviewer's constructive feedback.
>
> Best Regards,

---

### Official Review · Reviewer_e7GQ · 2024-10-31

**Soundness:** 3
**Presentation:** 2
**Contribution:** 3
**Rating:** 6
**Confidence:** 4

**Summary:**

This paper proposes to use a neural field framework to encapsulate and learn a dataset, i.e. dataset distillation. The neural field is prompted by coordinates to generate images, and can produce at different resolutions. The experiments show very promising results.

**Strengths:**

+ This paper proposes a very interesting extension of parameterization for dataset distillation, and uses a neural field to compress information.
+ The experimental results show very promising performance on challenging datasets
+ The generalization on different resolution capability is intriguing

**Weaknesses:**

- The paper's theme is on parameterization of image datasets, yet the introduction section rarely talks about the relevant topic and works to distinguish this paper out. I think the authors should specifically discuss and compare with [1, 2, 3, others] principal-wise to help the reader understand how the progress is made in DD parameterization and the delta in this paper.

- In the experiment section, I'd encourage the authors to add a table row showing the number of images (not ipc budget) used for re-training. Reparameterization methods tend to have more images, which I think is fine, but needs to be explicitly mentioned for clarity.

- One critical point in the experiment: is neural field better than simple basis decomposition used in RTP[1]? In the previous literature, the bases-coefficient decomposition seems to outperform HaBa quite a lot on simpler datasets. It should not be too hard to use RTP's decomposition method (bases + coefficient) and a DD loss that this paper is using (e.g., trajectory matching loss), and run experiments on the ImageNet benchmarks to perform a direct comparison. This will provide more support that neural fields can work very well and outperform the previous SOTA parameterization.

- How heavy is the data augmentation?

[1] Remember the Past: Distilling Datasets into Addressable Memories for Neural Networks

[2] Dataset Distillation via Factorization

[3] Farzi Data: Autoregressive Data Distillation

**Questions:**

See above. I think the paper is very interesting. If the authors addresses my questions above, I'd like to consider further recommendation of this paper.

---

> ### Author Response · Authors · 2024-11-22
> **Response to Reviewer e7GQ [1/2]**
>
> We appreciate the constructive reviews and valuable comments. We address the concerns below.
>
> >**W1. [More discussion about Parameterization methods]** *The paper's theme is on parameterization of image datasets, yet the introduction section rarely talks about the relevant topic and works to distinguish this paper out. I think the authors should specifically discuss and compare with [1, 2, 3, others] principal-wise to help the reader understand how the progress is made in DD parameterization and the delta in this paper. [1] Remember the Past: Distilling Datasets into Addressable Memories for Neural Networks [2] Dataset Distillation via Factorization [3] Farzi Data: Autoregressive Data Distillation*
>
> Thank the reviewer for their kind recommendation. We discuss the parameterization methods in the second paragraph of the Introduction (Lines 39-49), and a more detailed explanation in Section 2.2. Additionally, [2] is already cited in Line 44. In the final manuscript, we will add [1] and [3] in both the Introduction and Section 2.2 to provide a more comprehensive comparison. This will help clarify the relationship between our work and prior approaches, highlighting the contributions of this paper in advancing the dataset distillation research community.
>
> ---
>
> >**W2. [Explanation of Number of decoded synthetic instances]** *In the experiment section, I'd encourage the authors to add a table row showing the number of images (not ipc budget) used for re-training. Reparameterization methods tend to have more images, which I think is fine, but needs to be explicitly mentioned for clarity.*
>
> Under the fixed storage budget, the number of decoded synthetic instances are determined by the budget of single instance. We discussed how the storage budget for DDiF is calculated in the main paper (Lines 278-287), and the specific values used in the experiments are provided in Table 7 in Appendix. For a fair comparison, we configured the number of decoded synthetic instances such that the storage budget of DDiF has a smaller value than the storage budget of input-sized parameterization.
>
> As suggested by the reviewer, we have newly included the budget of a single instance and the increment of decoded instances in Table 7 in Appendix. For the reviewer’s convenience, we report several configurations at the table below:
>
> | Dataset            | IPC | Input dim (n) | Layers (L) | Width (d) | Output dim (m) | Budget per instance | Decoded instance per class | # of classes | Utilized storage budget | Allowed storage budget |
> |-|-|-|-|-|-|-|-|-|-|-|
> | CIFAR-10           | 1   | 2             | 2          | 6         | 3               | 81                  | 37                              | 10            | 29,970                  | 30,720                  |
> | ImageNet (128)     | 1   | 2             | 3          | 20        | 3               | 963                 | 51                              | 10            | 491,130                 | 491,520                 |
> | ImageNet (256)     | 1   | 2             | 3          | 40        | 3               | 3,523                | 55                              | 10            | 1,908,720                | 1,937,650                |
>
> We believe that reporting both the neural field configuration and the number of decoded instances together would facilitate a better understanding. However, due to space constraints in the main paper, we have included this information in the Appendix. Additionally, we kindly ask the reviewer to refer to Appendix C.3 for a more comprehensive understanding of how DDiF’s superiority arises not only from quantity improvements but also from quality enhancements.
>
> [1] Deng, Zhiwei, and Olga Russakovsky. "Remember the past: Distilling datasets into addressable memories for neural networks." Advances in Neural Information Processing Systems 35 (2022): 34391-34404.
>
> [2] Liu, Songhua, et al. "Dataset distillation via factorization." Advances in neural information processing systems 35 (2022): 1100-1113.
>
> [3] Sachdeva, Noveen, et al. "Farzi Data: Autoregressive Data Distillation." arXiv preprint arXiv:2310.09983 (2023).

---

> ### Author Response · Authors · 2024-11-22
> **Response to Reviewer e7GQ [2/2]**
>
> >**W3. [Comparison with RTP]** *One critical point in the experiment: is neural field better than simple basis decomposition used in RTP[1]? In the previous literature, the bases-coefficient decomposition seems to outperform HaBa quite a lot on simpler datasets. It should not be too hard to use RTP's decomposition method (bases + coefficient) and a DD loss that this paper is using (e.g., trajectory matching loss), and run experiments on the ImageNet benchmarks to perform a direct comparison. This will provide more support that neural fields can work very well and outperform the previous SOTA parameterization.*
>
> We want to point out that there are performance comparisons between RTP and DDiF in Appendix C, Table 8. For the reviewer’s convenience, we report the performance comparison again. As seen in the table, DDiF demonstrates better performance than RTP.
>
> | | |CIFAR-10       |       | |CIFAR-100  |          |
> |-|-|-|-|-|-|-|
> |IPC| 1      | 10     | 50     | 1      | 10     | 50     |
> | RTP (Official report) | 66.4 $\pm$ 0.4 | 71.2 $\pm$ 0.4 | 73.6 $\pm$ 0.5 | 34.0 $\pm$ 0.4 | 42.9 $\pm$ 0.7 | - |
> | DDiF             | **66.5** $\pm$ 0.4 | **74.0** $\pm$ 0.4 | **77.5** $\pm$ 0.3 | **42.1** $\pm$ 0.2 | **46.0** $\pm$ 0.2 | **49.9** $\pm$ 0.2 |
>
> Furthermore, as a reviewer’s suggestion, we additionally conducted the experiment on ImageNet datasets under the same dataset distillation loss (TM). The table below presents the results as follows:
> * RTP shows the performance improvement when it is applied to TM.
> * DDiF achieves higher performance improvement than RTP even though DDiF shows smaller increments of decoded instances.
>
> These experimental results support that DDiF shows high performance. Also, combining the neural field and decomposition idea could be an interesting research direction for future work.
>
> | Dataset             | Increment of decoded instances           | ImageNette (128)  | ImageWoof (128)   | ImageFruit (128)  |
> |---------------------|--------------------|-------------------|-------------------|-------------------|
> | TM                  | $\times$1     | 51.4 $\pm$ 2.3    | 29.7 $\pm$ 0.9    | 28.8 $\pm$ 1.2    |
> | TM + RTP            | $\times$64     | 69.6 $\pm$ 0.4    | 38.8 $\pm$ 1.1    | 45.2 $\pm$ 1.7    |
> | TM + DDiF           | $\times$51     | **72.0** $\pm$ 0.9    | **42.9** $\pm$ 0.7    | **48.2** $\pm$ 1.2    |
>
> ---
>
> >**W4. [Cost of Data augmentation]** *How heavy is the data augmentation?*
>
> Following the previous works, we used the same data augmentation as in DSA [4]. Specifically, we utilize color jittering, cropping, cutout, flipping, scaling, and rotation. Since most baselines also use this data augmentation, there is no additional cost incurred by DDiF. We will add this setting in implementation settings (Appendix B.4).
>
> [4] Zhao, Bo, and Hakan Bilen. "Dataset condensation with differentiable siamese augmentation." International Conference on Machine Learning. PMLR, 2021.

---

> > ### Comment · Reviewer_e7GQ · 2024-11-27
> > **Thank you**
> >
> > Thanks for the authors' response. These results and explanations are helpful. Please also consider moving and adding a row for RTP + TM in your table 2, it'll be a strong baseline to compare with and it doesn't make sense to stay in the appendix.

---

> > > ### Author Response · Authors · 2024-11-30
> > > **Thanks for reviewer e7GQ**
> > >
> > > Thank you for taking the time to read our response, additional experimental results, and the updated manuscript.
> > >
> > > We agree with the reviewer’s suggestion to include TM+RTP in Table 2.
> > >
> > > However, as we are currently unable to upload a revised PDF, we promise to incorporate this modification into the final manuscript.
> > >
> > > Please leave additional comments if you have any concerns or further questions.
> > >
> > > We would be happy to provide additional clarification.
> > >
> > > Again, Thank you so much for your time and effort.

---

### Official Review · Reviewer_Y1Uh · 2024-11-03

**Soundness:** 3
**Presentation:** 3
**Contribution:** 2
**Rating:** 5
**Confidence:** 4

**Summary:**

- This paper proposed using Neural Fields to parameterize distilled datasets (DDiF)
- The basic idea is to replace each distilled image by a neural field which implicitly defines the image (by evaluating the field at grid points)
- The method gets performance gains by reducing the parameter count per images (provided each neural field is small enough), so you can you use more images
- The method is robust to different dataset distillation loss choices

**Strengths:**

- The method is straightforward and is easily compatible with existing DD algorithms.
- Good experimental results, particularly on non-image domains (with high dimensionality, where this method would have the greatest wins)
- Interesting results with cross-resolution generalization, which is a new idea in dataset distillation (although I have questions about the evaluation - see the questions section)

**Weaknesses:**

- I don't think the theoretical analysis is particularly enlightening. The theoretical analysis can be boiled down to "if you have more degrees of freedom, then you can get lower distillation loss, and DDiF has more degrees of freedom than FreD since DDiF can modify the frequencies as well as their coefficients, whereas FreD can only modify the coefficients". However, this theoretical analysis is rather meaningless as adding the ability to control the frequencies (and the biases) adds more parameters. This analysis is only useful if you can fix the number of free parameters between FreD and DDiF. I understand this is analysis is difficult and can really only be verified experimentally (which the authors have done), but this caveat with the theoretical analysis needs to be mentioned.
- There is no mention about the additional cost of evaluating the neural field at the coordinate points. The additional cost of evaluating the field at all datapoints could very significant, especially for high-resolution/higher dimensional data, but is not discussed in the paper. It would be useful to compare the additional runtime requirements, compared to directly parameterizing the images/data themselves, particularly for the high-resolution data.

**Questions:**

- See weaknesses
- For the resolution generalization experiments, can you additionally report the performance of downsampling the full dataset down to the reduced distilled dataset resolution (as opposed to upsampling the distilled datasets, as you have presented). If there is better performance by downsampling the original data, then there would not be a convincing argument why one should upsample the distilled dataset
- Is it not possible to upsample FreD distilled images by just evaluating at more points (like with equation 6)? Did the authors use this method for the upsampling experiments? If not, then this would be very important to make the comparison fair.
- Can you also report for the performance of vanilla dataset distillation performance, with the same number of distilled images in table 1,2,3? If I understand correctly, right now you only present the performance matching the number of parameters, but it would a useful datapoint to consider what performance hit is given by parameterizing with DDiF as opposed to directly parameterizing the images
- Is it possible to produce a curve showing the effect of neural field architecture choices on the performance? For example, given a fixed parameter budget, how does the width of the neural field (for example), affect the distillation performance?

---

> ### Author Response · Authors · 2024-11-22
> **Response to Reviewer Y1Uh [1/5]**
>
> We appreciate the constructive reviews and valuable comments. We address the concerns below.
>
> >**W1. [Discussion about Theoretical Analysis]** *I don't think the theoretical analysis is particularly enlightening. The theoretical analysis can be boiled down to "if you have more degrees of freedom, then you can get lower distillation loss, and DDiF has more degrees of freedom than FreD since DDiF can modify the frequencies as well as their coefficients, whereas FreD can only modify the coefficients". However, this theoretical analysis is rather meaningless as adding the ability to control the frequencies (and the biases) adds more parameters. This analysis is only useful if you can fix the number of free parameters between FreD and DDiF. I understand this is analysis is difficult and can really only be verified experimentally (which the authors have done), but this caveat with the theoretical analysis needs to be mentioned.*
>
> The primary goal of the theoretical analysis is to propose a framework for comparing parameterization methods, which have traditionally been evaluated solely based on performance, through expressiveness—specifically, the size of the feasible space. While Observations 1 and 2 are straightforward and intuitive, no previous work has explicitly articulated or proven these concepts. Furthermore, although numerous parameterization methods have been proposed, there has been no prior research that theoretically compares these methods to justify their validity. In this paper, we addressed this gap by comparing the expressiveness of DDiF with that of prior work (FreD) in Theorem 2. We believe that the proposed theoretical analysis framework will serve as a cornerstone for future theoretical comparisons of parameterization methods in dataset distillation area.
>
> As the reviewer mentioned, DDiF has higher expressiveness than FreD since DDiF can control the coefficient, frequency, phase, and shift, while FreD can only control the coefficient. The reviewer pointed out that adding additional parameters to FreD to enhance its control ability could improve its expressiveness. However, it could be essential to clarify what constitutes meaningful parameters and how they should be added. Here, it is important to note that not all parameters are meaningful, and there is a provable necessity of adding a specific parameter, which is delivered through Theorem 1 and Remark 1. Finally, more flexibility does not always conclude better dataset distillation performance, so Theorem 2 and Observation 2 provide theoretic ground for such justification of flexibility from the distillation perspective. We believe that the proposed theoretical analysis supports these considerations through Fourier view. Furthermore, we want to highlight that DDiF is one of the methods which can implicitly control these meaningful parameters as shown in theoretical analysis.
>
> The reviewer also pointed out that this analysis is only valuable when the number of free parameters (i.e. budget) is same. However, we emphasize that there are cases in Figure 5 where DDiF achieves higher performance than FreD, despite having less free parameters. Moreover, we further compare the performance on the ImageNette dataset when the number of decoded synthetic instances is set to 8 (FreD can decode 8 synthetic instances per class at a resolution of 128). The table below shows that DDiF exhibits higher performance than FreD, even while utilizing a smaller budget
>
> | Method | Decoded instances per class | Utilized budget | Accuracy (%)
> |-|-|-|-|
> | FreD  | 8         | 491,520         | 66.8 $\pm$ 0.4 |
> | DDiF  | 8         | 77,040         | 67.1 $\pm$ 0.3 |
>
> In addition, we kindly ask the reviewer to refer to Appendix C.3, which provides a further comparison with other parameterization methods. These experimental results serve as empirical evidence to defend the reviewer’s point. Even though the proposed theoretical analysis in this paper is experimentally verified through extensive results, this theoretical analysis still has room for further improvement since we consider the fixed number of decoded instance scenarios, not the fixed storage budget. We believe that constructing a theoretical framework to compare the expressiveness of parameterization methods under a fixed storage budget is necessary, and the proposed theoretical analysis in this paper can serve as a foundational background for such efforts. We added this discussion and caveat to revised paper in Appendix D.2.

---

> ### Author Response · Authors · 2024-11-22
> **Response to Reviewer Y1Uh [2/5]**
>
> >**W2. [Additional Cost of Decoding Process]** *There is no mention about the additional cost of evaluating the neural field at the coordinate points. The additional cost of evaluating the field at all datapoints could very significant, especially for high-resolution/higher dimensional data, but is not discussed in the paper. It would be useful to compare the additional runtime requirements, compared to directly parameterizing the images/data themselves, particularly for the high-resolution data.*
>
> We acknowledge the reviewer’s concern regarding the additional cost of evaluating the neural field at the coordinate points, particularly for high-dimensional data. It is true that the time cost of DDiF increases as resolution grows. This is primarily due to the need to forward a larger number of coordinates through the neural field, as well as the requirement to increase the size of the neural field itself to accommodate higher resolutions.
>
> To address this concern, we conducted additional experiments measuring the wall-clock time for decoding a single instance. The results with 128 image resolution demonstrate that while the time cost of DDiF is larger than methods relying on non-parameterized decoding functions, such as IDC and FreD, it remains comparable to methods that use parameterized decoding functions, such as HaBa and SPEED, and exhibits a lower time cost than methods that depend on pretrained generative models, such as GLaD. At a 256 resolution, we observe that the computation cost of DDiF slightly larger than non-parameterized decoding functions. In conclusion, although the decoding process time of DDiF increases as the resolution increases, it does not differ significantly from that of conventional parameterization methods. We attribute this to 1) the use of a small neural network structure for the synthetic neural field and 2) the full-batch forwarding of the coordinate set in the implementation.
>
> | ImageNet | $128\times128$ | $256\times256$ |
> |-|-|-|
> | Vanilla  | 0.31         | 0.31         |
> | IDC      | 0.40         | 2.83         |
> | FreD     | 0.46         | 1.20         |
> | HaBa     | 2.81         | -              |
> | SPEED    | 2.20         | -              |
> | GLaD     | 31.33        | -              |
> | DDiF     | 2.49         | 3.25         |

---

> ### Author Response · Authors · 2024-11-22
> **Response to Reviewer Y1Uh [3/5]**
>
> >**Q1. [Comparison with Full Dataset Downsampling on Cross-resolution Generalization]** *For the resolution generalization experiments, can you additionally report the performance of downsampling the full dataset down to the reduced distilled dataset resolution (as opposed to upsampling the distilled datasets, as you have presented). If there is better performance by downsampling the original data, then there would not be a convincing argument why one should upsample the distilled dataset.*
>
> We interpreted what the reviewer suggested as follows:
> 1)	Assume the full dataset consists of $N$ images at $256\times256$ resolution, resulting in a budget of $N\times 3\times 256\times 256$.
> 2)	Let the resolution for the downsampled images as $R$, and the budget becomes $N\times 3\times R\times R$.
> 3)	Note that the budget of synthetic dataset is $10\times 3\times 128\times 128$. To match the budget, $3<R< 4$. For experiment, we set $R=4$.
>
> The table below shows the experiment results where the full dataset was downsampled and then upsampled to the original resolution during the test phase. When the budget is similar ($R=4$), DDiF outperforms the downsampled dataset clearly.
>
> | Test network | Method | Original resolution | Downsampled resolution | Relative budget ratio | Accuracy         |
> |-|-|-|-|-|-|
> | ConvNetD5 | Downsample | 256 | 4 | 1.3 | 45.2 $\pm$ 2.3 |
> | | | 256 | 128 | 1289.4 | 91.3 $\pm$ 0.5 |
> | | DDiF | 128 | - | 1.0 | 66.3 $\pm$ 1.9 |
> | ConvNetD6 | Downsample | 256 | 4 | 1.3 | 44.7 $\pm$ 0.4 |
> | | | 256 | 128 | 1289.4 | 91.2 $\pm$ 0.0 |
> | | DDiF | 128 | - | 1.0 | 70.6 $\pm$ 1.2 |
>
> We also conducted the experiment when the downsized resolution $R$ is matched to the synthetic data resolution itself. In this case, the downsampled dataset achieved better performance, as expected. However, this setting requires 1,289 times more budget, which does not align with the dataset distillation framework. Such a setup deviates from the core purpose of dataset distillation, which aims to optimize performance under strict budget constraints.
>
> We want to discuss that there are two drawbacks of the reviewer’s proposed method.
> * The assumptions differ. We newly introduced the cross-resolution generalization to evaluate the dataset distillation ability to generalize to higher resolution settings. This involves training on a low-resolution (128×128) synthetic dataset and testing on high-resolution (256×256) data. The reviewer’s suggestion diverges from this setting since it assumes the availability of a high-resolution dataset (256×256).
> * The training time during the test phase increases since the number of data points remains the same as the full dataset. While the memory budget may be comparable, this increased time cost is undesirable, especially in scenarios where efficiency is critical.
> These differences fundamentally change the experiment's assumptions and objectives, making direct comparisons challenging.

---

> ### Author Response · Authors · 2024-11-22
> **Response to Reviewer Y1Uh [4/5]**
>
> >**Q2. [Comparison with Frequency upsampling on FreD]** *Is it not possible to upsample FreD distilled images by just evaluating at more points (like with equation 6)? Did the authors use this method for the upsampling experiments? If not, then this would be very important to make the comparison fair.*
>
> Thank the reviewer for pointing out. Currently, we upsampled synthetic instances from FreD using spatial-domain upsampling. However, as the reviewer suggested, FreD can also increase resolution by applying frequency-domain upsampling, as it stores masked frequency coefficients.
>
> The most widely used method for frequency-domain upsampling involves zero-padding the frequency coefficients, assigning zeros to the high-frequency components [1]. For example, using FreD’s default setting of DCT, the process of upsampling an $N$-resolution frequency coefficient $F$ to an $M(>N)$-resolution image can be described as $\text{IDCT} \left( \begin{bmatrix} \lambda \times F & 0 \newline 0 & 0 \end{bmatrix} \right)$ where $\lambda =(\frac{M}{N})^2$ is the scaling factor.
>
> The table below presents the results of cross-resolution generalization experiments using frequency-domain upsampling for FreD. These results reveal two key points as follows:
> * The performance of frequency-domain upsampling is comparable to that of spatial-domain upsampling.
> * DDiF still achieves higher performance across all metrics, demonstrating its superior cross-resolution generalization capability.
>
> | Test resolution | Test network | Method | Upsampling          | Accuracy (↑) | Diff (↓) | Ratio (↓) |
> |-----------------|--------------|--------|---------------------|--------------|----------|-----------|
> | 256             | ConvNetD5    | FreD   | Bicubic             | 56.4         | 10.4     | 0.16      |
> |                 |              |        | Zero padding        | 53.8         | 13.0     | 0.19      |
> |                 |              | DDiF   | Coordinate Interpolation | **66.3**     | **5.7**      | **0.08**      |
> | | ConvNetD6    | FreD   | Bicubic             | 61.4         | 5.8      | 0.09      |
> |                 |              |        | Zero padding        | 61.8         | 5.0      | 0.07      |
> |                 |              | DDiF   | Coordinate Interpolation | **70.6**     | **1.4**      | **0.02**      |
> | 512             | ConvNetD5    | FreD   | Bicubic             | 41.6         | 25.2     | 0.38      |
> |                 |              |        | Zero padding        | 42.9         | 23.9     | 0.36      |
> |                 |              | DDiF   | Coordinate Interpolation | **58.7**     | **13.3**     | **0.18**      |
> | | ConvNetD7    | FreD   | Bicubic             | 54.4         | 12.4     | 0.19      |
> |                 |              |        | Zero padding        | 56.3         | 10.5     | 0.16      |
> |                 |              | DDiF   | Coordinate Interpolation | **69.0**     | **3.0**      | **0.04**      |
>
> [1] Dugad, Rakesh, and Narendra Ahuja. "A fast scheme for image size change in the compressed domain." IEEE Transactions on Circuits and Systems for Video Technology 11.4 (2001): 461-474.

---

> ### Author Response · Authors · 2024-11-22
> **Response to Reviewer Y1Uh [5/5]**
>
> >**Q3. [Comparison with Vanilla under Same number of Decoded instances]** *Can you also report for the performance of vanilla dataset distillation performance, with the same number of distilled images in table 1,2,3? If I understand correctly, right now you only present the performance matching the number of parameters, but it would a useful datapoint to consider what performance hit is given by parameterizing with DDiF as opposed to directly parameterizing the images.*
>
> As the reviewer pointed out, we report the additional experimental results to compare the performance of vanilla and DDiF under the same number of decoded synthetic instances. Under the IPC=1 setting, DDiF can decode 51 and 55 synthetic instances per class on 128 and 256 resolution, respectively. Therefore, we conducted experiments under two scenarios: 1) number of vanilla’s setting and 2) number of DDiF’s setting. The table below presents the results as follows:
> * When the number of decoded instances is small, we admit that there are some performance drop cases. However, we emphasize that 1) this performance drop does not larger than 2.3%p and 2) DDiF utilize much smaller budget. Moreover, it is interesting that DDiF achieves higher performance than vanilla in some cases. We believe that it is related to previous findings that input-sized parameterization includes superfluous or irrelevant information [2,3].
> * When the number of decoded instances becomes larger, DDiF clearly uses much less budget but achieves performance comparable to vanilla dataset distillation.
> These experimental results support that DDiF involves sufficient representational power while using a much smaller budget compared to the vanilla dataset distillation.
>
> | Resolution | Decoded instances per class | Methods         | Utilized budget | ImageNette | ImageWoof | ImageFruit |
> |------------|----------------------------------|-----------------|-----------------|-------------------------|------------------------|-------------------------|
> | 128        | 1                                | Vanilla (TM)    | 491,520          | 51.4 $\pm$ 2.3            | 29.7 $\pm$ 0.9           | 28.8 $\pm$ 1.2           |
> |            |                                  | DDiF            | 9,630            | 49.1 $\pm$ 2.0            | 29.4 $\pm$ 0.7           | 27.3 $\pm$ 1.3           |
> |            | 51                               | Vanilla (TM)    | 51 $\times$ 491,520     | 73.0 $\pm$ 0.7            | 42.8 $\pm$ 0.7           | 48.2 $\pm$ 0.7           |
> |            |                                  | DDiF            | 491,520          | 72.0 $\pm$ 0.9            | 42.9 $\pm$ 0.7           | 48.2 $\pm$ 1.2           |
> | 256        | 1                                | Vanilla (DM)    | 1,966,080         | 32.1                    | 20.0                   | 19.5                   |
> |            |                                  | DDiF            | 79,530           | 31.2 $\pm$ 0.8            | 21.2 $\pm$ 0.9           | 21.3 $\pm$ 1.5           |
> |            | 55                               | Vanilla (DM)    | 55 $\times$ 1,966,080    | 70.1 $\pm$ 1.0            | 37.5 $\pm$ 1.2           | 41.3 $\pm$ 0.8           |
> |            |                                  | DDiF            | 1,966,080         | 67.8 $\pm$ 1.0            | 39.6 $\pm$ 1.6           | 43.2 $\pm$ 1.7           |
> ---
> >**Q4. [Effect of Neural field architecture]** *Is it possible to produce a curve showing the effect of neural field architecture choices on the performance? For example, given a fixed parameter budget, how does the width of the neural field (for example), affect the distillation performance?*
>
> As mentioned in Section 4.3 of the main paper, we have already provided experimental results that explore the effect of neural field architecture choices, including ablation studies and the results presented in Figure 7. Specifically, Figure 7 shows a small performance degradation when the number of layers (a) and the network width (b) are increased. This is due to the fact that increasing the number of parameters in each synthetic neural field leads to a decrease in quantity. Additionally, we also report results when the structure of the neural field is changed to FFN (c). Despite these changes, all performance results are still better than that of the vanilla method, as indicated by the black dashed line at the bottom of the figure.
>
> [2] Lei, Shiye, and Dacheng Tao. "A comprehensive survey of dataset distillation." IEEE Transactions on Pattern Analysis and Machine Intelligence (2023).
>
> [3] Yu, Ruonan, Songhua Liu, and Xinchao Wang. "Dataset distillation: A comprehensive review." IEEE Transactions on Pattern Analysis and Machine Intelligence (2023).

---

> ### Author Response · Authors · 2024-12-02
> **Reminder for Reviewer Y1Uh**
>
> Dear Reviewer Y1Uh,
>
> We sincerely appreciate your time and effort in reviewing our manuscript.
>
> We fully understand you are busy, but we would like to know if our responses have adequately addressed your concerns.
>
> As the discussion period is ending, we kindly ask the reviewer to consider our responses and the revised version, where we have tried to incorporate your feedback as thoroughly as possible.
>
> Please do not hesitate to let us know if you have any additional concerns or questions.
>
> Again, we appreciate the reviewer's constructive feedback.
>
> Best Regards,

---

### Official Review · Reviewer_Rr6g · 2024-11-04

**Soundness:** 2
**Presentation:** 3
**Contribution:** 3
**Rating:** 6
**Confidence:** 4

**Summary:**

This paper proposes a parameterization-based dataset distillation method called DDIF. The authors enhance traditional parameterization-based methods by introducing a neural field representation. According to the authors, their contributions lie in the theoretical analysis of neural fields for parameterized dataset distillation and the performance gains achieved by their proposed method. The experiments conducted on ImageNet subset and cifar10/100 demonstrates the outstanding performance of DDiF as stated in this paper. Besides the performance, the authors claim that the advantage of their DDiF is the coding efficiency introduced by their neural field representation. Additionally, DDiF is well-suited to datasets with varying resolutions due to its resolution-invariant nature.  The proposed DDiF has a good potential to boost the development of dataset distillation. However, there exists some unclear part of the proposed method.

**Strengths:**

1. The idea of leveraging neural field for dataset distillation is innovative.

2. The performance of the parameterized dataset distillation is unexpectedly high in the extreme scenario with IPC = 1.

3. The authors conduct a lot of experiments including cross-architecture experiments and abaltion studies on different distillation loss functions. Also, the performance of DDiF is outstranding.

**Weaknesses:**

I believe this paper has the potential to be a milestone in dataset distillation. However, it is difficult to follow, which raises several concerns about the proposed method.


1. The paper is not easy to understand:

(a) The algorithm 1 does not tell the most significant improvement introduced by the DDiF, as it is very close to a standard parameterization-based method. What part makes the DDiF outstanding? The design of the decoder network? or the Eq(2).

(b) I cannot understand the coding efficiency mentioned in line 188. While I recognize the resolution-invariant nature, I see no evidence supporting an advantage in coding efficiency. I hope the authors could provide numerical results here instead of rewriting the paragraph from line 188 to 204.

(c) This paper is 27 pages long, yet the authors should provide more details about their experiments. Important settings are missing, such as the calculation of the distillation budget, the architecture of the decoder network, the number of parameters, the training datasets, and specific training details. Additionally, the evaluation metrics for the reported performance are unclear—how many times were the evaluation experiments repeated?

(d) How should I interpret the sentence in line 370: ‘We utilize trajectory matching (TM) for 128 resolution and distribution matching (DM) for 256 resolution’? Why is the performance at 256 resolution significantly lower than at 128 resolution?


2. What is the purpose to repeat Theorem 1(Novello, 2022) and theorem 2 here? I think the reader would like to have an explanation on the outstanding performance of the neural field, instead of a relatively correct theorem and proof.

3. One critical issue with parameterized dataset distillation is the neglect of the decoder network. Only the encoded patterns are included in the distillation budget, while the decoder network is excluded. That explains the greater advantages of parameterized dataset distillation with IPC = 1, due to the marginal information carried by the decoder network. Such advantages will be smaller when ipc increases.

4 Followed by question 3, why did the authors neglect the experiments with IPC=50? Is it due to the performance degradation of parameterized dataset distillation in higher IPC?

**Questions:**

Most of my questions are listed in the weakness part. The other main concern is how to ensure a fair comparison across parameterized dataset distillation methods. As the decoder network will not be counted into the distillation budget, the architectures, number of parameters, and the training dataset will heavily influence the performance of parameterized dataset distillation. If we could not constrain the decoder network well, the easiest way to boost performance would be using a larger network, better architecture, and training the network with more datasets to boost its generalization ability.

---

> ### Author Response · Authors · 2024-11-22
> **Response to Reviewer Rr6g [1/4]**
>
> We appreciate the constructive reviews and valuable comments. We address the concerns below.
>
> >**W1(a). [Advantages of DDiF in Algorithm]** *The algorithm 1 does not tell the most significant improvement introduced by the DDiF, as it is very close to a standard parameterization-based method. What part makes the DDiF outstanding? The design of the decoder network? or the Eq(2).*
>
> Firstly, there are two main reasons why DDiF leads to outstanding dataset distillation performance. The first reason is that DDiF requires a smaller budget to generate a single synthetic instance. Consequently, for the same storage budget, DDiF can generate a larger number of synthetic instances, which enhances dataset distillation performance through increased diversity. Second, DDiF has superior expressiveness compared to previous parameterization methods. In a nutshell, DDiF improves both the quantity and quality in dataset distillation.
>
> Algorithm 1 in the main paper outlines the overall training procedure of DDiF. We presented Algorithm 1 to be in general structure of any dataset distillation because DDiF can be applied to various dataset distillation loss functions without requiring any additional loss terms specific to DDiF.
>
> After noting the generality and compatibility of DDiF, from an algorithmic perspective, the advantage of DDiF comes from decoding process of neural field. Basically, neural field takes coordinates as input and output quantities. To generate a single synthetic instance, a coordinate set, which is associated with pixel locations, is first fed into the neural field to obtain a quantity set, which is related with pixel values. This quantity set is then reshaped into the size of the original instance. Below algorithm illustrates this synthetic instance decoding process of DDiF, and we will add this algorithm in Appendix B.5. This parameterization by the parameters of neural field is much more efficient than previous decoding structure by 51~55 times in needed storage budget. Please refer to Table 7 and Appendix C.3 for further information.
>
> ```
> Algorithm: Decoding process of Synthetic instances in DDiF
> Input: Set of parameters of synthetic neural fields $\Psi$; Coordinate set $\mathcal{C}$
> Output: Set of decoded synthetic instances $X_S$
> 1. Initialize $X_S\leftarrow\emptyset$
> 2. For $j=1$ to $\vert\Psi\vert$
> 3.     Initialize $\tilde{x}_{j}$
> 4.     For $c \in \mathcal{C}$
> 5.         $\tilde{x}_{j}^{(c)}\leftarrow F_{\psi_{j}}(c)$
> 6.     $X_S\leftarrow X_S \cup \tilde{x}_j$
> ```
> ---
> >**W1(b). [Numerical Evidence on Coding Efficiency]** *I cannot understand the coding efficiency mentioned in line 188. While I recognize the resolution-invariant nature, I see no evidence supporting an advantage in coding efficiency. I hope the authors could provide numerical results here instead of rewriting the paragraph from line 188 to 204.*
>
> We have demonstrated that DDiF exhibit high coding efficiency in dataset distillation through extensive experiments. High coding efficiency refers to the ability to store a large amount of information with a small budget. In experiment section, DDiF outperforms baselines not only on various high-resolution image datasets but also on non-image modalities such as video, audio, and 3D voxel datasets. It indicates that DDiF effectively stores the necessary information from the original dataset. Especially, Figure 5 in the main paper highlights that DDiF achieves highest performance than previous parameterization methods when the number of decoded instances per class and utilized budget are same. Furthermore, in some scenarios shown in same figure, DDiF outperforms the baselines, even with a smaller budget. These extensive experimental results serve as direct evidence of DDiF’s high coding efficiency.
>
> In addition, we conducted a performance comparison with FreD and SPEED, which show strong performance on high-resolution image datasets. Under the IPC=1 setting, FreD and SPEED can decode 8 and 15 synthetic instances per class on 128 resolution, respectively. The table below presents the results as follows:
> 1)	Under the same number of decoded instances, DDiF achieves higher performance while using less budget compared to FreD and SPEED.
> 2)	Under the same budget, DDiF generates more decoded instances and still achieves superior performance.
> These results repeatedly support the claim that DDiF exhibits high coding efficiency.
>
> | |Decoded instances per class|Utilized Budget|Accuracy|
> |-|-|-|-|
> |FreD|8|491,520|66.8 $\pm$ 0.4|
> |SPEED|15|491,520|66.9 $\pm$ 0.7|
> |DDiF|8|77,040|67.1 $\pm$ 0.4|
> | |15|144,450|68.3 $\pm$ 1.1|
> | |51|491,520|72.0 $\pm$ 0.9|

---

> ### Author Response · Authors · 2024-11-22
> **Response to Reviewer Rr6g [2/4]**
>
> >**W1(c). [Details about Experimental Configuration]** *This paper is 27 pages long, yet the authors should provide more details about their experiments. Important settings are missing, such as the calculation of the distillation budget, the architecture of the decoder network, the number of parameters, the training datasets, and specific training details. Additionally, the evaluation metrics for the reported performance are unclear—how many times were the evaluation experiments repeated?*
>
> Most of the important experimental settings mentioned by the reviewer are already included in the main paper and appendix. The specific locations and descriptions for each setting are as follows:
> * Calculation of the distillation budget: We discussed how the distillation budget for DDiF is calculated in the main paper (Lines 278-287), and the specific values used in the experiments are provided in Table 7 in Appendix.
> * Architecture of the decoder network: This is mentioned in Appendix B.4. We utilized a multilayer perceptron with a sinusoidal activation function, specifically the SIREN model [1].
> * Number of parameters: The number of parameters in DDiF is equivalent to the storage budget. While we did not include the exact values, as noted earlier, it can be computed through the budget calculation.
> * Training datasets: The description of the training datasets used in the experiments, categorized by modality, is provided in Appendix B.1.
> * Specific training details: Key training details are available in Appendix B.4. Specific hyperparameter values, such as the learning rate, have been newly added in Table 23 in Appendix.
> * Number of repeated experiments: We conducted a total of 5 repeated experiments (Lines 942-944).
> ---
> >**W1(d). [Experimental Setting on Resolution 128 and 256]** *How should I interpret the sentence in line 370: ‘We utilize trajectory matching (TM) for 128 resolution and distribution matching (DM) for 256 resolution? Why is the performance at 256 resolution significantly lower than at 128 resolution?*
>
> Trajectory matching (TM) is well-known to have a high computation cost even for small-resolution datasets [2]. Therefore, for large-resolution datasets, Gradient matching (DC) or Distribution matching (DM), which generally have lower computation costs compared to TM, are commonly used. To ensure a fair comparison with prior studies, we used TM for the 128 resolution and DM for the 256 resolution.
>
> We recognized that, despite the neural field used in the 256 resolution experiment having sufficient expressiveness, it may still contain unnecessary parameters. Therefore, we conducted additional experiments by reducing the size of the neural field and increase the diversity. As a result, we achieved better performance than what was reported in the submitted paper. However, the performance still falls short compared to the 128 resolution. The reasons for this include: 1) the general observation that DM tends to perform worse than TM [3], and 2) the lack of sufficient hyperparameter tuning. We updated the performance on 256 resolution in the revised paper.
>
> | |ImageNette|ImageWoof|ImageFruit|ImageYellow|ImageMeow|ImageSquawk|
> |-|-|-|-|-|-|-|
> |Reported  | 63.5 $\pm$ 0.6 | 37.1 $\pm$ 1.4 | 40.5 $\pm$ 0.6 | 58.3 $\pm$ 1.8 | 43.5 $\pm$ 1.5 | 58.8 $\pm$ 1.1 |
> |Updated | 67.8 $\pm$ 1.0 | 39.6 $\pm$ 1.6 | 43.2 $\pm$ 1.7 | 63.1 $\pm$ 0.8 | 44.8 $\pm$ 1.1 | 67.0  $\pm$ 0.9|
>
> [1] Sitzmann, Vincent, et al. "Implicit neural representations with periodic activation functions." Advances in neural information processing systems 33 (2020): 7462-7473.
>
> [2] Cui, Justin, et al. "Scaling up dataset distillation to imagenet-1k with constant memory." International Conference on Machine Learning. PMLR, 2023.
>
> [3] Cazenavette, George, et al. "Dataset distillation by matching training trajectories." Proceedings of the IEEE/CVF Conference on Computer Vision and Pattern Recognition. 2022.

---

> ### Author Response · Authors · 2024-11-22
> **Response to Reviewer Rr6g [3/4]**
>
> >**W2. [Explanation of Theoretical Analysis]** *What is the purpose to repeat Theorem 1(Novello, 2022) and theorem 2 here? I think the reader would like to have an explanation on the outstanding performance of the neural field, instead of a relatively correct theorem and proof.*
>
> The main goal of the theoretical analysis is to provide a framework for comparing parameterization methods, which have been compared solely based on performance in general, through expressiveness (specifically, the size of the feasible space). Observations 1 and 2 indicate that a larger feasible space for decoded synthetic instances through input-sized parameterization or parameterization results in lower dataset distillation loss. Theorem 1 [4] demonstrates the expressiveness of a special case of neural field. Based on Theorem 1, we propose Remark 1 and Theorem 2 to show that DDiF has higher expressiveness than prior work (FreD). In conclusion, we claim that DDiF can achieve a lower dataset distillation loss than FreD since it has higher expressiveness.
>
> Additionally, in Appendix D.1, we provide a detailed comparison of the expressiveness of DDiF and FreD, further supporting the claim that DDiF has higher expressiveness than FreD. The main points are as follows:
> * While FreD can only tune amplitude, DDiF can tune amplitude, frequency, phase, and shift.
> * FreD represents a finite sum of fixed frequencies, whereas DDiF represents an infinite sum of tunable frequencies.
> * FreD is a discrete function, while DDiF is a continuous function.
> ---
> >**W3. [Budget Calculation of Parameterization Methods]** *One critical issue with parameterized dataset distillation is the neglect of the decoder network. Only the encoded patterns are included in the distillation budget, while the decoder network is excluded. That explains the greater advantages of parameterized dataset distillation with IPC = 1, due to the marginal information carried by the decoder network. Such advantages will be smaller when ipc increases.*
>
> The parameterization method typically consists of a combination of a code and a decoding function to form a synthetic dataset, with both being the target of learning and storage. Therefore, the storage budget of a parameterization method is calculated based on the number of total parameters that make up the code and decoding function. For a fair comparison, the structure of the code and decoding function is set such that the number of total parameters is smaller than the storage budget of input-sized parameterization methods (i.e., Number of classes × Instances per class (IPC) × Size of a single instance). To ensure a fair comparison, even with a very small budget (IPC=1), the parameterization method uses 1) non-parametric [5,6], 2) lightweight decoder networks [7,8], or 3) an open-access pretrained model [9] as the decoding function.
>
> Similar to previous studies, DDiF constructs a synthetic dataset using a combination of a coordinate set (code) and a synthetic neural field (decoding function). However, as mentioned in Section 3.2 in the main paper, the coordinate set can be easily obtained by determining the size of the decoded instance, so it does not require learning or storage. Therefore, DDiF only learns and stores the parameters of the synthetic neural field that correspond to the decoding function. This “decoder-only” framework structurally differs from conventional parameterization methods. In conclusion, the storage budget of DDiF is calculated based on the number of total parameters of the synthetic neural field, and the structure of the synthetic neural field (such as width and number of layers) is adjusted to ensure that its storage budget is smaller than that of input-sized parameterization methods.
>
> [4] Novello, Tiago. "Understanding sinusoidal neural networks." arXiv preprint arXiv:2212.01833 (2022).
>
> [5] Kim, Jang-Hyun, et al. "Dataset condensation via efficient synthetic-data parameterization." International Conference on Machine Learning. PMLR, 2022.
>
> [6] Shin, Donghyeok, Seungjae Shin, and Il-Chul Moon. "Frequency domain-based dataset distillation." Advances in Neural Information Processing Systems 36 (2024).
>
> [7] Liu, Songhua, et al. "Dataset distillation via factorization." Advances in neural information processing systems 35 (2022): 1100-1113.
>
> [8] Wei, Xing, et al. "Sparse parameterization for epitomic dataset distillation." Advances in Neural Information Processing Systems 36 (2024).
>
> [9] Cazenavette, George, et al. "Generalizing dataset distillation via deep generative prior." Proceedings of the IEEE/CVF Conference on Computer Vision and Pattern Recognition. 2023.

---

> ### Author Response · Authors · 2024-11-22
> **Response to Reviewer Rr6g [4/4]**
>
> >**W4. [Experimental Results on Large IPC]** *Followed by question 3, why did the authors neglect the experiments with IPC=50? Is it due to the performance degradation of parameterized dataset distillation in higher IPC?*
>
> The table below shows test accuracies (%) on CIFAR-10, CIFAR-100, and ImageNet (128) under IPC=50. We utilize trajectory matching (TM) for dataset distillation loss. The table below presents the results as follows:
> * As with IPC=1,10, DDiF achieves performance improvement on TM. It shows that DDiF also works well on large IPC setting.
> * DDiF under IPC=50 achieves higher performance than DDiF under IPC=1,10. It shows that there is no performance degradation as budget increase.
> * DDiF achieves highly competitive performances (Best or Second best) with previous studies.
>
> | IPC=50| CIFAR-10 | CIFAR-100 | ImageNette |
> |---|----------|-----------|------------|
> |TM| 71.6 $\pm$ 0.2 | 47.7 $\pm$ 0.2      | 72.8 $\pm$ 0.8 |
> |FRePo| 71.7 $\pm$ 0.2 | 44.3 $\pm$ 0.2      | - |
> |IDC| 74.5 $\pm$ 0.2 | -         | -          |
> |FreD| 75.8 $\pm$ 0.1    | 47.8 $\pm$ 0.1      | - |
> |HaBa| 74.0 $\pm$ 0.2     | 47.0 $\pm$ 0.2      | - |
> |RTP| 73.6 $\pm$ 0.5     | -         | -          |
> |HMN| 76.9 $\pm$ 0.2     | 48.5 $\pm$ 0.2      | - |
> |SPEED| **77.7** $\pm$ 0.4     | *49.1* $\pm$ 0.2      | - |
> |NSD| 75.2 $\pm$ 0.6     | -         | -  |
> |DDiF| *77.5* $\pm$ 0.3     | **49.9** $\pm$ 0.2      | **75.2 $\pm$ 1.3**|
> ---
> >**Q1. [Fair Comparison across Parameterized Dataset Distillation Methods]** *Most of my questions are listed in the weakness part. The other main concern is how to ensure a fair comparison across parameterized dataset distillation methods. As the decoder network will not be counted into the distillation budget, the architectures, number of parameters, and the training dataset will heavily influence the performance of parameterized dataset distillation. If we could not constrain the decoder network well, the easiest way to boost performance would be using a larger network, better architecture, and training the network with more datasets to boost its generalization ability.*
>
> We would like to ask reviewer to read last paragraph of Section 3.2 and Appendix B.4. These sections already provide the information for budget calculation of DDiF. As mentioned in response to weakness 3 (W3), the storage budget of a parameterization method is calculated based on the number of total parameters that make up the code and decoding function. Additionally, for a fair comparison, the structure of the code and decoding function is set such that their total number of parameters is smaller than the storage budget of input-sized parameterization methods. We calculated the budget in the fair comparison to the previous studies by including necessary inputs (which becomes none because these inputs are coordinate set in DDiF and it can be easily obtained if the shape of the decoded instance in defined) and parameters of decoding functions.
>
> Herein, we will provide a more specific explanation of DDiF’s budget calculation for clarify. DDiF uses a single synthetic neural field to generate one decoded instance, with each synthetic neural field utilizing an MLP with sine activation. Therefore, the budget required to generate a single instance is calculated as follows: $d_{0}(n+1)+\sum_{l=1}^{L-1} d_{l}(d_{l-1}+1)+m(d_{L-1}+1)$. Here, $n$ is the dimension of the coordinate space, $L$ is the number of layers, $d_{l}$ is the width of layer $l$, and $m$ is the dimension of the quantity. In the actual implementation, we used the same width for each layer i.e. $d_l=d$. For a fair comparison, we adjusted the width, layers, and decoded instances per class of the neural field so that “Total number of classes × Decoded instances per class × Budget per neural field” is smaller than the storage budget of input-sized parameterization methods (i.e., # of classes × instances per class (IPC) × size of a single instance). The specific values used in the experiment and their corresponding storage budget are outlined below. Please refer to Table 7 in Appendix for the specific configurations on other datasets.
>
> | Dataset            | IPC | Input dim (n) | Layers (L) | Width (d) | Output dim (m) | Budget per instance | Decoded instance per class | # of classes | Utilized storage budget | Allowed storage budget |
> |-|-|-|-|-|-|-|-|-|-|-|
> | CIFAR-10           | 1   | 2             | 2          | 6         | 3               | 81                  | 37                              | 10            | 29,970                  | 30,720                  |
> | ImageNet (128)     | 1   | 2             | 3          | 20        | 3               | 963                 | 51                              | 10            | 491,130                 | 491,520                 |
> | ImageNet (256)     | 1   | 2  | 3 | 40        | 3   | 3,523                | 55  | 10  | 1,908,720   | 1,937,650  |

---

> > ### Comment · Reviewer_Rr6g · 2024-11-25
> > **I will raise my score**
> >
> > Thank you for the detailed responses. Most of my concerns have been addressed, and I will raise my score to 6.

---

> > > ### Author Response · Authors · 2024-11-25
> > > **Thanks for reviewer Rr6g**
> > >
> > > Thank you for taking the time to read our response, additional experimental results, and the updated manuscript.
> > >
> > > Also, we're glad to hear that most of your concerns have been addressed.
> > >
> > > If you have further questions or uncleared concerns, please feel free to let us know.
> > >
> > > Again, Thank you so much for your time and effort.

---

### Author Response · Authors · 2024-11-22
**Global response to Reviewers**

Dear Reviewers:

We would like to express sincere gratitude for the constructive comments on our work and thank you so much for your time and effort. We provide individual responses to each reviewer's concerns and update the manuscript to reflect the reviewer's comments. We highlight the changes in purple. Please check our response and feel free to post additional comments. We are willing to provide further explanation on any unclear point.

Best regards.

---

### Author Response · Authors · 2024-12-04
**Summary of Discussion Period**

Dear AC and Reviewers,

We sincerely appreciate the time, effort, and dedication you have devoted to reviewing our manuscript. The reviewers' comprehensive feedback has guided us in enhancing the clarity and quality of our research. We are grateful for the recognition from the reviewers for the **novelty** (`Rr6g`, `e7GQ`), **simplicity and ease of compatibility** (`Y1Uh`, `XRWD`), **high performance in extensive experiments including non-image domains** (`Rr6g`, `Y1Uh`, `e7GQ`), and **presenting a new experimental design**, cross-resolution generalization (`Y1Uh`, `e7GQ`) in our work.

The reviewers also raised some concerns, and we have attempted to address these concerns as thoroughly as possible. Below, we summarize the raised key concerns and our responses.

* Clarification of budget usage
    * **[Budget calculation]** The reviewers pointed out that the budget calculation is not given (`Rr6g`, `XRWD`). To ensure a fair comparison, we configured the hyperparameters in all experimental settings such that the utilized budget did not exceed that of input-sized parameterization. The formula for budget calculation and the corresponding hyperparameter values were already included in the initial manuscript. During the discussion period, we provided additional explanations and detailed calculation results.
    * **[Number of decoded instances]** The reviewers suggested specifying the number of decoded instances for the proposed method (`e7GQ`, `XRWD`). The proposed method generates more decoded instances within the same budget due to its high coding efficiency. During the discussion period, we specified the exact increment in the number of decoded instances achieved by the proposed method.
* Additional Experiments
    * **[Large IPC setting]** The reviewer inquired whether the proposed method performs well even in large IPC setting (`Rr6g`, `XRWD`). We demonstrated that the proposed method is sufficiently effective even in large IPC setting through performance comparisons across various datasets. Furthermore, we showed that applying the soft label technique, which is used in the state-of-the-art methods (RDED, DATM), allows the proposed method to achieve higher performance than SOTA methods with a significant margin (`XRWD`).
    * **[Fixed number of decoded instances]** To support the high coding efficiency (`Rr6g`) and expressiveness (`Y1Uh`) of the proposed method, and to demonstrate that the performance improvement is not solely attributed to the increase in the number of decoded instances (`XRWD`), we additionally conducted the experiments under the fixed number of decoded instances. In this setting, the proposed method demonstrated highly competitive or even superior performance while using significantly less budget compared to input-sized and previous parameterization methods (FreD, SPEED).
    * **[Additional baselines in cross-resolution generalization]** The reviewer requested a comparison between downsampling of the full dataset and frequency-domain upsampling in the cross-resolution generalization (`Y1Uh`). We conducted additional experiments and demonstrated that the proposed method still exhibits strong cross-resolution generalization.
    * **[Comparison with bases-coefficient decomposition]** The reviewer requested a performance comparison with the based-coefficient decomposition method (RTP) (`e7GQ`). We have already included the performance comparison for CIFAR-10 and CIFAR-100 in the initial manuscript, and we explained that the proposed method achieves high performance in the ImageNet-Subset (128) experiments we additionally conducted.
* Theoretical analysis
    * **[Additional explanation]** We provided additional explanations regarding the objectives, significance, and contributions of the theoretical analysis. (`Rr6g`, `Y1Uh`)
    * **[Empirical evidence]** We provided additional experimental evidence to support the theoretical analysis. (`Y1Uh`, `XRWD`)
* Further explanation
    * We provided further clarification regarding the experimental setting (`Rr6g`, `e7GQ`, `XRWD`), algorithm (`Rr6g`), computational cost (`Y1Uh`), ablation study (`Y1Uh`), comparison with previous studies (`e7GQ`), and cost of data augmentation (`e7GQ`).

We have incorporated the points raised by the reviewer in the first round into the revised manuscript, which has been uploaded, and the comments mentioned in the second round will be reflected in the final manuscript. We hope that the summary above helps provide a deeper understanding of the proposed method and addresses the reviewers’ concerns.

Once again, we sincerely express our gratitude to the AC and all reviewers.

Best Regards,

---

### Meta-Review · Area_Chair_76pV · 2024-12-19

**Metareview:**

This paper introduces a novel parameterization-based dataset distillation (DD) approach by leveraging neural fields to factorize data. The proposed method offers a fresh perspective for extending DD beyond image data, effectively generalizing to diverse modalities such as video, audio, and 3D point clouds. The approach demonstrates impressive performance under high compression ratios, achieving notable improvements in various tasks. While its effectiveness under large IPC settings remains a limitation, this does not detract from the contributions the work makes to the dataset distillation community. Therefore, I recommend accepting this paper.

**Additional Comments On Reviewer Discussion:**

This paper’s novel dataset distillation via neural fields shows strong performance and is recommended for acceptance.

---

### Decision · Program_Chairs · 2025-01-22

Accept (Poster)